# Seismic detection of a 600-km solid inner core in Mars

Huixing Bi[1,2], Daoyuan Sun[1,2]✉, Ningyu Sun[2,3], Zhu Mao[1,2], Mingwei Dai[1,2] &
Douglas Hemingway[4]

For rocky planets, the presence of a solid inner core has notable implications on the composition and thermal evolution of the core and on the magnetic history of the planet[1–3]. On Mars, geophysical observations have confirmed that the core is at least partially liquid[4–7], but it is unknown whether any part of the core is solid. Here we present an analysis of seismic data acquired by the InSight mission, demonstrating that Mars has a solid inner core. We identify two seismic phases, the deep core-transiting phase, PKKP, and the inner core boundary reflecting phase, PKiKP, indicative of the inner core. Our inversions constrain the radius of the Martian inner core to about $613 \pm 67$ km, with a compressional velocity jump of around 30% across the inner core boundary, supported by additional inner-core-related seismic phases. These properties imply a concentration of distinct light elements in the inner core, segregated from the outer core through core crystallization. This finding provides an anchor point for understanding the thermal and chemical state of Mars. Moreover, the relationship between inner core formation and the Martian magnetic field evolution could provide insights into dynamo generation across planetary bodies.

The existence of an inner core (IC) within a planet holds importance in planet evolution, as its growth affects the thermal state and dynamo processes of the planet[1,8]. As a consequence of core crystallization, the current size and physical state of the IC serve as direct indicators of the thermal and compositional properties of the planet. Although Earth has a well-established IC[9], decisive identification of an IC has not been made for other planetary bodies, apart from the Moon[10].

The core structure of Mars may differ markedly from that of the Earth, given their distinct magnetic histories. Despite its strongly magnetized ancient crust, the current lack of a global dynamo field on Mars implies the cessation of a past dynamo[11–13]. The processes occurring within the Martian core, including potential crystallization, and their implications for dynamo activity remain poorly understood[3,14]. Geodetic observations rule out a fully solid core[4,5,15], and cosmochemistry data suggest that the core is probably liquid, given its content of light elements[16–18]. Recent findings from the InSight mission to Mars[19] further confirm the existence of a liquid Martian core, with detections of seismic reflected wave at the core–mantle boundary (CMB)[7], core-transiting wave[20] and constraints from the free core nutation[6]. The discovered low-density liquid core implies the presence of light elements such as carbon (C), oxygen (O) and hydrogen (H) apart from sulfur (S) (refs. 7,20), raising the possibility of crystallization under Martian core condition[21]. However, the question of whether Mars possesses a solid IC remains unanswered.

The discovery of the IC-transiting seismic phase (PKIKP), which passes the core shadow caused by the liquid outer core (OC), has been critical in defining the IC structure of Earth, whereas normal modes and PKJKP observations further confirm the presence of a solid IC[9,22–26].

However, on Mars, the observation of normal modes is limited by a low signal-to-noise ratio (ref. 27). Moreover, even the most distant event S0976a, with an epicentral distance ($\Delta$) of $146 \pm 7°$ from InSight[28,29], remains situated within the shadow zone[7,20]. At a distance less than 40°, at which most Martian seismicity is located[29], core phases PKP-PKP (P′P′) and PKKP (detailed phase names can be found in the caption of Fig. 1), with extra reflection at the nearly antipodal surface and the CMB, respectively (Fig. 1a), could sample the deepest parts of the Martian core twice. Therefore, these short-distance data offer a unique opportunity to test the existence of the IC. Here we present observations of these core phases, particularly highlighting the detection of the reflection wave at the inner core boundary (ICB) and the transiting wave through the IC, which provides robust constraints on the size and physical properties of the IC.

## Evidence for IC phases

To better capture weak core phases and minimize contamination from the mantle phases (Supplementary Information section A2), we leverage the substantial number of low-frequency marsquakes at $\Delta$ of 27°–40° (Fig. 1c and Extended Data Table 1) and treat them as a source array for array analysis[30]. By implementing array analysis on the waveform or envelope data (Methods), we can generate vespagrams, in which a robust energy peak at a certain time represents coherent seismic phases with the characterized horizontal slowness. Thus, different phases with different ray parameters can be effectively isolated (Supplementary Information section A2), as supported by the robust identification of multiple mantle phases (Supplementary Fig. 9).

[1]National Key Laboratory of Deep Space Exploration/School of Earth and Space Sciences, University of Science and Technology of China, Hefei, China. [2]CAS Center for Excellence in Comparative Planetology, Hefei, China. [3]State Key Laboratory of Precision Geodesy, University of Science and Technology of China, Hefei, China. [4]Institute for Geophysics, Jackson School of Geosciences, University of Texas at Austin, Austin, TX, USA. ✉e-mail: sdy2014@ustc.edu.cn

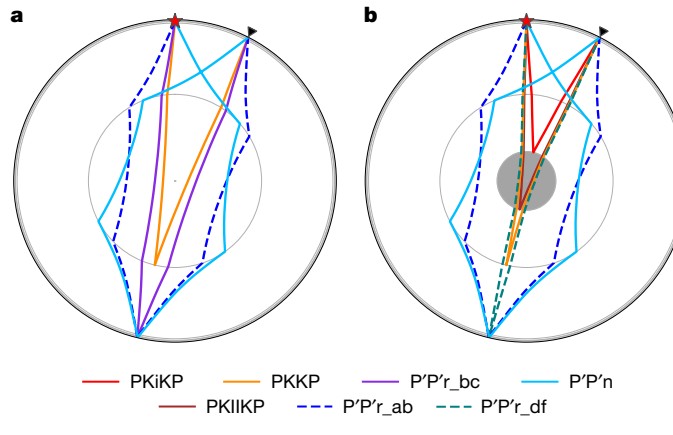

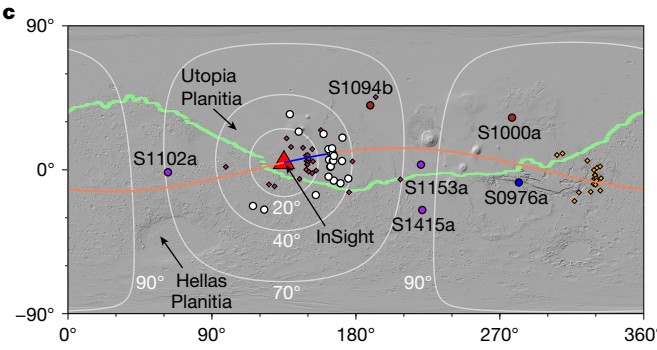

**Fig. 1 | Raypaths for different Martian core seismic phases. a,b,** Models without (**a**) and with (**b**) an IC are compared. Here, P'P'r represents rays leaving from the event towards the station along the major arc and sampling different depths of the core, whereas P'P'n represents a ray taking off along the minor arc. In the presence of an IC, the PKiKP phase reflects at the ICB (red curve in **b**). In **b**, the labelled PKKP phase transiting the IC should be PKIKKIKP, but we use PKKP here for consistency with a model without IC. Moreover, an extra P'P'r_df phase (green dashed curve) transits the faster IC and arrives much earlier than the P'P'r_ab (blue dashed curve). The red star and black triangle denote a marsquake at a depth of 33 km and a seismometer at an epicentral distance of 27°, respectively. **c**, Locations of the marsquakes used in vespagram analysis (Methods) (white circles), two impacts (red circles), four large-distance events (magenta and blue circles) and the InSight seismometer (red triangle). The red and orange diamonds represent the bouncing points of PKiKP at the inner-core boundary and P'P' at the surface near the antipode of the InSight location (Fig. 1b), respectively. White lines mark equal-distance circles around InSight with epicentral distances of 20°, 40°, 70° and 90°. Most Martian seismicity is at the epicentral distance range of 27°–40° (blue-shaded region in Extended Data Fig. 1). The thin blue and orange lines denote the great circle paths for event S0235b PKiKP and PKKP phases, respectively. The locations of the marsquakes are plotted based on ref. 48. The heavy green line marks the equatorial dichotomy boundary on Mars based on ref. 49.

Although locating a marsquake with a single station could have large uncertainties in back-azimuth, its epicentral distance is relatively well determined[29], which makes the vespagram analysis feasible. Moreover, when taking P as the reference phase, the uncertainties of the event depth have negligible effects on both the travel time and slowness of the core phases (Supplementary Information section A2.5). Bootstrap resampling tests provide support for robust identifications of these core phases and enable us to estimate the uncertainties in travel time and slowness (Methods and Supplementary Information section A3). In each vespagram generated from the resampling tests, grids with energy exceeding 85% of the peak are assigned a value of 1. By analysing the occurrence percentage of these grids, we identify the phases with the highest occurrence and fit Gaussian functions to derive their mean travel times and slownesses and uncertainties (Fig. 2e). Examples of P'P'r_ab and P'P'n are shown in Supplementary Fig. 17, with travel time

of 1,835 ± 4 s and 2,008 ± 3 s, slowness of −12.1 ± 0.7 s per degree and −5.0 ± 0.6 s per degree, relative to P, respectively. These arrivals fall within the predicted travel times and are slightly below the predicted slowness of −10.8 to −9 s per degree and −4.0 to −3.7 s per degree for both phases, derived from the available seismic core models[7,20,31,32].

Despite the evidence provided by vespagrams for observing core phases, the direct identification of these phases in individual waveforms offers the most compelling confirmation, as demonstrated by the detections of ScS (ref. 7) and SKS (ref. 20) phases. Therefore, we also use two complementary approaches[7,20,33]: (1) polarized waveforms and their time-domain envelopes (filter bank); and (2) frequency-dependent polarization analysis (FDPA) (Methods), to detect core phases of individual events. The detections of individual events are shown in Supplementary Information section B2, which further validates the identified core phases in the vespagrams.

Strong arrivals with the PKKP slowness can be observed only at a time window of 50–200 s earlier than predictions for models featuring a pure liquid core (Fig. 2a,b). In particular, a strong signal appears at about 1,340 s. An example of PKKP identification for event S0235b is shown in Fig. 2g. In a model with a molten silicate layer (MSL) above the core[31,32], two PKKP arrivals are anticipated: the earlier reflection at the CMB (PKKP_CMB) and the later reflection at the top of the MSL (PKKP_MSL), with an interval of around 60 s (Supplementary Fig. 12). Thus, two arrivals at 1,290 ± 3 s and 1,341 ± 5 s in Fig. 2b may correspond to PKKP_CMB and PKKP_MSL, respectively. Although similar two arrivals are also observed for several individual events (Supplementary Fig. 11), their relatively low amplitudes make the MSL interpretation less definitive. Nevertheless, the significantly early arrival of PKKP indicates a much faster core towards the centre, such as a velocity gradient of 0.25 km s⁻¹ faster over 100 km in the central 880 km of the core compared with the shallow core (Supplementary Information section A3.2.2.1.3). However, having a much steeper velocity gradient towards the centre may be difficult for a pure liquid core[16,34–36]. Alternatively, an IC with a higher velocity, which is sampled by the PKKP, provides a more feasible explanation, further supported by the identification of other IC-related phases.

If a fast IC does exist, a reflection from the interface of the IC and OC (ICB), that is, PKiKP phase (Fig. 1b), would be anticipated. The PKiKP phase has been used as the decisive phase for determining the presence of the lunar IC[10]. In the vespagram, we find a prominent arrival at about 604 ± 2 s after P (Fig. 2e). This arrival is characterized by a relative slowness of −6.5 ± 0.6 s per degree, aligning with the anticipated slowness range (−7.4 to −6.4 s per degree) for PKiKP, considering an IC radius ranging from 0 km to 900 km. Bootstrap tests and waveform analysis for individual events (Fig. 2h and Extended Data Fig. 2) confirm the robustness of this PKiKP phase (Supplementary Information section A3.3). Moreover, the vespagram analysis following the deconvolution of the PKiKP waveform (Methods) shows that PKKP and PKiKP exhibit opposite polarities and an amplitude ratio of about 0.5 (Extended Data Fig. 3), which is consistent with the prediction and further supports the identification of these core phases.

## Inversion for the IC structure

By having the travel times of PKKP transiting the IC, PKiKP reflecting at the ICB and OC transiting phases (that is, P'P'r_ab and P'P'n), we can invert for the size and velocity of the IC. Here, rather than performing an entire Martian velocity structure inversion by combining our new measurements with those from previous studies, we focus on only inverting the P-wave velocity of the core (Methods). To assess sensitivity to mantle structure, we test a range of possible mantle models and obtain consistent results (Supplementary Information section A4), suggesting that jointly inverting for mantle structure does not have a large impact on the final core model. Furthermore, we use the differential time between the core phase and direct P phase, rather than

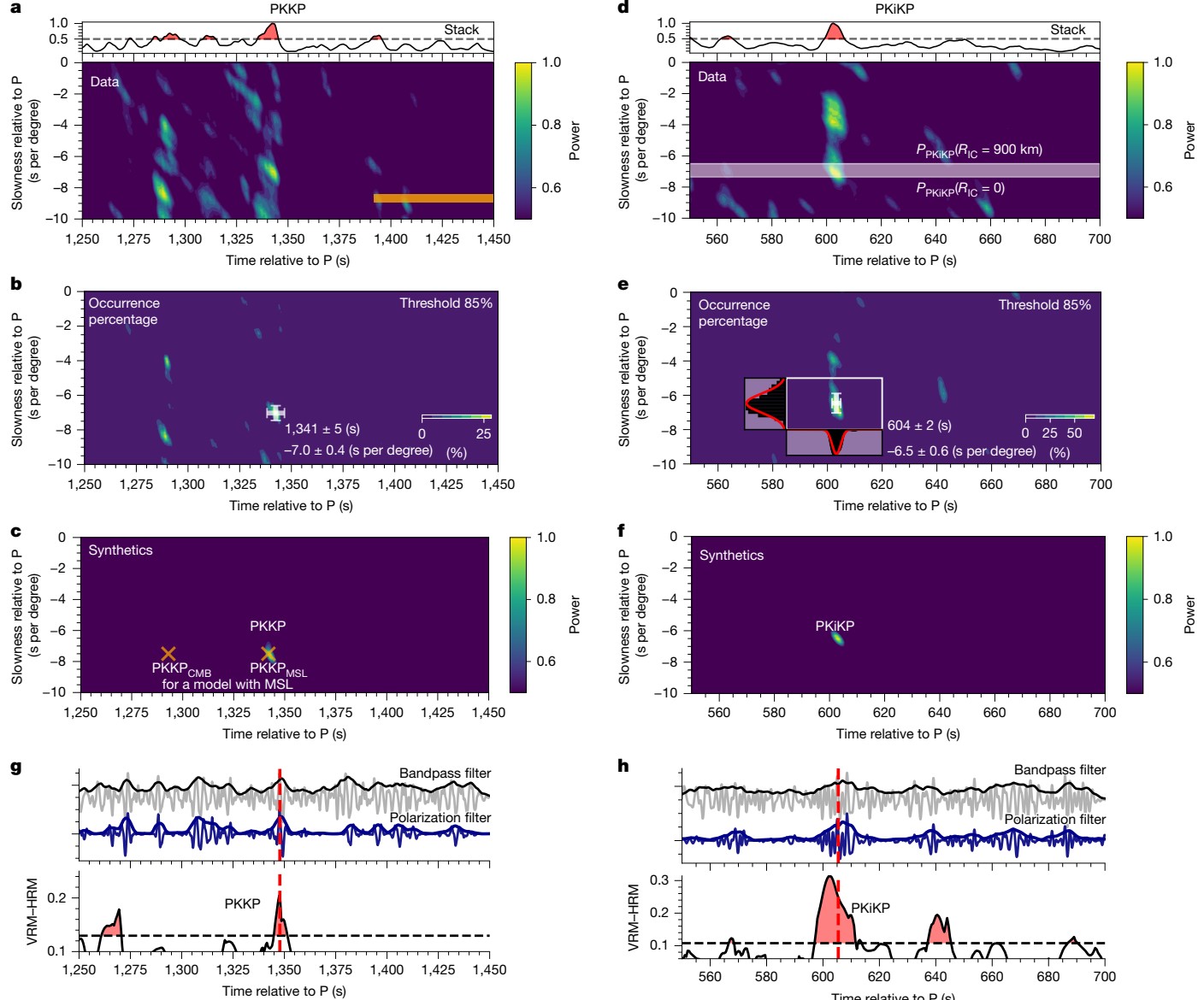

**Fig. 2 | Identifications of Martian IC phases. a**, PKKP vespagram constructed from 23 events with distance <40° in Extended Data Table 1, showing two distinct peaks at (about 1,290 s, −8 s per degree) and (about 1,340 s, −7 s per degree). The later arrival is interpreted as PKKP or alternatively as $PKKP_{MSL}$, a reflection from the top of the proposed molten silicate layer overlying core[31,32]. Orange shading outlines predicted PKKP travel time or slowness at the reference distance of 29° for available velocity models. Top panel shows the envelope stack for the picked PKKP slowness in **b**. **b**, Occurrence percentage of grids with energy exceeding 85% of the peak value in the vespagram, with white cross marking the identified PKKP arrival and the associated $1\sigma$ errors from Gaussian fitting (Methods and Supplementary Information section A3.2.2.3). **c**, Synthetic vespagram for the BS_SKS_GD_IC model (Extended Data Table 2, case 8). Orange crosses denote predicted $PKKP_{CMB}$ and $PKKP_{MSL}$ for case 6. **d**–**f**, Same as

**a**–**c**, but for PKiKP. White shading in **d** shows the expected PKiKP slowness range for IC size of 0–900 km. The energy at (600 s, −4 s per degree) has a similar slowness to that of ScS (see Supplementary Information section A3.3 for possible origins of this arrival). **g**, Polarization analysis on event S0235b confirms PKKP identification: top, bandpass-filtered waveforms (grey line); middle, polarization-filtered waveforms (blue line) with envelopes; and bottom, vertical–horizontal summed FDPA intensity (VRM–HRM) (Methods), showing a strong polarized signal at about 1,348 s with large VRM–HRM. Red dashed lines mark the predicted travel time from the PKKP slowness in **c**. Black-dashed line marks the mean plus one standard deviation of VRM–HRM values in the 100-s window preceding PKKP. **h**, Same as **g** but for event S1015f, showing a PKiKP arrival at about 605 s.

the absolute travel times, to reduce the effects of possible location errors. With four sets of differential travel times, $\Delta T_{P'P'r\text{-}ab\text{-}P}$, $\Delta T_{P'P'n\text{-}P}$, $\Delta T_{PKKP\text{-}P}$ and $\Delta T_{PKiKP\text{-}P}$, we conduct Bayesian inversions to determine the P-wave velocity ($V_P$) structure of the core (Methods and Extended Data Fig. 4). Figure 3 shows an example of inversion from the vespagram measurements by adopting the SKS_GD[20] mantle velocity model. Results using various mantle velocity models and different inversion strategies are listed in Extended Data Table 2, which shows a mean IC radius ($R_{IC}$) of 613 ± 67 km (Extended Data Fig. 5). The consistent

inverted $R_{IC}$ using different mantle models primarily arises from jointly including both $\Delta T_{PKKP\text{-}P}$ and $\Delta T_{PKiKP\text{-}P}$ in the inversion, as PKKP and PKiKP share similar paths through both the mantle and the OC beneath the event and station (Fig. 1b). Our inverted IC size also agrees with the upper limit on the radius, estimated at around 750 km, based on the maximum depth reached by SKS for event S1000a (ref. 20). However, we recognize that three-dimensional mantle structures or uncertainties in the OC velocity could increase the uncertainty in the estimated IC size (Supplementary Fig. 32).

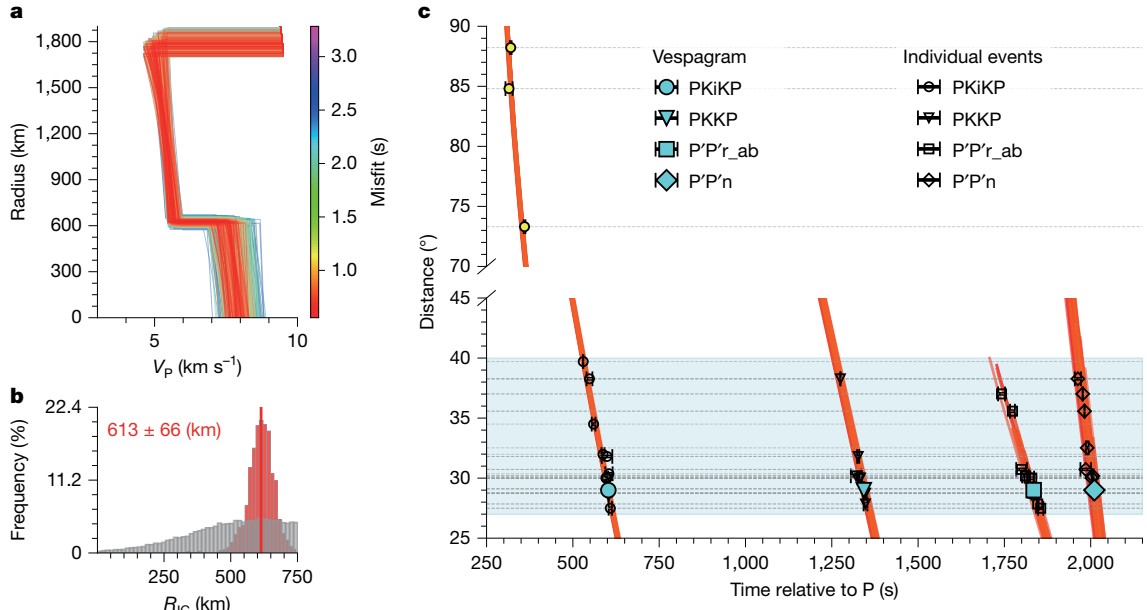

**Fig. 3 | Inversion for the seismic velocity of the Martian IC. a**, Example of case 1 from Extended Data Table 2. In this case, the P-wave velocity for the Martian core is inverted by adopting the SKS_GD model[20] as the mantle velocity model and using the M_vesp strategy (Methods). By fitting the travel time measurements from vespagram (four cyan solid symbols in **c**), five core parameters (Extended Data Fig. 4), including OC radius ($R_{OC}$), $V_P$ at the CMB ($V_{P\_CMB}$), the $V_P$ on the OC side at the ICB ($V_{P\_OC\_ICB}$), IC radius ($R_{IC}$) and the percentage of $V_P$ jump across the ICB ($\delta V_{P\_ICB}$) are inverted. The mean $V_P$ gradient in the IC is assumed to be the same as that in the OC. **b**, Marginal distribution of inverted $R_{IC}$. The red and grey histograms represent the marginal distribution of the inverted $R_{IC}$ and the priori distribution of the $R_{IC}$, respectively. Mean values and 85% confidence intervals are indicated with red text at the top. **c**, Observed (symbols) and predicted (lines) differential travel times for different phases. Open symbols and cyan solid symbols denote the individual event picks and vespagram measurements, respectively, with error bars indicating the uncertainties in travel times. Note that this example uses only the four travel times (cyan solid symbols) from the vespagram measurements for the inversion. Predicted travel times from 100 sampled models with the lowest travel time misfits in **a** are plotted. The light-blue-shaded region denotes the epicentral distance range of the events used for vespagram analysis and inversion. Three events with epicentral distances larger than 70° are not included in the inversion but are used to support the inverted Martian IC model.

The other three OC parameters, OC radius ($R_{OC}$), $V_P$ at the CMB ($V_{P\_CMB}$), the $V_P$ on the OC side at the ICB ($V_{P\_OC\_ICB}$), however, are less well constrained (Supplementary Information section A4.2 and Supplementary Information section B3). Strong trade-offs among these parameters suggest that a seismological core model derived solely from the limited measurements of core phases may be subject to large uncertainties. Nonetheless, the mean value of our inverted $R_{OC}$ of 1,799 ± 66 km (Supplementary Fig. 33), is similar to 1,780–1,810 km in ref. 20. Furthermore, as only PKKP travel time provides constraint on the IC velocity and its trade-offs with OC parameters, the percentage of $V_P$ jump across the ICB ($\delta V_{P\_ICB}$) is not well resolved. To address this, we adopt the OC parameters from the best geodynamical inverted model presented in ref. 20, which efficiently explains our P'P' measurements (Supplementary Fig. 42), and invert only for $R_{IC}$ and $\delta V_{P\_ICB}$. This inversion yields a $\delta V_{P\_ICB}$ of 32 ± 8%, equivalent to an IC $V_P$ of 7.3–8.3 km s$^{-1}$ (Supplementary Fig. 43). In the subsequent discussion, we use these IC parameters for the sake of simplicity.

## Further support for the IC model

IC phases at distances beyond the range used for the inversion are also observed. Here, to avoid possible influences from mantle heterogeneities on travel times, we do not include them in the inversion. Rather, we assess how our IC models predict these phases (Supplementary Information section A5). First, three events, S1102a (73.3°), S1153a (84.8°) and S1415a (88.2°) (Fig. 1c), all exhibit visible PKiKP phases occurring at ±10 s around the predictions (Extended Data Fig. 6a–c). Second, we observe the presence of P'P'r_df (Fig. 1b) across a wider range of epicentral distance. Should a high-velocity IC exist, one branch of the P'P'r could sample the IC as an early arrived P'P'r_df (Fig. 1b) and extend to much larger distance compared with models without an IC

(Extended Data Fig. 1). This is evident from an array analysis on the data within a distance range of 27°–40°, showing a robust energy peak aligned with the predicted arrival times and slowness of P'P'r_df (Extended Data Fig. 6d). Third, a bottom reflection from the ICB, named PKIIKP (Fig. 1b), can be identified in the vespagram, despite possible interference of the strong PcSScP (Extended Data Fig. 6h). However, we note that the amplitude of PKIIKP is half of PKKP in the synthetics, making its identification challenging.

Apart from seismic velocity, variation in density within the core plays an important part in constraining its composition. A notable aspect of this determination involves analysing the amplitude ratio between PKKP and PKiKP ($A_{PKKP}/A_{PKiKP}$) (Methods). Alongside the measurement from the vespagram analysis (Extended Data Fig. 3a,b), event S0235b offers concurrent, well-defined observations of both PKKP and PKiKP. To match the observed $A_{PKKP}/A_{PKiKP}$, a density jump across the ICB ($\delta\rho_{ICB}$) of 7 ± 5% is preferred (Extended Data Fig. 7), despite large uncertainties due to noise and potential anomalous structures at the CMB. Furthermore, the potential presence of IC anisotropy and scatterers, similar to those observed in Earth[9,22,37], could also affect the accuracy of determining $\delta\rho_{ICB}$. This small $\delta\rho_{ICB}$ is also supported by the need to reconcile the mean density and moment of inertia of Mars (Supplementary Information section A6).

## Implications for core composition and dynamics

Our identification of about 0.18 Mars radii solid IC, proportionally similar in size to 0.19 Earth radii IC (Fig. 4), confirms the existence of core crystallization. The obtained $V_P$ and density jump across the ICB provide further constraints on the composition of the Martian core. Our calculation suggests that a pure solid Fe–Ni IC cannot explain the observed IC properties, which require the presence of substantial light

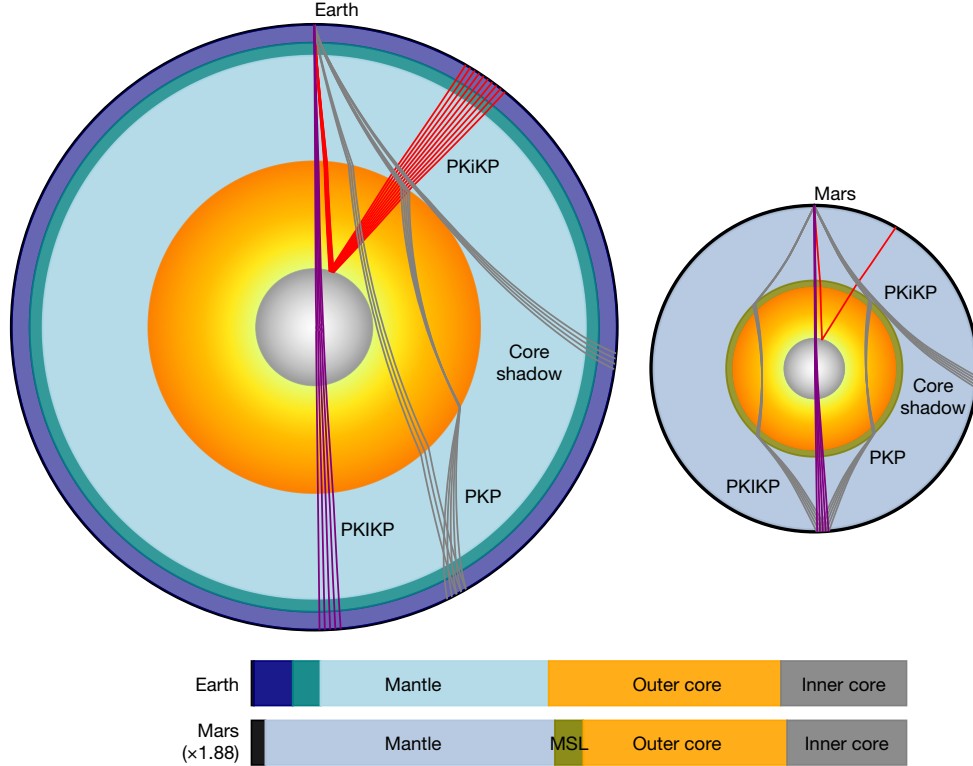

**Fig. 4 | Comparison between the interior structures of Earth and Mars.** With an IC, Mars appears as a scaled-down Earth, featuring proportional reductions in the IC, OC and mantle, and their corresponding core-transiting and reflecting phases are also similar.

elements in the core (Supplementary Information section A7). Based on cosmochemical and experimental constraints, S, C, O and H are candidate light elements in the Martian core[38,39]. However, the presence of H substantially depresses the melting point of Fe, potentially inhibiting the formation of a solid IC[21,40,41]. Although a small amount of H cannot be ruled out, its abundance is probably low, and our models focus on S, C and O. Previous experimental results suggest that the crystallization of an $Fe_3C$-dominated solid IC requires more than 4 wt% C (refs. 42,43), resulting in an Fe–S–O–C OC. Under these conditions, the Martian core is estimated to contain approximately 12–16 wt% S, less than 6 wt% O and 4.0–4.7 wt% C (ref. 21). However, this composition would, on crystallization, produce density and velocity jumps across the ICB of 20–27% and around 22%, respectively (Supplementary Fig. 49), which are inconsistent with our observations. By contrast, models with 12–16 wt% S, 6.7–9.0 wt% O and ≤3.8 wt% C (ref. 21) favour the crystallization of an FeO-rich solid IC under Martian core pressure–temperature conditions. Accounting for the partitioning of O and C between the OC and IC, the resulting density (3–9%) and velocity (24–31%) jumps across the ICB match our seismic observations (Extended Data Fig. 8). Meanwhile, the absolute velocity of an FeO-rich solid IC is also consistent with our observed seismic velocity. Therefore, an O-enriched core with distinct distributions of S and C between the OC and IC provides a possible explanation for the presence of a solid Martian IC with a radius of about 600 km. Although Martian core temperature estimates remain uncertain, the liquidus of the Fe–O–C–S system suggests that even if the core temperature is approximately 10% higher than current estimates, FeO can still crystallize to form a solid IC[21]. This high core temperature might also support the existence of an MSL at the CMB[31,32]. However, given the limited constraints on light element partition, elasticities and melting behaviour under Martian IC conditions, alternative compositional scenarios remain possible and should be considered in future interpretations. Moreover, large uncertainties in estimating the velocity and especially the density jumps across the ICB further complicate precise estimates of core composition.

Although a solid IC implies past or even present core crystallization, this is not necessarily incompatible with the absence of a dynamo today because a crystallization-driven dynamo depends on many factors, including the rate and style of core crystallization and the way light elements are partitioned between the solid and liquid phases[14,18,44] (Supplementary Information section A7.2). Our results are consistent with previously considered scenarios in which the Martian core initially cooled rapidly, driving an early thermal dynamo[11–14], but is now cooling too slowly to drive thermal convection and, in spite of ongoing crystallization, is unable to drive a dynamo now due to some combination of the crystallization proceeding too slowly or a lack of density contrast associated with the formation (or remelting) of crystals in the core[3,14,18,44–47] (Supplementary Information section A7.2). Further understanding the formation of the Martian IC and its implications for dynamo evolution requires more detailed modelling and improved knowledge of Martian core composition as well as mantle viscosity. These investigations are important not only for clarifying Martian interior dynamics but also for understanding dynamo generation in other planetary bodies such as Mercury and Ganymede.

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

# Methods

## Data preprocess

To avoid pitfalls related to transient one-sided pulses, typically referred to as glitches[50], we first use two glitch-removal algorithms sequentially: the MPS, followed by the UCLA method proposed in ref. 51. Next, the UVW components are rotated to the ZNE components[52]. A Butterworth bandpass filter is then used to filter the glitch-reduced data to 0.2–0.8 Hz to avoid the 1-Hz tick noise, high-frequency idiosyncratic signals and long-period noises. Finally, we apply a time-domain polarization filter method[53], routinely used in planetary seismology[7,10,20,54–56], to suppress the non-linearly polarized noise and enhance the linearly polarized body wave phases.

## Array analysis

Vespagram analysis[30,57] can efficiently distinguish different seismic phases based on their unique travel time and slowness. By stacking the traces shifted along a specified slowness, a given phase is enhanced. Vespagram analysis has been applied to the detection of core-reflected phase ScS on Mars[7]. Unlike typical receiver array analyses for earthquakes, which allow for direct stacking waveforms to preserve important information about the amplitude and polarity, the vespagram analysis here is based on a source array[30]. Variations in marsquake properties, such as focal mechanism, magnitude and depth, combined with complicated propagation effects from possible three-dimensional structures, cause large differences in absolute amplitudes and waveforms among different events, making vespagram analysis more challenging.

To minimize absolute amplitude differences among different marsquakes, we normalize each polarization-filtered trace within the targeted time window as

$$X_{\text{Normalized}} = \frac{X - \mu}{\sigma}, \tag{1}$$

where $\mu$ and $\sigma$ are the mean value and standard deviation of waveform data $X$, respectively. In the subsequent section on determining the amplitude and polarity of PKKP and PKiKP, we also normalize the entire waveform record using the amplitude of the reference phase or by deconvolving the reference phase.

Both polarization-filtered waveforms from the vertical component and their envelopes (Supplementary Information section A3.2.2.1) are tested as inputs for vespagram analysis. To improve the slowness resolution, non-linear stacking methods are applied. For waveform data, we test both fourth-root vespagram and phase-weighted stack (PWS)[30], whereas envelope data use only the fourth-root vespagram. It is important to note that differences in waveform polarities and source durations among events may reduce the stacking energy even when they are aligned with a specified slowness. Vespagrams for synthetic P waveforms, assuming different source mechanisms and durations for different events, demonstrate that envelope data produces stronger and more focused energy than waveforms. Moreover, owing to the variation and incoherence in waveforms across different events, PWS cannot produce focused energy (Supplementary Fig. 8).

Moreover, FDPA (see section 'Analysis for individual events' for details) shows distinct differences in vertical (VRM) and horizontal (HRM) rectilinear motion between compressional and shear waves. To further enhance vertically polarized signals, we test multiplying the VRM−HRM to the polarization-filtered waveforms and conduct vespagram analysis subsequently (Supplementary Fig. 24).

Here we select 23 low-frequency marsquakes (Extended Data Table 1) and focus on using the fourth-root vespagram to detect core phases (Supplementary Information section A3.2.2). The traces are aligned with the direct P provided in the MQS[29]. Therefore, the travel time and slowness derived from the vespagram analysis are relative to those of the direct P. The reference epicentral distance is set at 29°. The grid exhibiting the highest coherent energy within the targeted time-slowness window, and demonstrating the most consistent presence in the bootstrap tests, is then picked as our candidate phase.

## Assessment of vespagram result robustness

To address uncertainties and assess the reliability of our vespagram results, bootstrap resampling test[58] is used. We conduct three types of bootstrap resampling tests.

In type I, we randomly select two-thirds of the events to generate a vespagram, minimizing the influence of events with anomalous amplitudes.

In type II, given that most events are concentrated at a distance range of 29°–32° (Extended Data Table 1 and Supplementary Fig. 7), which could bias the vespagram results, we randomly select half of the events from this distance range, along with events at other epicentral distances, to generate the vespagram.

In type III, we randomly shift each trace within a range of −10 s and 10 s before stacking to mitigate possible uncertainties in the epicentral distance and picked P-arrival time.

For all tests, we repeat the random resampling process 200 times and calculate the mean vespagram along with the corresponding 95% confidence levels.

Contamination from glitches could potentially introduce artefacts[50], even with the application of glitch-removal processes. To evaluate possible effects of glitches, we conduct additional vespagram analysis on the raw data, and the data contain only identified glitch signals (Supplementary Fig. 23). The results indicate that the energy of the core phases identified on the vespagram is not attributable to these glitches.

## Analysis for individual events

Guided by the results from the vespagram analysis, we apply complementary approaches to identify core phases for individual events, including time-domain envelopes (filter bank)[56] and polarized analysis[33].

In the filter bank approach, we first apply a zero-phase-shift bandpass filter to the vertical velocity trace of each event, using a bandwidth half an octave wide around each central frequency, ranging from 1/16 Hz to 2 Hz. Next, we use a time-domain polarization filter to construct a filter bank and compute envelopes in 5-s-long time windows. Given the uncertainty in marsquake locations, to better identify the arrivals, we select the time-domain envelope peaks within a ±15 s time window around the predictions from the vespagram analysis, focusing on a frequency band from 1/4.8 Hz to 1/1.2 Hz, for which clear energy packages are consistently observed.

We further use the FDPA method[7,20,59]. We start by computing the S-transform of three-component waveforms and a 3 × 3 cross-spectral covariance matrix using 90% overlapping time windows, with duration varying inversely with frequency. The relative sizes of the eigenvalues of this covariance matrix indicate the degree of polarization (DOP) of the particle motion, ranging from 0 and 1, whereas the complex-valued components of the eigenvectors describe the particle motion ellipsoid. A polarized wave results in a notably larger eigenvalue and a DOP close to 1. The orientation of the semi-major axis provides information on the inclination, with the back-azimuth derived from the projection of the axis onto the horizontal plane and the deviation from the vertical plane determining the inclination. We then identify seismic arrivals with rectilinear particle motion dominantly polarized in vertical (VRM) and horizontal directions (HRM). Given the nearly vertical incidence of the core phases, a strong peak in the VRM−HRM plot would confirm their presence (Extended Data Fig. 2).

## Travel time picks and measurement uncertainties

For the vespagram results, we estimate the uncertainties in the slowness and travel time of the candidate phases using the bootstrap resampling

test type I (refs. 60,61). In each vespagram from the resampling tests, grids with energy exceeding 85% of the peak are assigned a value of 1, whereas others receive a value of 0. We then sum the values at each grid and calculate their percentage occurrence among all tests (Fig. 2b,e), showing a concentrated distribution at the slowness and travel time of the candidate core phases. To evaluate the effect of the threshold selection, we also test two alternative levels: 50% and 70% (Supplementary Fig. 16). Although lower thresholds yield higher percentage occurrences, they also produce more dispersed energy distributions. Accordingly, we adopt the 85% threshold as an optimal balance between robustness and precision. Focusing on a window around these phases, we fit the distribution separately along the travel time and slowness axes with Gaussian functions (Fig. 2e) to derive mean values for each. The uncertainties are represented by the $1\sigma$ values obtained from these analyses.

In the analysis for individual events, the RMS value of envelope peaks from 1/4.8 Hz to 1/1.2 Hz in the filter bank is used to determine the arrival time of our candidate phase (Extended Data Fig. 2a). The corresponding standard deviation is computed to estimate the travel time uncertainty for subsequent inversions.

## Amplitude and polarity of PKKP and PKiKP

To better capture the relative amplitude between different phases, we first normalize the time window for a reference phase (for example, P or PKiKP) using equation (1). Other phases are then normalized using the same ratio as the reference phase. Both stackings produce similar results, which shows that a stacked amplitude ratio of around 0.5 between PKKP and PKiKP ($A_{PKKP}/A_{PKiKP}$) (Supplementary Fig. 28).

For the polarity, we first adjust the traces to ensure all P waveforms have positive polarity. However, the polarity of PKiKP can vary, either positive or negative, depending on the focal mechanism. Therefore, this approach does not yield focused energy. We then test by setting the polarity of all targeted PKiKP arrivals to positive, and analyse the corresponding vespagram for PKKP. Nevertheless, this adjustment does not markedly improve the results because of possible source complexity and complicated propagation effects (Supplementary Fig. 29).

Source complexity and propagation effects at shallow depths can be partially removed by deconvolving the P (using −2 s to 8 s of the P-arrival time here) or PKiKP (−5 s to 5 s) waveforms. Vespagrams from deconvolved waveforms show more focused energy than those from the original waveforms (Supplementary Fig. 30). Notably, deconvolving PKiKP shows that the vespagram of PKKP has opposite polarity to PKiKP, with an $A_{PKKP}/A_{PKiKP}$ of about 0.5, consistent with the individual event measurement (Extended Data Fig. 3 and Supplementary Fig. 31). However, we exercise caution about whether the selected P or PKiKP waveforms accurately represent the source. Consequently, extracting amplitude and polarity information for marsquakes with low signal-to-noise ratios and unknown source mechanisms remains challenging. Moreover, using envelope data proves to be more effective and stable for vespagram analysis in detecting core phases.

We also measure the $A_{PKKP}/A_{PKiKP}$ for event S0235b, which has concurrent and clear observations of both phases. In the bandpass-filtered data (Extended Data Fig. 7a), the amplitudes of PKiKP and PKKP are determined as the maximum amplitude in the target windows. We estimate the amplitude uncertainties as the maximum amplitude in the noise window, defined as 5–15 s before the main phases. This allows us to calculate the $A_{PKKP}/A_{PKiKP}$, along with its uncertainty, as shown in the blue-shaded region in Extended Data Fig. 7c.

## Seismic inversion

To solve the inverse problem of determining core seismic velocity profile, we use the probabilistic approach[62], with a solution given by

$$\sigma(\mathbf{m}) = kf(\mathbf{m})L(\mathbf{m}), \tag{2}$$

where $k$ is a normalization constant, $f(\mathbf{m})$ is the prior model parameter probability distribution, $L(\mathbf{m})$ is the likelihood function, which is a

measure of the similarity between the data and the predictions from model $\mathbf{m}$, and $\sigma(\mathbf{m})$ is the posterior probability distribution.

Assuming that data noise is uncorrelated and described by a Laplace distribution ($L_1$-norm), the likelihood function takes the form

$$L(\mathbf{m}) \propto \exp(-\varphi), \tag{3}$$

where $\varphi$ is the misfit function. The general expression for the misfit is

$$\varphi = \frac{1}{N}\sum_{j}^{N}\frac{\left|\mathbf{d}_{obs_j} - \mathbf{d}_{cal_j}\right|}{\sigma_j}, \tag{4}$$

where $\mathbf{d}_{obs}$ and $\mathbf{d}_{cal}$ denote the vectors of observed and synthetic travel times relative to the P, with $N$ expressing the total number of travel times, which comprises all possible combinations of phases with respect to P: $\Delta T_{PKiKP}-T_P$, $\Delta T_{PKKP}-T_P$, $\Delta T_{P'P'_{r\_ab}}-T_P$, $\Delta T_{P'P'_n}-T_P$ and $\sigma$ is the travel time uncertainty obtained from either vespagram or individual event analysis.

We use two strategies to define the travel time dataset and the associated $\sigma$. The first strategy, referred to as M_vesp, uses travel time picks of core phases and their uncertainties from the vespagram as $\mathbf{d}_{obs}$ and $\sigma$, respectively (Supplementary Table 1 and Supplementary Fig. 2). In this case, each core phase has only one travel time pick and the corresponding $\mathbf{d}_{cal}$ represents the theoretical travel time at a single epicentral distance of 29°. Alternatively, the second strategy, M_pick, uses travel times relative to P for selected individual events at their epicentre distances as $\mathbf{d}_{obs}$ (Supplementary Tables 3–6).

As the slowness of PKiKP offers additional constraints on IC size, we also incorporate the slowness into the M_vesp by modifying the misfit function $\varphi$ as

$$\varphi = \frac{1}{N}\sum_{j}^{N}\frac{\left|\mathbf{d}_{obs_j} - \mathbf{d}_{cal_j}\right|}{\sigma_j} + \mathbf{w}\frac{\left|\mathbf{p}_{PKiKP\_obs} - \mathbf{p}_{PKiKP\_cal}\right|}{\sigma_p}, \tag{5}$$

where $\mathbf{p}_{PKiKP\_obs}$ and $\sigma_p$ represent the observed slowness and its corresponding uncertainty derived from the vespagram analysis in Fig. 2e, whereas $\mathbf{p}_{PKiKP\_cal}$ is the calculated slowness for a select model; $\mathbf{w}$ denotes the weight balancing the contributions of travel time and slowness in the misfit function. By incorporating the slowness of PKiKP (Extended Data Table 2, case 9), the inverted IC radius is consistent with those obtained by inverting only travel times, but with a more concentrated posterior distribution.

Finally, to sample the posterior distribution (equation (2)), we use the Metropolis sampling algorithm[62]. This algorithm randomly samples the model space, ensuring more frequent sampling of models that align with previous information while simultaneously fitting the data.

## Seismic model parameterization

Given that our target seismic phases are primarily related to the Martian core, a crucial consideration is whether to invert the Martian mantle structure simultaneously. As PKiKP and PKKP travel with nearly identical paths in the mantle (Fig. 1b), their differential travel time are primarily affected by the core structure, with only minimal impact from the one-dimensional mantle structure variations (see Supplementary Information section A4.1 for more details). This supports our decision to focus solely on inverting core velocity structure with $\Delta T_{PKKP-P}$ and $\Delta T_{PKiKP-P}$. By opting for differential travel times over absolute travel times, we also mitigate the effects of the uncertainties of the mantle model. Furthermore, the mantle near the antipode sampled by the P'P' could potentially be heterogeneous[59,63–66]. Therefore, we refrain from amalgamating travel times obtained in previous studies[7,20,31,32,56,59] and our new measurements to invert the entire Martian velocity structure. Instead, we invert the P-wave velocity ($V_P$) structure of the core based on

different mantle models and assess how different models can affect the inverted core models. Seven representative mantle models: the mean, lower and upper bound $V_P$ models from the AK_subset models[7]; the two inverted $V_P$ models incorporating SKS travel times[20]; and the two models with MSL[31,32] are selected to perform inversion separately. These mantle models encompass a range of models produced in previous studies.

The Martian core P-wave velocity structure is then parameterized with five model parameters (Extended Data Fig. 4): the OC radius ($R_{OC}$), the $V_P$ at the CMB ($V_{P\_CMB}$), the $V_P$ of the OC side at the ICB ($V_{P\_OC\_ICB}$), the IC radius ($R_{IC}$), and the $V_P$ jump at ICB ($\delta V_{P\_ICB}$). Here, the model spaces of the OC are based on the results in ref. 20. In the case of the model with an MSL overlying the liquid OC (Extended Data Fig. 4c), the thickness of MSL, the mean velocity gradient in MSL and the $V_P$ jump at the bottom of MSL (the real CMB) are set according to the two models, MSL_ETH[31] and MSL_IPGP[32]. To simplify the travel time calculation using TauP[67], we treat the MSL as a part of the liquid OC. Thus, our identified PKKP corresponds to PKKP$_{MSL}$ in this case and the inverted $R_{OC}$ contains the thickness of MSL.

We further assume that the mean velocity gradients in the OC and IC are the same. The core velocity at a certain depth $r$ is interpreted with a smooth function as

$$V_P = \cosh^{-1}\left(\frac{r-b}{K}\right), \tag{6}$$

$$K = \frac{r_2 - r_1}{\cosh(V_{P_2}) - \cosh(V_{P_1})}, \quad b = r_1 - K \cdot \cosh(V_{P_1}). \tag{7}$$

Here, $r_1$ and $r_2$ are the depths of the CMB and ICB, respectively. $V_{P_1}$ and $V_{P_2}$ denote the corresponding velocities at these interfaces.

The inverted core velocity profiles, based on different mantle models and strategies, are listed in Extended Data Table 2.

### Composition of the Martian core

The presence of H substantially lowers the melting point of Fe, making it difficult to form a solid IC[41]. Therefore, in our density and velocity models, we considered only O, C and S as light elements in the Martian core. We examined two distinct core composition models. In the first model, when the C content exceeds 4 wt%, a C-rich IC may precipitate[42,43]. Under this scenario, the O and S contents range from 0 to 6 wt% and 12 to 16 wt%, respectively[21]. The second model assumes an O-enriched core. A solid O-rich IC will precipitate when the oxygen content exceeds about 6.7 wt% (ref. 21). Based on the seismologically inferred radius of the IC, we estimate the O content in the Martian core to be 6.7–9.0 wt%, corresponding to C and S concentrations of approximately 0–4 wt% and 12–16 wt%, respectively[21]. Both models account for the partitioning of O and C between the Martian OC and IC. The density and velocity profiles of solid Fe$_3$C and FeO under high pressure and temperature conditions are as follows[68,69]:

$$V_{Fe_3C\text{-solid}} = 1.09 \times \rho_{Fe_3C} - 1.79, \tag{8}$$

$$V_{FeO\text{-solid}} = [1.55 + 4.3 \times 10^{-5} \times (T - 300)] \times \rho_{FeO} - 2.03 \\ - 5.6 \times 10^{-4} \times (T - 300). \tag{9}$$

The influence of O, C and S on the OC velocity and density are calculated as follows[70]:

$$\rho_{Fe-S-O-C-liquid} = 8.63 - 2.71 \times \text{Cont.}_C - 4.36 \times \text{Cont.}_O \\ - 5.24 \times \text{Cont.}_S, \tag{10}$$

$$V_{Fe-S-O-C-liquid} = 5.76 + 2.88 \times \text{Cont.}_C - 0.002 \times \text{Cont.}_O \\ - 1.24 \times \text{Cont.}_S, \tag{11}$$

where Cont.$_O$, Cont.$_C$ and Cont.$_S$ are mole per cent of O, C and S used in our modelling.

## Data availability

The marsquake event catalogue V14 (ref. 29) is available at the Incorporated Research Institutions for Seismology (IRIS) through the Data Management Center (DMC). The InSight seismic data presented here are available from the IRIS-DMC, NASA PDS and IPGP Data Center Services[71]. The AK_subset interior models used in this study are available from MSDS[72].

## Code availability

The codes for vespagram analysis and core velocity structure inversion are available at GitHub (https://github.com/HX-B/Mars_IC). Figures were created using matplotlib[73], seismic data processing was done in ObsPy[74], and inversions were done in NumPy and SciPy (refs. 75,76).

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

**Acknowledgements** We thank D. J. Stevenson for his insights and discussions, which have improved this study. We thank N. Schmerr, M. Panning and two anonymous reviewers for their comments, which have improved the manuscript from both methodological and scientific perspectives. We thank Q. Huang for providing the SKS_GD and SKS_GP inversion models, and X. Song, B. Chao and H.-Y. Yang for comments on the paper. We acknowledge the National Aeronautics and Space Administration (NASA), Centre National d'Études Spatiales (CNES), their partner agencies and institutions (UK Space Agency (UKSA), Swiss Space Office (SSO), Deutsches Zentrum für Luft- und Raumfahrt (DLR), Jet Propulsion Laboratory (JPL), Institut de

Physique du Globe de Paris–Centre National de la Recherche Scientifique (IPGP-CNRS), Eidgenössische Technische Hochschule Zürich (ETHZ), Imperial College London and Max Planck Institute for Solar System Research–Max-Planck-Gesellschaft (MPS-MPG)) and the flight operations team at JPL, SEIS on Mars Operations Center (SISMOC), Mars SEIS Data Service (MSDS), Incorporated Research Institutions for Seismology Data Management Center (IRISDMC) and Planetary Data System (PDS) for providing SEED SEIS data. We appreciate the Supercomputing Center of the University of Science and Technology of China (USTC) for high-performance computing services. This research was supported by the B-type Strategic Priority Program of the Chinese Academy of Sciences, grant XDB41000000; the National Natural Science Foundation of China 42241117, 42394113 and 41722401.

**Author contributions** All authors contributed to the Article. D.S. designed the project. H.B. and D.S. performed the seismic studies. N.S., Z.M. and D.S. carried out mineral physics interpretations. D.H. and M.D. provided the discussion. D.S. and H.B. wrote the original draft. All authors discussed the results and commented on the paper.

**Competing interests** The authors declare no competing interests.

**Additional information**
**Correspondence and requests for materials** should be addressed to Daoyuan Sun.

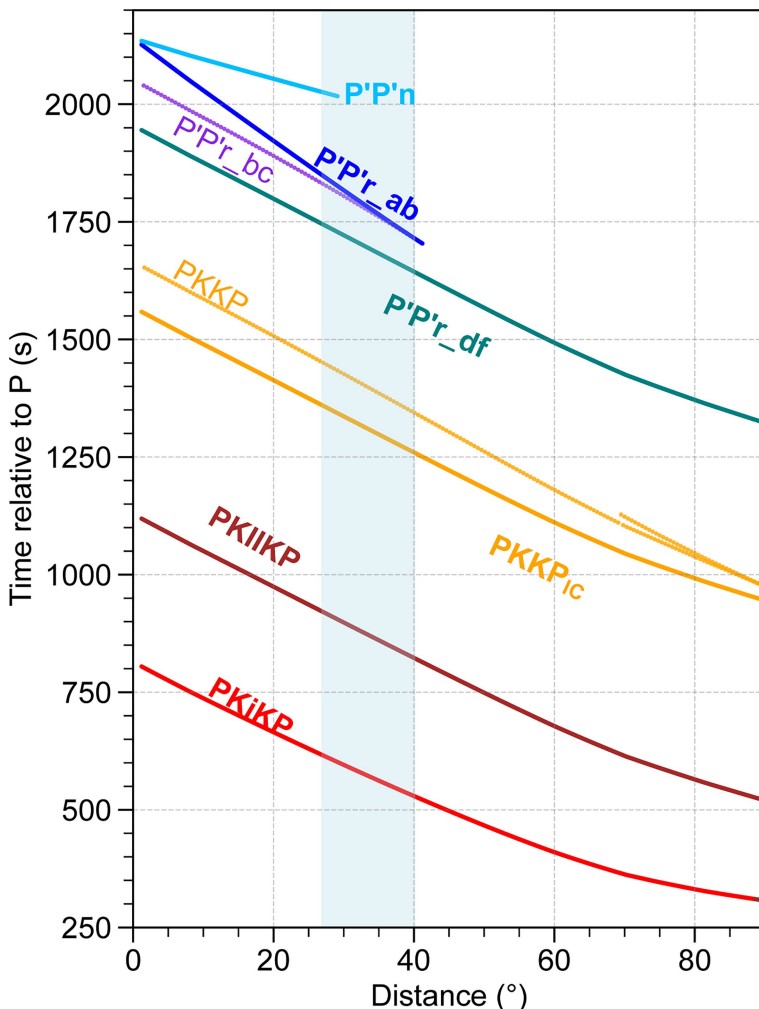

**Extended Data Fig. 1 | Travel-time curves for different Martian core phases.** Light and heavy lines represent travel times for models without and with IC, respectively. The travel times are calculated for a marsquake at a depth of 33 km, using the best "geodynamical" inversion model (SKS_GD model) from ref. 20 as the mode without IC and assuming a + 30% velocity jump across the ICB for the IC model with an IC radius of 618 km.

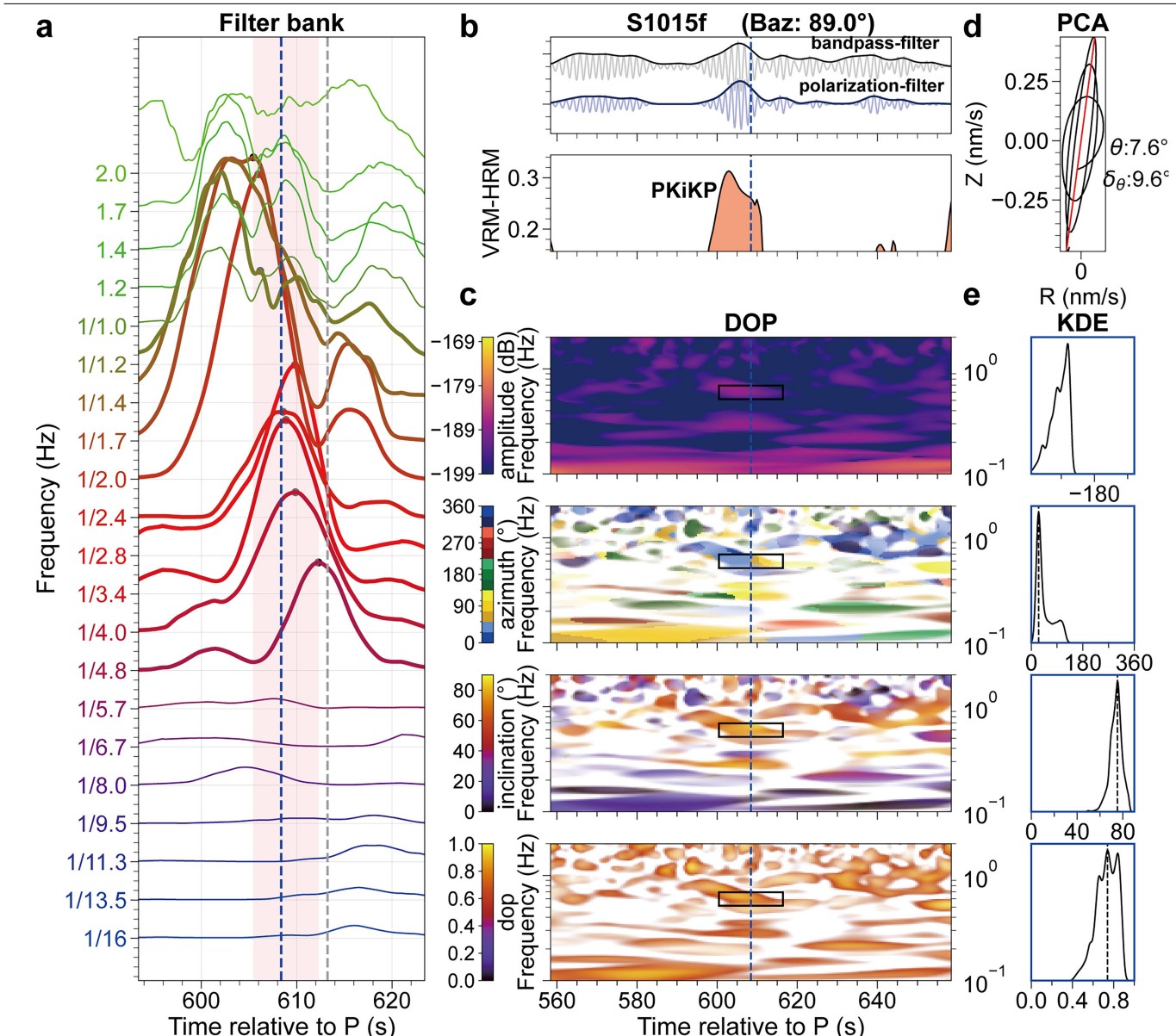

**Extended Data Fig. 2 | Identification of the PKiKP phase on event S1015f.**
**a**, Filter bank analysis. The coloured waveforms represent filtered data with
narrow bandpass filters with different centre frequencies, as indicated by the
coloured text. The grey dashed line denotes the travel time computed by
vespagram analysis and the blue dashed line denotes the root-mean-square
value of time-domain envelope peaks (black circles) in a frequency range from
1/4.8 Hz to 1/1.2 Hz (heavy coloured lines). The pink shaded region outlines the
time distribution of the peaks. **b**, Waveform and FDPA analysis. The upper panel
shows the vertical component waveforms and corresponding envelopes of
the bandpass- (grey line) and polarization- (blue line) filtered traces within a
frequency band indicated by the black rectangles circled in **c**. The back-azimuth
is provided by MQS[29]. The lower panel is the vertical-horizontal summed FDPA
intensity as a function of time. **c**, Polarization analysis. From top to bottom,
spectrum, azimuth, inclination angle of the major axis of particle motion, and
ellipticity (DOP) obtained from polarization analysis. We reject all parts of the
signal with a DOP < 0.6 by setting the corresponding part of the S-transformed
data to zero, which allowed us to suppress some weakly polarized signals.
The inclination angle represents the angle deviating from the horizontal plane.
The black dashed lines denote the travel-time picks obtained from filter bank
analysis. The black rectangles outline the travel-time pick uncertainties of ± s in
the frequency range 0.5–0.7 Hz. The blue dashed lines in **b**,**c** denote the travel-
time picked from filter bank analysis in **a**. **d**, The particle motions on the vertical
and radial components in a time window of ±2.5 s based on the travel time
picked from filter bank analysis. The red line denotes the fitted line and its
incident angle deviating from the vertical plane ($\theta$) and error ($\delta_\theta$) are annotated
below. **e**, Kernel density estimation. The black dashed lines denote the maximum
values of probability density computed by marginalizing over the frequency
and time axes as the black rectangle shown in **c**.

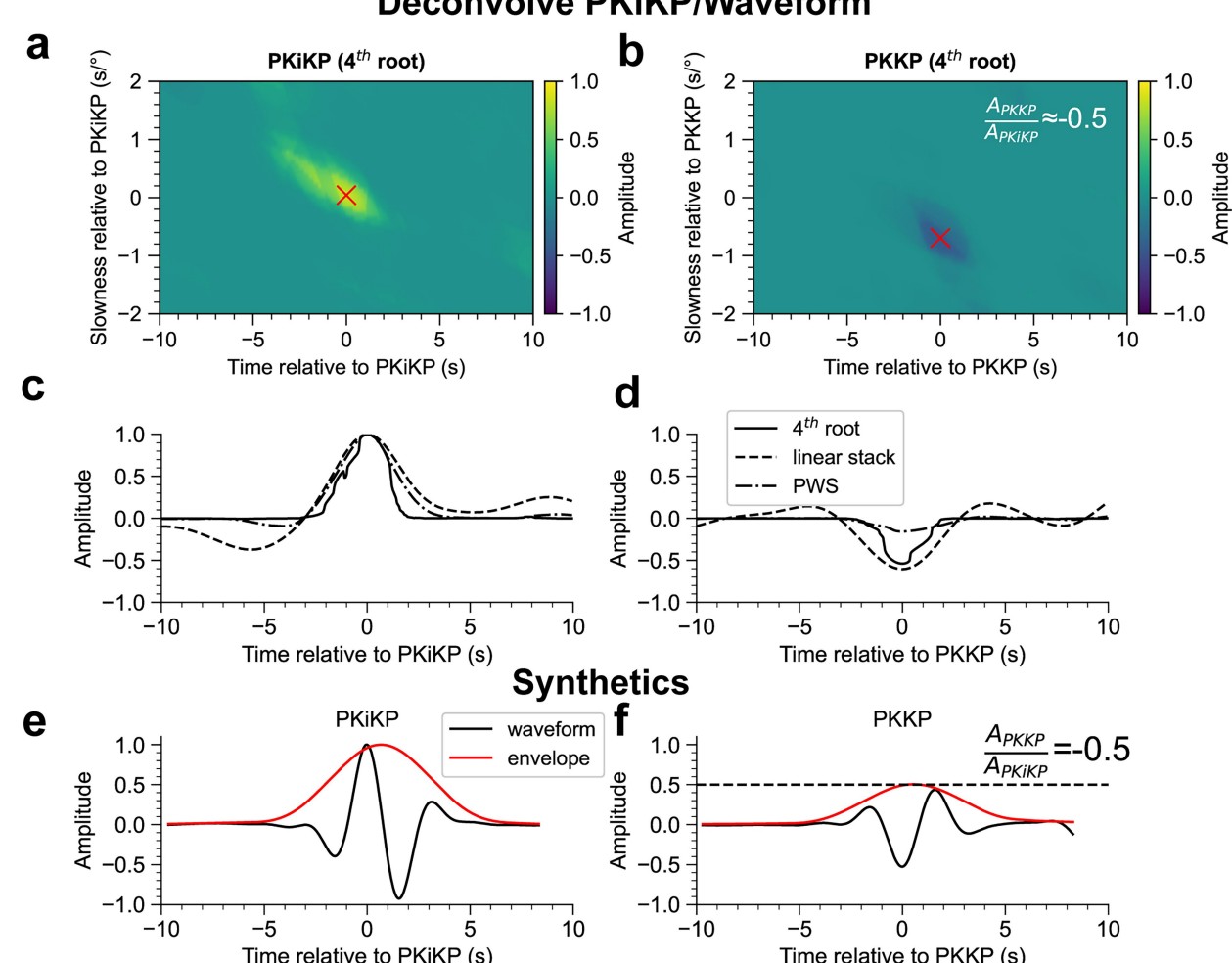

**Extended Data Fig. 3 | Amplitude and polarity of PKKP and PKiKP.**
**a**,**b**, Vespagram analysis of (**a**) PKiKP and (**b**) PKKP using 4th-root stacking method, following deconvolving the PKiKP waveform. The red crosses mark the energy peak. In **b**, $A_{PKKP}/A_{PKiKP}$ is ~−0.5, suggesting their reversed polarities. **c**,**d**, Stacking waveforms of (**c**) PKiKP and (**d**) PKKP using linear, 4th-noot, and PWS stacking methods, with the slowness marked in the vespgrams in **a** and **b**, respectively. **e**,**f**, Synthetic waveforms and envelopes of (**e**) PKiKP and (**f**) PKKP, with both traces normalized to the absolute amplitude of PKiKP.

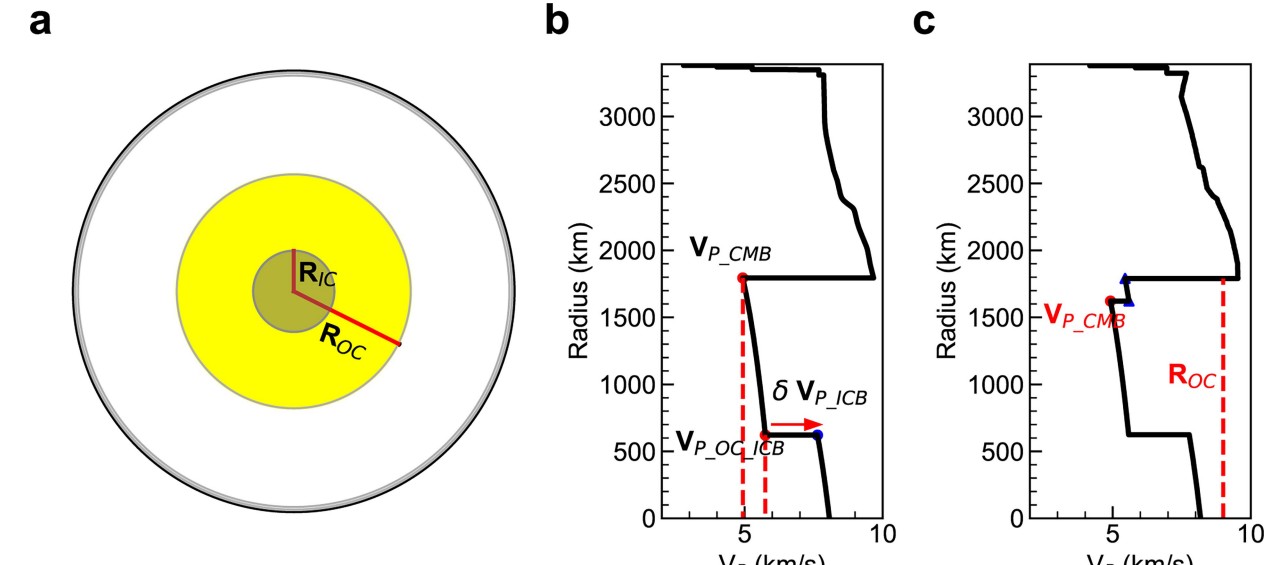

**Extended Data Fig. 4 | Inverted parameters used to define the Martian core velocity model. a**, The outer core radius ($R_{OC}$) and inner core radius ($R_{IC}$). **b**, The core $V_P$ at the CMB ($V_{P\_CMB}$), the $V_P$ of the outer-core side at the ICB ($V_{P\_OC\_ICB}$), and the $V_P$ jump at ICB ($\delta V_{P\_ICB}$). **c**, Definition of $V_{P\_CMB}$ and $R_{OC}$ in the case of a MSL overlying the liquid OC. Here, the $R_{OC}$ is the sum of the real core radius and the thickness of MSL. The core velocity at a certain depth is interpreted with a smooth function as defined in Methods.

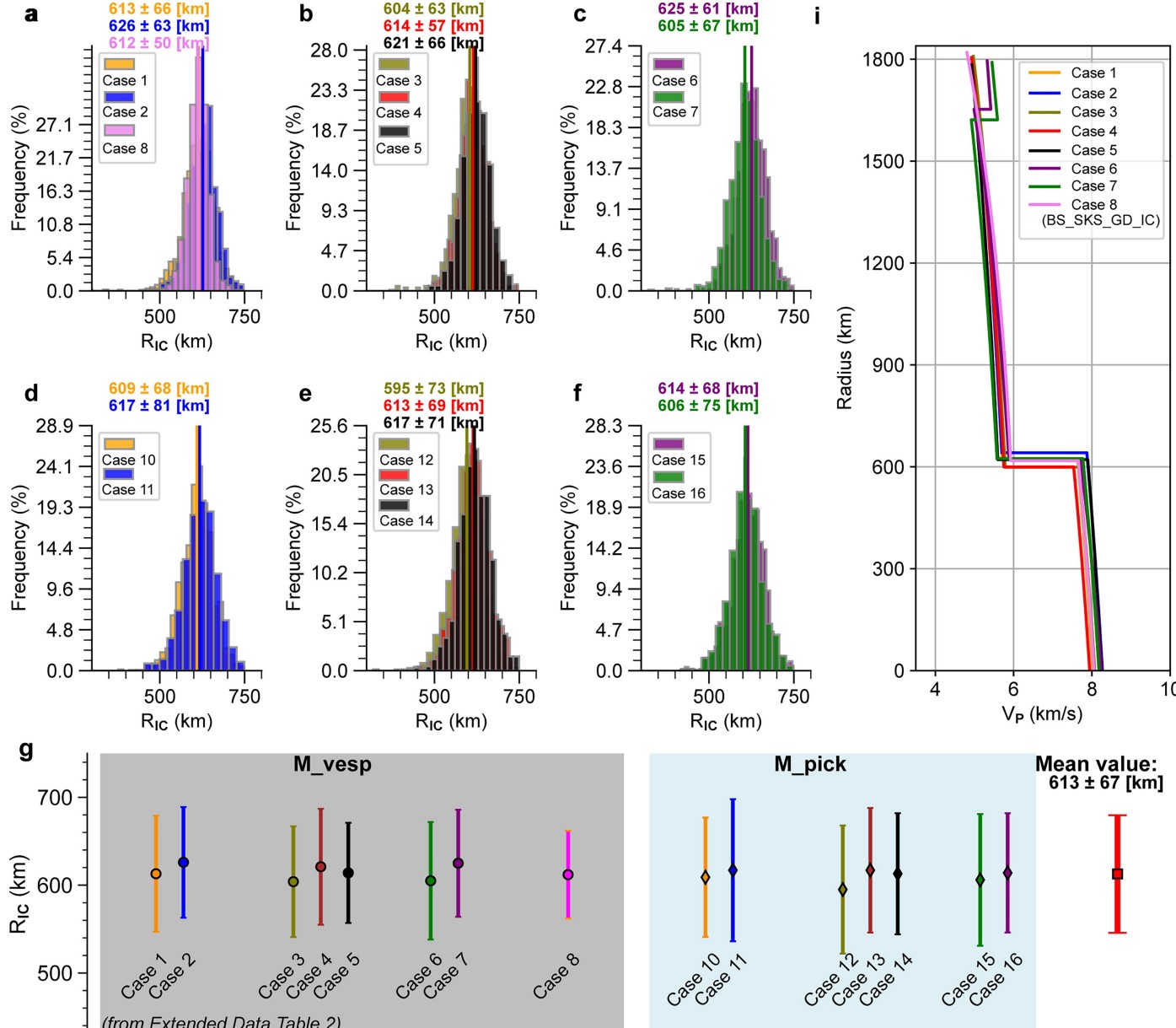

**Extended Data Fig. 5 | Inverted results on IC size for different mantle models and inversion strategies.** In **a**–**f**, for each case listed in Extended Data Table 2, the mean value is depicted by the solid line of the corresponding colour. Mean value and 85% confidence interval are labelled in coloured text at the top. **g**, Mean inverted IC sizes and their corresponding 85% confidence intervals for different mantle models and inversion strategies as listed in Extended Data Table 2. **i**, Best inverted Martian core models based on different mantle models, selected for minimal misfit from those within the ranges of $V_{P,CMB}$ (4.9–5.0 km/s) and $R_{OC}$ (1,780–1,810 km), in agreement with ref. 20. The BS_SKS_GD_IC model is inverted by fixing both the mantle and outer-core structure as the SKS_GD model (case 8 in Extended Data Table 2).

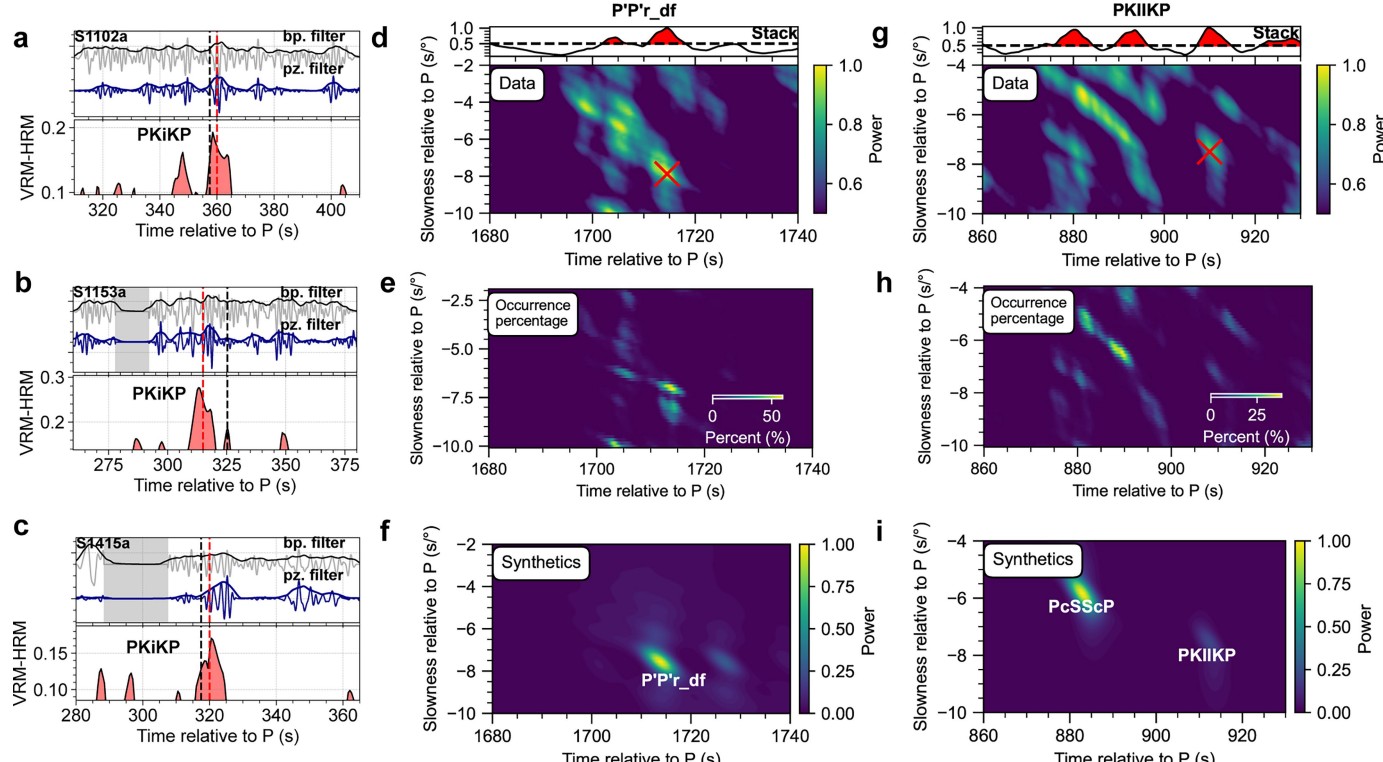

**Extended Data Fig. 6 | Identification of other IC related phases.**
**a**–**c**, Polarization analysis for identification of PKiKP in events S1102a (73.3°), S1153a (84.8°), and S1415a (88.2°) (Extended Data Table 1). The detailed legend follows that of Fig. 2g,h. The black and red dashed lines denote the predicted travel time from BS_SKS_GD_IC model and the measurement by FDPA, respectively. Grey shaded regions indicate the presence of known instrument glitches. **d**, Vespagram of P′P′r_df using all 23 events, same as Fig. 2a. The red cross denotes the candidate phase. **e**, Occurrence percentage of grids with high energy in the vespagram for P′P′r_df, same as Fig. 2b. **f**, Synthetic vespagram of P′P′r_df, based on the inverted BS_SKS_GD_IC model. **g**–**i**, Same as in **d**–**f**, but for PKIIKP and PcSScP.

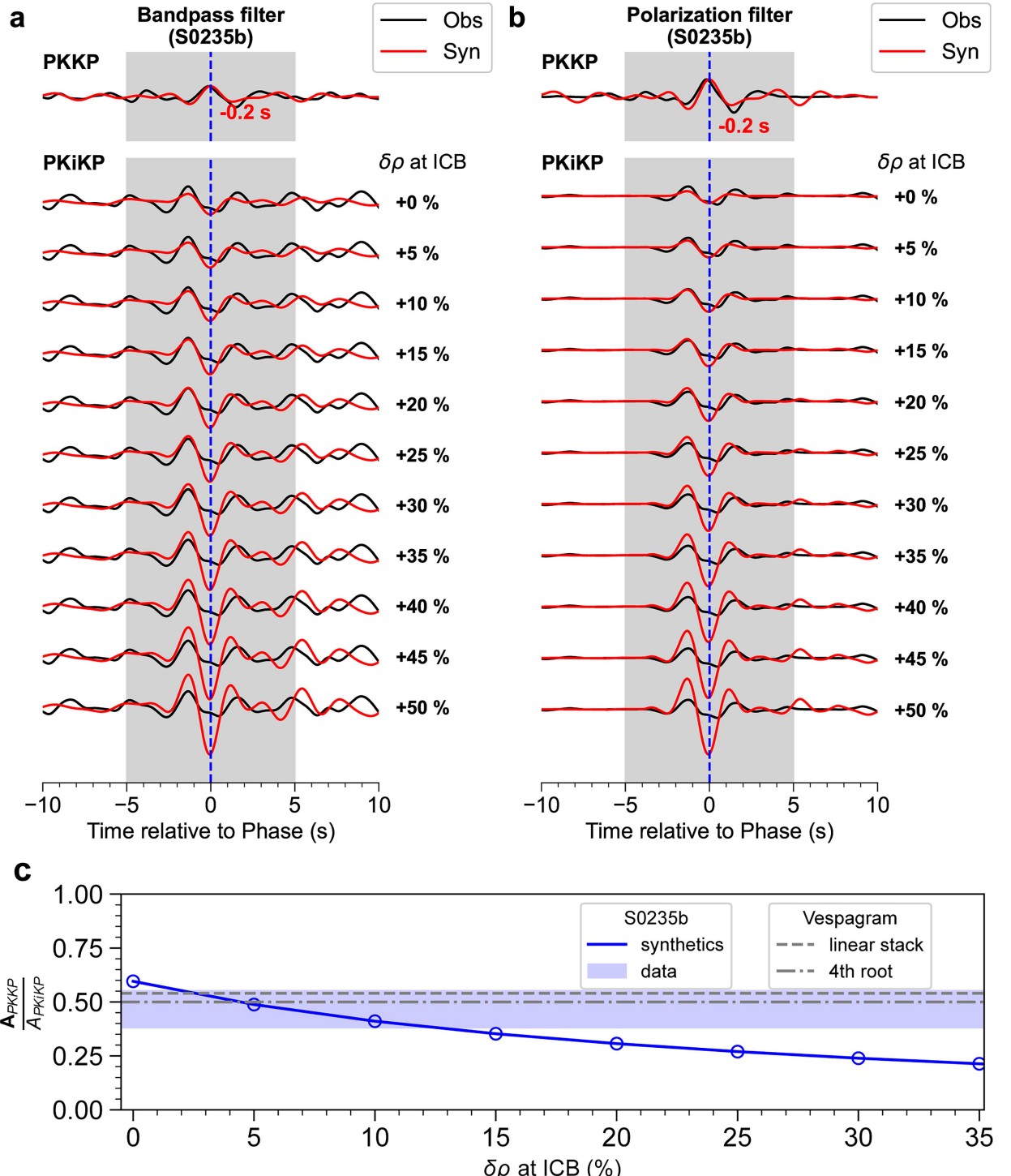

**Extended Data Fig. 7 | Amplitude ratio between PKiKP and PKKP.**
**a**, Bandpass and **b**, polarization filtered data in a frequency range of 0.2–0.8 Hz, with all data traces are aligned on PKKP or PKiKP arrival times. Both data (black) and synthetics (red) are normalized on the amplitude of PKKP. Our inverted model BS_SKS_GD_IC, with +30% P-velocity jump across the ICB exhibits an excellent fit to the travel times, requiring −0.2 s shift of PKKP in the synthetics to match the waveform of PKKP and PKiKP simultaneously. The density jump ($\delta\rho$) at the ICB used for generating synthetics is indicated by the number following each trace. **c**, Amplitude ratio of PKiKP and PKKP ($A_{PKKP}/A_{PKiKP}$). The blue open circles denote the synthetics $A_{PKKP}/A_{PKiKP}$ assuming different $\delta\rho$. The blue shaded region outlines the measured $A_{PKKP}/A_{PKiKP}$ for the bandpass filtered data with its uncertainty (Methods). The dashed and dashed-dotted lines represent the measured $A_{PKKP}/A_{PKiKP}$ obtained from vespagram analysis (Extended Data Fig. 3b) using linear and 4$^{th}$-root stacking, respectively. A density jump at the ICB of 7 ± 5% is preferred to explain the observed $A_{PKKP}/A_{PKiKP}$.

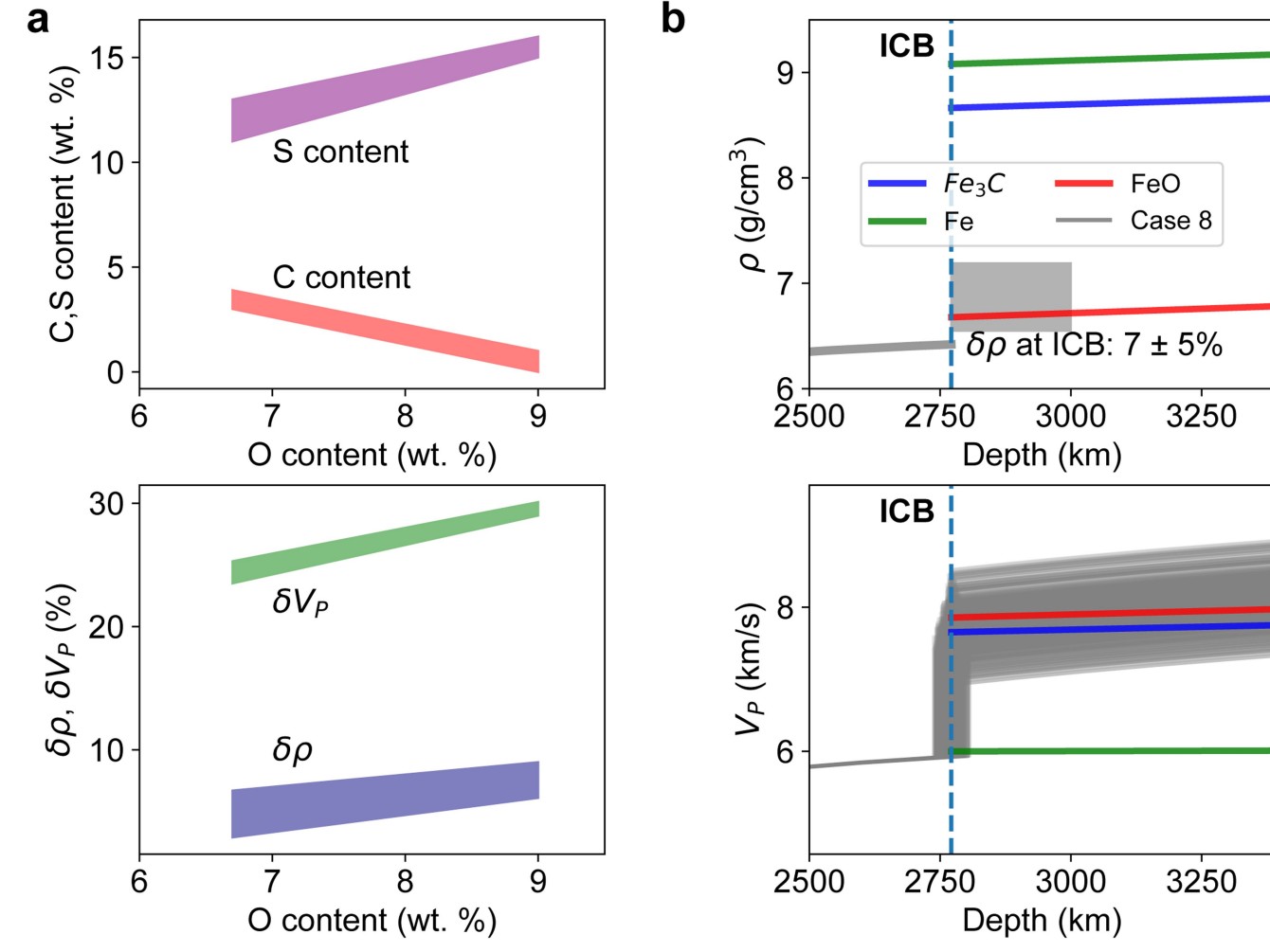

**Extended Data Fig. 8 | Influence of light elements on the density and velocity jumps across the ICB assuming an O-enriched Martian core.** **a**, From top to bottom: amount of O, C, and S in the Martian core used in our modelling; influence of O content on the density and velocity jumps across the inner and outer core boundary. The amount of O, C, and S and their partitioning between outer and inner core are from ref. 21. **b**, Calculated density and velocity of Fe-light element alloys in the Martian IC. Grey lines: seismic observations in this study (Case 8); green: Fe; blue: Fe$_3$C; red: FeO.

**Extended Data Table 1 | List of marsquakes used in this study**

| Index | Event ID | Quality | Origin time | Distance (°) | Baz (°) |
|---|---|---|---|---|---|
| 1 | S1015f | A | 2021-10-04T04:55:55.150429Z | 27.5 ± 2.0 | 89.0 |
| 2 | S0918a | B | 2021-06-27T05:35:04.667245Z | 27.8 ± 6.7 | 136.8 |
| 3 | S0235b | A | 2019-07-26T12:19:16.991001Z | 28.7 ± 1.5 | 77.0 |
| 4 | S0864a | A | 2021-05-02T01:01:05.534422Z | 28.7 ± 3.5 | 90.0 |
| 5 | S0474a | B | 2020-03-28T00:34:17.978182Z | 29.1 ± 17.5 | 96.9 |
| 6 | S0916d | B | 2021-06-25T05:17:12.455430Z | 29.3 ± 5.9 | 96.9 |
| 7 | S0802a | B | 2021-02-28T06:11:05.701380Z | 30.0 ± 3.5 | 82.0 |
| 8 | S0173a | A | 2019-05-23T02:22:58.043698Z | 30.0 ± 1.4 | 88.0 |
| 9 | S0820a | A | 2021-03-18T14:54:19.835741Z | 30.2 ± 2.4 | 105.9 |
| 10 | S1048d | A | 2021-11-07T22:04:04.408860Z | 30.2 ± 1.3 | 96.9 |
| 11 | S1133c | A | 2022-02-03T08:08:25.290742Z | 30.2 ± 1.3 | 91.0 |
| 12 | S0290b | B | 2019-09-21T03:19:05.310837Z | 30.4 ± 8.2 | — |
| 13 | S1039b | B | 2021-10-29T16:15:56.296678Z | 30.4 ± 17.5 | — |
| 14 | S1022a | A | 2021-10-11T23:18:20.004721Z | 30.7 ± 2.0 | 63.1 |
| 15 | S0484b | B | 2020-04-07T08:52:05.561869Z | 31.8 ± 5.9 | 99.9 |
| 16 | S1197a | B | 2022-04-09T22:09:33.753550Z | 32.0 ± 1.5 | — |
| 17 | S0105a | B | 2019-03-14T21:03:28.123118Z | 32.5 ± 8.2 | 111.9 |
| 18 | S0189a | B | 2019-06-09T05:40:05.486385Z | 33.4 ± 8.2 | — |
| 19 | S0784a | B | 2021-02-09T12:15:46.472278Z | 34.5 ± 3.6 | 114.9 |
| 20 | S1157a | B | 2022-02-27T01:07:02.379242Z | 35.6 ± 2.5 | — |
| 21 | S1222a | A | 2022-05-04T23:27:45.331451Z | 37.0 ± 1.6 | 100.9 |
| 22 | S1012d | B | 2021-10-02T00:34:03.626131Z | 38.3 ± 3.3 | — |
| 23 | S0325a | B | 2019-10-26T06:58:55.171919Z | 39.7 ± 6.1 | 57.2 |
| 24 | S1102a | A | 2022-01-02T04:35:28.844688Z | 73.3 ± 4.6 | 285.9 |
| 25 | S1153a | B | 2022-02-23T21:09:33.746256Z | 84.8 ± 10.5 | — |
| 26 | S1415a | B | 2022-11-19T21:55:49.119346Z | 88.2 ± 9.6 | — |

All information of these LF events comes from V14 catalog[29]. The events within the green shaded region, with small distances, are used to perform vespagram analysis. The events in blue shaded regions are used for validation purposes.

**Extended Data Table 2 | Summary of inverted Martian core models**

| Case # | Inversion strategy | Include slowness | Base models | Fix outer core | $R_{IC}$ (km) | $\delta V_{P\_CMB}$ (%) | $R_{OC}$ (km) |
|---|---|---|---|---|---|---|---|
| 1 | M_vesp | – | SKS_GD | – | 613±66 | 35.3±16.6 | 1799±66 |
| 2 | | – | SKS_GP | – | 626±63 | 35.0±17.0 | 1792±61 |
| 3 | | – | AK_min | – | 604±63 | 33.9±17.3 | 1787±61 |
| 4 | | – | AK_mean | – | 614±57 | 37.6±16.0 | 1812±68 |
| 5 | | – | AK_max | – | 621±66 | 35.6±15.5 | 1816±67 |
| 6 | | – | MSL_ETH | – | 625±61 | 30.7±17.1 | 1626±54* |
| 7 | | – | MSL_IPGP | – | 605±67 | 36.1±17.0 | 1641±68* |
| 8 | | – | SKS_GD | yes | 612±50 | 32.0±7.5 | – |
| 9 | | w=100 | SKS_GD | – | 631±30 | 35.9±15.1 | 1790±67 |
| | | | | | | | |
| 10 | M_pick | – | SKS_GD | | 609±68 | 35.6±13.8 | 1789±56 |
| 11 | | – | SKS_GP | – | 617±81 | 37.4±13.5 | 1790±59 |
| 12 | | – | AK_min | – | 595±73 | 37.0±13.8 | 1786±60 |
| 13 | | – | AK_mean | – | 613±69 | 39.6±13.9 | 1802±63 |
| 14 | | – | AK_max | – | 617±71 | 42.4±12.9 | 1828±57 |
| 15 | | – | MSL_ETH | – | 614±68 | 32.6±14.5 | 1621±48* |
| 16 | | – | MSL_IPGP | – | 606±75 | 38.3±14.5 | 1634±61* |

* Note that the outer-core radius ($R_{OC}$) here is smaller than those inverted from models without MSL, based on the assumption that the inverted PKKP is actually PKKP$_{MSL}$ (Supplementary Figs. 12–13).