## [Peer Review File · Nature]

Seismic Detection of a 600-km Solid Inner Core in Mars

Corresponding Author: Professor Daoyuan Sun

Version 0:

Reviewer comments:

Referee #1

(Remarks to the Author)

Review of "Seismic Detection of a 600-km Solid Inner Core on Mars"

This paper gives seismic evidence provided by the InSight mission's seismic experiment SEIS for the presence of a solid inner core in Mars. The authors used a source array analysis of 23 marsquakes to search for core interacting phases and found candidate arrivals that match predictions for PKKP and another arrival that the authors interpret as PKiKP, a reflection from a postulated inner core. The authors then inverted the arrival times of these phases to determine that the PKiKP phase indicates an inner core of ~610 km, with a P-wave contrast of ~30% into the inner core and used waveform modeling to infer a density contrast of ~10%. The authors then interpret their findings using mineral physics to infer that the inner core must contain light elements and be relatively low temperature; and state that their findings have implication for the generation of dynamo in the martian past and understanding of the evolution of terrestrial planets in general.

The detection of an inner core inside Mars is an important and somewhat puzzling finding; the martian core has generally been considered too small and enriched in light elements to support formation of an inner core, and thus a robust detection of such a structure would be an important contribution to understanding the evolution of smaller bodies in the Solar System. That said, Nature seems like an excellent fit for presenting the authors' work and getting the conversation started in the literature about this topic from an observational standpoint. A similar finding for the Moon by Weber et al., 2011 in Science, required a core that is much higher in sulfur (or other light elements) content than previously thought, and it seems that a similar implication would exist for Mars. The presence of a lunar inner core is still controversial. Presumably such inner core formation for Mars would be recent, so it is perplexing that the authors chose to discuss implications for the past dynamo, but it seems like that the compositions required to get to an inner core may have a more important role in that. However, all of this is predicated on the robustness of the seismology observations (the key component of this paper), which is essential for determining the presence, size, and properties of the inner core (if it exists). The InSight data are challenging in a variety of ways (a vast understatement perhaps), but at the same time are powerful constraints on Mars; my familiarity is primarily in the realm of seismology, so I will focus my comments and criticisms there.

I want to say from the forefront, that the seismological analysis here is thorough and carefully follows many of the established approaches used by the InSight team itself. I must lament that most of that care is buried in the supplement to the paper, but that is the challenge of shorter papers. The authors have done a detailed job of using multiple working lines of evidence for their phase detection analysis (vespagrams created with source arrays, filter banks to identify phases on single seismograms), which closely mirror some of the same approaches used by the InSight team itself to establish the phases detected by the mission (like ScS, and SKS, mantle triplications and other seismic phases) that are now primary constraints on the deep martian interior. Martian seismology is notably tough, so congratulations to the authors for doing such a thorough and careful job! They present several lines of evidence for the presence of an inner core: 1) An early detection of PKKP (on vespagrams and single seismograms with polarization filtering), which interestingly enough seems to also have evidence for the somewhat controversial basal magma layer postcursor and is early enough (~100 seconds) that it makes sense for a high velocity thing (like an inner core) to be in the way. 2) Detection of a PKiKP phase both in vespagrams and in single seismograms at a depth that would put the inner core around ~600 km in radius, 3) the identification of other core interacting phases and several mantle phases to illustrate the robustness of their approach. I've outlined a few things I'd like to see to better shore up those detections, but to my eye their detections seems nearly as robust as any of the other InSight data papers published in the literature about body waves detected in Mars and should certainly be entered into the scientific record (and hopefully follow on studies can refute or verify). My main seismology concern is that the answers do not seem to fully incorporate some of the uncertainties on the measured arrival times and amplitudes, and that the vespagram evidence

presented in the main paper and supplements sometimes seems to have discrepancies that need some explanation. In terms of the seismological modeling and inversion, these approaches to my eye are robust and use standard seismological methodologies (Bayesian approaches, waveform matching, forward wave propagation modeling); but again, I'd like to see a more complete picture of the uncertainties since those flow readily into the geodynamical and mineral physical models that follow. I will let colleagues more fluent in the mineral physics and nature of cores comment on the implied compositional models, but there are a few things that seem rather odd, like an oxygen enriched inner core, which even to my novice eye is seemingly not compatible with how elements partition into solidifying Fe that would form the inner core. Also, the authors do not address the thermal implications of an inner core (lower core temperature) and how that matches with the thermal implications of a basal magma layer (temperatures high enough to sustain a silicate melt layer). Regardless of the interpretation for the mechanism generating an inner core, it seems that the seismological implications for the martian core and its evolution will be a big contributor to those future discussions.

Detailed Comments

Title: "Core on Mars" – shouldn't it be core "in" Mars? I'd recommend that change for clarity.

Line 31: The Earth's inner core is postulated to be relatively young (a few billion years), and may not have a huge role in the history of the magnetic field; thus this opening statement is a bit puzzling. Are the authors suggesting the inner core has been around since the Noachian? If so, what is the evidence for that? For Mars, it appears that the dynamo died very early in the evolution of the planet, so it's not entirely clear how this statement holds in terms of shaping the magnetic evolution of the planet. Perhaps there are clues in the presence of an inner core on the composition and temperature of the early martian core, but I'd suspect that any presence of an inner core would have little effect.

Line 35: There is seismological evidence for a lunar inner core (Weber et al., 2011) from the Apollo seismic experiments, which consisted of 4 seismic stations (over Insight's singular station). How is that detection any less decisive than the identification made in this paper? The manuscript actually says this exact thing later on Line 148. Or are you distinguishing terrestrial "planets" from "moons" – if so, I'd argue that's not a very meaningful distinction.

Line 39: "absence of chemical convection associated with inner core growth" – I'd think that the terrestrial dynamo existed well before the presence of an inner core on Earth (the oldest rocks tell us these things), so this statement seems strange. Dynamos need a conductive fluid, planetary rotation, and a source of convective motions; on Earth this could be a combination of things, but I'd suspect that the nature of heat exchange over the CMB plays a bigger role than the inner core, and would be a feature present for nearly all of Earth (and martian) history.

Line 40: "snowing down" this is a pretty speculative mechanism, and I don't know if we can sort this out on the Earth where geophysical data is abundant, much less so Mars, where we have a very limited dataset from seismology and much less so from other data types. The real finding here is that Mars could have an inner core, do we have to have a model that explains the observation for it to be an observation?

Line 57: "potentials" should be potential.

Line 63, Fig. 1: I'm not sure the elevation data here are particularly helpful, the color palette makes it hard to see the bounce points and some of the lines are over the labels for events. A few geographic labels are really all that is needed to establish the locations of seismic information, so I'd recommend simplifying panel c by removing the topography.

Line 98 (and Line 103): "Striking" and "closely aligning" and "strong evidence" are all qualitative and imprecise descriptions for the observations you show here. The observation of P'P'n in Fig. 2b falls outside the prediction window, and PKKP is over 100 seconds from the prediction (more on that later). It should not come as any surprise that the "reference" models would be imprecise where the martian core is concerned (we have limited observations), but is that error commensurate with the uncertainties? To address this, I would recommend that you quantify your results, the arrivals shown in Fig. 2a-b come between the predictions, for example, in 2a (P'P'r_ab) falls at a slowness of 9.1 s/deg (+/- some error), and arrival time of 1830 s (+/- some error); this falls within the +/- 10 seconds from the prediction, and just below the predicted slowness of -9.5 to -10.5 s/deg (and predictions here are qualitative too, given the uncertainties in the deep martian models).

Line 139: Why not show where this prediction would fall for the two models proposed on Fig. 2d? I think that would help to strengthen this point.

Line 141, Figure 2: I agree that the early arrivals of PKKP and its postcursor are commensurate with a faster object in between and that this is a good line of evidence for a solid inner core (or at least a faster lower core). But how much faster? You do not quantify this (yet). Furthermore, I am a bit puzzled by the difference between the results in Fig. 2d and those in Fig. S13; the vespagrams are not identical, and the earlier arrival near 1290s seems to be lower amplitude in Fig. S13 than in Fig. 2d. What is the source of this discrepancy (see note about vespagrams below)?

Line 155 (and Line 208): If PKKP is indeed sampling the inner core and getting sped up, technically, wouldn't this phase then be called PKIKKIKP? Wouldn't you technically predict a PKIKP as well (a reflection from the other side of the core), on top of P'P'r_df? Did you search for this phase after PKIKP? It seems like that phase should exist if PKKP does (and P'P'r_df), unless the inner core is strongly attenuating for P waves (in this case you wouldn't predict P'P'r_df to show up!), or the geometry is not particularly good for producing the reflection (e.g., it would be at a different distance). But it seems like it should be there if these other phases are showing up as well. Another line of evidence that could further support (or refute)

your interpretation?

Line 166: Why do you report a different error (± 90 km) here than in the abstract (± 50 km)? For this number, I am a bit skeptical of even ± 90 given the uncertainty in the martian reference models (e.g., the structure of the crust and upper mantle, plus the uncertainty in core properties and lowermost martian mantle) that you can get anything this precise. I agree that the paths through the mantle are similar for PKKP and PKiKP, but the CMB on Mars seems like it is relatively complicated (i.e., can really slow down a wave at something like the basal magma layer). Regardless if you support that model or not, it seems reasonable that CMB structure would be highly variable and could map onto the PKKP and PKiKP differential times (much as it does on Earth). What happens in the inversion if you increase the uncertainties of the 1-D reference models?

Line 175; Fig 3. Why only use SKS_GD? And what happens when you remove the 3 events at greater distance?

Line 192: I'd report your error bars on these values, which seems to be ± 100 km for the R_{oc} , and ± 0.5 km/s for the $V_{p_{oc}}$?

Line 199: This seems like it's on the high end of predicted solid core velocities from Antonangeli et al., 2015 Figure 2 showing the predicted P-wave velocity for a solid martian core? <https://www.pnas.org/doi/full/10.1073/pnas.1417490112>. A light element enriched solid seems unusual, based on my limited knowledge of Fe-partitioning. Why would there be any oxygen at all? Should that all stay in the fluid?

Line 221, Extended Data Fig. 5: Examining the fits to the data in this figure suggests that δp 5% is the better fit for both the bandpass and the polarization filter of S0235b over the author's preferred 10%; how was the $15 \pm 5\%$ arrived at? It seems you could define a waveform misfit and quantify the density misfit at the ICB. This number has big implications for the following discussion, so I'd like to know more about how this range was arrived at.

Line 256: "Fe-Ni alloy with 4-9 wt% O, 0-4 wt% C, and 0-2 wt% H in the IC" – How do you get O (oxygen) in the Fe-Ni alloy? Isn't oxygen incompatible with the solid alloy (iron) and would preferentially stay in the melt? It isn't clear to me how this modeling was done or how the light element partitioning into the inner core would work.

Lines 262-283: What about the dynamical role of a postulated basal magma layer? Wouldn't this impede efficient heat loss across the martian CMB that would drive inner core crystallization? Interestingly enough, the Moon also is thought to have such a layer, and the core also appears to be layered? Is there a similar process at work there? And how do you rectify the thermal state of the core, where the BML would suggest a hotter core, and an inner core would suggest a colder core? Aren't these two hypotheses in conflict with each other?

Methods:

Vespagram results: I am familiar with this method and have a few questions for the authors.

1) For a detection to be considered robust, how large of a time window and slowness window was searched? The plots in the main text (Fig. 2) and in the supplement (multiple figures of vespagrams) are very narrow slowness ranged (-6 to -8 s/deg for example). A robust detection would be a slowness that is well resolved and isn't surrounded by other arrivals at lower (or higher slownesses). Just showing the window for the predicted phase isn't enough, does the vespagram capture other body wave phases at the predicted times? It's easy to get some "noise" in a window that looks like "signal" you are searching if you don't evaluate it against the other arriving phases (for example, I'd expect P to have a nicely constrained beam as well as other phases, but the window is so tight there is no way to evaluate the quality of the vespagram outside the window shown, nor is the P result shown). Your Supplement showing the analyses of PPPP and PPPPP is a good start (I appreciate the care given to show some other phases in the time window!), but I'd like to see at least one global (large range of time and slowness) vespagram showing the same for early arriving phases as well (P, PP, S, and surface waves where appropriate).

2) Several of the vespagrams show multiple arrivals (e.g., PKKP in Fig. 2d) but power is normalized in the window to 1. In the supplement this is not done; and the other arrivals are similar in power to the one picked (e.g., P'P'r_ab in FS10 has multiple arrivals, and PKiKP in FS29a has two big arrivals). What determined your choice/selection of arrivals? Peak energy alone? Peak energy plus slowness? Either phase could be correct (or noise). The relative uncertainty in the following inversions should consider the uncertainties in these picks (and grow as a result), the bootstrap helps but doesn't necessarily avoid noise (you could be stacking noise, which isn't as easily suppressed if you use envelopes!). There also seems to be differences between what is presented in Fig. 2 and the supplemental vespagrams. What is the source of the amplitude discrepancy?

3) Azimuth resolution and the corresponding slowness resolution. In the Supplemental tests, the vespagrams degrade fairly rapidly with decreasing event count (thank you for showing this analysis), which is not unexpected; but given the low number of events in the stack for the vespa analysis, your slowness resolution will be primarily determined by the small number of events at different azimuths and distances (e.g., with a peak at 30-31 deg), and singular events at higher distances. I am less concerned about azimuth, but it seems like the exclusion of a singular distance can change the result quite a bit; the bootstrap handles this somewhat, but how do you equalize the events (the FPGA might help, but then again if a weaker event is flooded by noise, the event might not show a lower amplitude phase), given that the source magnitudes are not well constrained and attenuation of Mars is also poorly known?

Line 442, Methods: I was completely surprised (and a bit appalled) that the authors did not cite the Kim et al. 2021 InSight "Pitfalls" paper here. Their analysis is so thorough, why wasn't this paper referenced? The authors even go so far as to label some of the problems in the data as pitfalls(!), so it seems like there is some awareness of the findings in that essential paper for the martian seismologist. I would highly recommend the authors read this paper if they haven't, but I will qualify that recommendation in that their processing steps seem to have captured many of the essential problems that have been readily

documented in the InSight data. Reference follows:

Doyeon Kim, Paul Davis, Ved Lekić, Ross Maguire, Nicolas Compaire, Martin Schimmel, Eleonore Stutzmann, Jessica C. E. Irving, Philippe Lognonné, John-Robert Scholz, John Clinton, Géraldine Zenhäusern, Nikolaj Dahmen, Sizhuang Deng, Alan Levander, Mark P. Panning, Raphaël F. Garcia, Domenico Giardini, Ken Hurst, Brigitte Knapmeyer-Endrun, Francis Nimmo, W. Tom Pike, Laurent Pou, Nicholas Schmerr, Simon C. Stähler, Benoit Tauzin, Rudolf Widmer-Schmidrig, William B. Banerdt; Potential Pitfalls in the Analysis and Structural Interpretation of Seismic Data from the Mars InSight Mission. *Bulletin of the Seismological Society of America* 2021;; 111 (6): 2982–3002. doi: <https://doi.org/10.1785/0120210123>

Line 624: “it is impossible to have a solid inner core if the S content in the Martian core is greater than 12 wt %” I’m not entirely sure if this is a constraint, higher sulfur contents for the Moon were proposed in Antonangeli et al., 2015; and the same models could extend to Mars. Why is it impossible? More recent literature doesn’t seem to agree. It also seems like there is a strong dependence on the assumption of temperature as well (and the cited paper on Mars was published 2007, which is well before we had any seismology on the martian core). The presence of a basal magma layer seems like it should put some constraint on the CMB, are there a reasonable set of temperature models then for the core that also satisfy the MOI constraints?

(Remarks on code availability)

I did not have time to review the codes provided.

Referee #2

(Remarks to the Author)

This well-written and interesting paper examines seismic data from Mars, proposing to detect two signals influenced by a solid inner core. One of these signals is identified as a reflection from an inner core (PKiKP), the other as an early arrival of a phase which traverses the inner core twice (PKIKKIP, described in the paper as PKKP). Inversions are performed to obtain best matching seismic velocity profiles, utilizing the travel times and the proposed interpretation of these signals. The results are consistent with an inner core of radius approximately 610 km, and a compressional wave velocity jump of 30% at the inner core boundary. As a result, the authors propose an inner core rich in light elements generated by a snowfall model. While I enjoyed reading the paper, and am very supportive of the interdisciplinary aspect, I feel that the overarching conclusions related to Martian inner core growth are a little too speculative. The most fundamental issue is that the seismic observations central to the premise of the paper do not appear robust in the way they are currently presented.

My primary concern is that the main observations (vespagrams in Figure 2) are not convincing in the form they are shown, and this is the underpinning result on which the rest of the paper’s conclusions and inferences are formed. Envelopes are used due to difficulties in combining events with varying magnitudes, but the P amplitude could be used as a first approximation. Even the use of normalised amplitude retaining polarity would be an improvement, as the envelopes remove the polarity information which is required to identify reflections. Phase weighted stacking could be used in place of fourth root stacking. The vespagrams have unsuitable colour scales/palettes meaning they are difficult to evaluate as the smaller amplitude signals are masked. Colour scales should be sequential and even so as to image all signals, and these other signals should be identified as far as possible. Larger time/slowness windows should be used to place the signals in the wider context of others. This will provide a much more convincing and robust set of observations, and potentially more phases to include in the inversions.

Specific comments on Figure 2 and related observations:

- Amplitudes are presented as normalised envelopes meaning there is no information on polarity. This is especially important for identifying reflected signals.
- Amplitudes are normalised by panel and so there is no information on relative amplitude between panels.
- There are lots of arrivals that are not identified, particularly obvious in panel d, but the choice of colour scale means that they cannot be discerned easily.
- A diverging colour scale is not suitable for the measurements here; a sequential scale should be used. Otherwise, the small amplitude signals are mostly masked.
- The colour scales are uneven, so that the mid-point is not at the middle of the scale, which effectively masks smaller amplitude signals. The mid-point also varies between panels, and particularly noticeable for panel c, so they cannot be compared.
- The time/slowness windows are too small, panels a-c in particular. Larger windows would be more informative.
- Panel a appears to have very poor slowness resolution (streaks) – the phase appears to continue down beyond -11.0 s/deg. Similar streaks also apparently in panel c but masked by colour scale.
- Synthetic vespagrams presented alongside the observations would further help strengthen the figure, for example a comparison of a solid inner core vs a high velocity fluid inner.
- Vespagrams in supplement should be plotted using the same colour scale.
- Some of the supplementary vespagrams are not convincing, e.g. Fig. S21a is very streaky.

Minor general comments:

- 49-51 Normal mode data and the observation of PKJKP provide the conclusive observational evidence that the Earth’s inner core is solid.
- x is not defined on Figure 2.

- I really appreciate the inclusion of the code, and in a format straightforward to run online, but it is not commented at all.

(Remarks on code availability)

I am not familiar enough with Python to provide a meaningful review for the code. It seems to run fine, but it is barely commented, and some of the comments are not in English.

Version 1:

Reviewer comments:

Referee #1

(Remarks to the Author)

As this is a re-review, I will focus on the changes made by the authors from the original manuscript. I want to thank them for so carefully addressing my numerous comments and suggestions (and those of the other reviewer). To me, this is a really interesting paper and I think that it will generate discussion for years to come!

First the vespagram analysis is now incredibly thorough, and the inclusion of synthetic tests demonstrates that the data have the ability to resolve the slowness of arrivals when examined in this approach. The addition of the synthetics was a really useful extra test, and I appreciate the significant extra work that was likely required to include it. By incorporating a bootstrap approach, the authors have also further demonstrated the robustness of the detections and quantified if an arrival is resolved or not. Personally, I like this approach, it gives a quantitative way to evaluate which arrivals are robust, and is a bog standard approach in seismology for detecting small amplitude seismic phases. The FDPA and filter bank detections of the waveform phases really knock home that the authors are finding energy at the right polarization, component of motion, and are self-consistent across events at greater distance with the source array analysis of the vespagrams. I feel like the seismic detections are pretty rock solid at this point (well as solid as single station approaches get), and feel like the authors have done above and beyond the work to show that there is energy where they indicate in the data. It's a great example of how a source array approach can improve the science return from a single station. Bravo.

The inversion of the data observations are also now much clearer, and the results are better articulated, and the statistics better explained with uncertainties tied to confidence intervals (95%) and standard statistical tools (bootstrap, rms, etc) are being applied. The original manuscript was a bit vague on the actual uncertainties, and I think that the authors have much better clarified that density and velocity range (and depth range) uncertainty in their detections. These values (R_ICB of 613 +/-67 km, dVP_ICB of 32 +/- 8%, $drho_ICB$ of 7 +/- 5%) are the primary conclusions of the paper, and I think that the authors have done a great job using statistical tools and Bayesian inversion to put constraints on the data observations.

I'm not a mineral physicist, but I think that the hypothesis of a light-element enriched inner core is a bit more thoroughly explored in this version and a bit better justified than in the prior version. I cannot comment on its validity, but it explored some of the new findings in mineral physics about the more complex C-H-O-S systems at relevant martian pressure and temperatures that I think was missing from the original manuscript. I think that this part of the paper is far more interpretative and less well constrained, but in my book that is okay, as I think that the seismology result is provocative by itself and might require some reconsideration about what happens inside planetary cores by the modelers and experimentalists. A subject matter expert might have more detailed comments for them on it.

Line by Line Comments/Corrections:

Line 47: I think there is a typo here, S is repeated.

Line 111: Predicted travel times from what model?

Line 112: Is the predicted slowness discrepancy from the uncertainties in the model (i.e., the slowness and travel time rely on know the velocity structure of the core/inner core which are unknown?)

Line 130: this line is confusing, why not just say 0.25 km/s spread over 100 km?

Figure 2 (part g) the time scale for this panel is different from the panels above. Please plot on the same scale.

Line 177: What would the prediction suggest here? You have synthetics you can draw from to quantify?

Figure 3: Part b seems to be missing the colored text. What is the orange part? Part d has a typo, "tabel" is misspelled. To be honest, I don't think you need part d at all, the information is in the table and text. Part c has a strange title that is cut off. It would be helpful to see the spread in the velocity fit at the best fitting IC radius (this is written about in the text).

Line 227: Report the value from Irving et al, to illustrate the overlap in values

Line 267: maybe replace "form" with produce. Also, droplets sound like a liquid. Maybe crystals or metal particulates?

Line 267: The colder areotherm in direct conflict with a basal magma layer model (which requires a reasonably high temperature in the core, which would impede the snow model). You need a sentence here suggesting that the snow model can also explain the BML if the light element model allows for a hotter core. This was written about in the response to reviewers, but I did not see it discussed in the main manuscript.

Line 279: This is a pretty abrupt transition that there must be light elements in an inner core for this to work. Are there any other models aside from bottom up or top down crystallization? Like in an inner mush layer? What about anisotropy in the core itself (we do have some of that in the Earth's inner core), could that throw up a relatively high contrast (you're only sampling the core from one azimuth). How about energy loss, either from scattering or attenuation of PKiKP at the ICB that would make it weaker than PKKP that could be throwing off the reflection coefficient (and thereby underestimating the density jump)? What if the boundary isn't sharp, and more of a gradient? Would that affect the estimate?

Line 621: exercise "caution"

Line 1096: Martian is misspelled

(Remarks on code availability)

I looked through the code but did not run it (not familiar with python explicitly). The code now has much more detailed commentary and enough guidance to understand what each step is doing.

Referee #3

(Remarks to the Author)

This study is a very careful analysis of long period events observed on Mars by the InSight mission looking for phases associated with a potential inner core. The authors perform extremely careful data analysis, including all recommended pre-processing of a challenging dataset learned and published through the experience of the InSight team, and including a wide variety of techniques to pull out challenging phases to detect. The key observations are vespagram analysis of two phases identified as PKKP and PKiKP. The PKKP is significantly earlier than predicted by a fluid core model consistent with previously published PKP observations, while the presence of a PKiKP phase is only possible with the existence of a solid inner core. The vespagram analysis is based on a source array approach for the significant number of observed marsquakes in the 27-40 degree distance range, which is a similar technique to that used to identify ScS, which was a key constraint on the size of the martian core. Beyond the vespagrams, the authors also identify these phases and other core phases in individual events using a range of techniques also previously employed on InSight data for identifying body wave phases. The authors then also look for other lines of evidence to support the interpretation of these phases as inner core sensitive by looking at polarization and amplitude ratio, which while difficult to perform, also seem to be consistent with the phase identification. In response to reviewers, they've expanded the presented vespagrams to cover a broader region of time/slowness space, making the picks of these phases more convincing. If true, the presence of an inner core on Mars could have strong implications on the thermal and chemical state of the core, which is a critical element of understanding the planetary evolution of Mars. While this data is challenging, and the observation still has some potential weaknesses I will discuss later, this is about as robust evidence as possible with the amount of data we have for Mars, and appears to be strongly suggestive of the existence of an inner core on Mars.

The biggest concerns I have with the data analysis in the current manuscript relate to a) the existence of possible interfering energy at the arrival time of the PKiKP at greater slowness interpreted as a possible ScS-related phase, and the relatively low "occurrence percentage" derived from the bootstrap analysis of the PKKP phase.

First, for the energy interpreted as ScS-related energy, I suggest possibly trying to better analyze the polarization of the data. All vespagrams are "polarization-filtered" in this study, but I'll admit that I did not have the time to go through the code to see exactly how this was implemented. However, based on the description, this polarization filtering was done to emphasize linearly polarized signals. While both PKiKP and ScS would be linearly polarized (absent the effect of any shear wave splitting on ScS), the relative proportion of vertical and horizontal rectilinear motion should be very different between the two. Would it be possible to take advantage of this to more conclusively identify the origin of the energy at the two different slownesses in figure 2d, and therefore increase the confidence of the interpretation of the lower slowness arrival as PKiKP?

Second, for the bootstrap "occurrence percentage" plots, the PKKP arrival seems to peak at 25% of the bootstrap samples, which does not sound like a very high level of confidence that the signal is required by the data. I suspect the low value, though, is somewhat driven by the definition of occurrence percentage used, which only includes energy above 85% of the peak within the time-slowness space explored. Since this space includes at least 2 peaks (potentially interpreted in the paper as reflections from the CMB and the top of a Mantle Silicate Layer), it seems likely that the relative amplitude of those peaks could vary greatly between bootstrap resamples, meaning sometimes one or the other could be more emphasized by this plotting approach. This gives me a little pause on the use of this approach, as it is obviously highly dependent on the space explored and the choice of threshold, leading me to question the use of it to quantitatively define the picks and uncertainties, although it does appear to be a decent tool to understand which features in the vespagram are most robust.

Here's some more specific comments on the manuscript:

Line 47: "S, O, and H in addition to S" should presumably be "C, O, and H in addition to S"

Line 94: Defining slowness as the reciprocal of ray parameter seems wrong. Most textbooks I know of define the ray parameter in units of slowness, so in horizontally layered media, the ray parameter is the horizontal slowness. Perhaps this is a convention that is not universal, though, as both $\sin \theta/v$ or $v/\sin \theta$ would remain constant in classic applications of Snell's Law.

Line 122: This is the first of a couple places where the authors make a point about the early arrival being too early for a pure liquid core. However, as the authors do correctly point out in other parts of the manuscript, the velocity and velocity gradient of the liquid core is only very weakly constrained at this time, and the early arrival of PKKP could be explained by a higher

velocity gradient with depth which given the combined uncertainty of composition, temperature, and behavior of various chemical systems at martian temperature and pressure regimes is difficult to exclude a priori.

Line 181: The choice to not invert for mantle structure certainly does not "avoid possible issues with mantle heterogeneity and different measurement errors". It is a reasonable choice to make for this study, which is focused on core phases, and certainly the available data would not be enough to independently resolve mantle heterogeneity that may complicate measurements, but not attempting to use all data and invert for a whole Mars model, doesn't mean you're not sensitive to it. You're just trying make a reasonable simplifying assumption, while exploring some possible impact of this decision by using a range of possible models for the structure you don't invert for. I want to emphasize that I think this is the right and reasonable decision for this study, but I took a little issue with the statement that not inverting for mantle structure and using all available picks "avoided" the problem.

In the discussion section, the authors emphasize recent studies of Fe-S-O-C systems and suggest cores with significant amounts of C and/or O may be consistent with the presence of an inner core without requiring the core to be perhaps unreasonably low temperature. This was work I was not aware of, and the authors appear to make a convincing case that this is a possible model, which could even be reconciled with the existence of a molten silicate layer above the CMB, if that does exist as some studies have suggested. This is not my field of expertise, so I cannot assess these models thoroughly, but it does seem to set up reasonable spaces to explore for future mineral physics studies and geodynamic simulations.

Overall, while I do still think the measurements are challenging, this may be as robust an observation as possible without future seismic data, and definitely serves as a reasonable interpretation of the data that should drive future work and serve as motivation for possible future missions if possible.

I do not need to be anonymous, and this review is from Mark Panning.

(Remarks on code availability)

I appreciate the presence of the code, but I unfortunately did not have the time to thoroughly review it. The included code appeared relatively readable, but seemed to only demonstrate the MCMC inversion. Unless I missed it, the code for the vespagram calculation and presentation was not included, and this seems more central to the paper than the velocity inversions. The key significance of the paper is the observation of inner core phases, and the available code I saw does not reproduce that analysis, only the model inversion.

Referee #4

(Remarks to the Author)

Review of Bi et al.

This manuscript reports the seismic detection of the Martian inner core and argues possible core composition, in particular of the solid inner core. The presence of the Mars' inner core by itself has far-reaching implications, and I will be supportive of the publication of this paper as long as their seismological analyses are robust. Such robustness must be critically assessed by other referees, considering the fact that the presence of the Martian inner core has never been reported in a series of earlier seismological studies based on identical dataset. Since I am not a seismologist, my comments focus on the discussion part relevant to Mars' core composition and convection.

First of all, the authors' arguments on the possible core composition are hard to follow unless readers are really familiar with the phase diagram of Fe alloyed with possible light elements. They discuss chemical composition (practically liquid composition), then crystallization, and finally consistency with seismological observations. Alternatively, they may first discuss which phase (Fe, FeO, Fe₃C, or FeH) matches the inner core observations.

They argued a mixture of FeO, Fe₃C, and stoichiometric FeH in the current ms, but indeed mixture (or layering) is not really likely. Yokoo et al. (2024) demonstrated that >4wt% C is necessary for Fe₃C crystallization to occur. It may not be feasible because of the known simultaneous solubility limit of S and C in liquid Fe. In addition, the crystallization of FeH is also unlikely because its melting temperature (in other words, crystallization temperature) is low below 40 GPa where the melting temperature of stoichiometric FeH is not a temperature maximum in the Fe-H liquidus phase diagram yet (Tagawa et al., 2022). These suggest the Martian inner core, if it really exists, is most likely to be single-phase FeO (not a mixture or layered). Considering that the inner core is composed only of FeO, they may argue the possible range of the liquid outer core composition, whose liquidus phase is FeO and density and velocity are consistent with observations. It is much simpler and readable.

Second, the last paragraph of the main text on the Martian dynamo includes almost nothing really meaningful. It is very important to discuss why the Martian core undergoes inner core crystallization but it does not drive liquid core convection. Does the authors show that upon FeO crystallization, a residual liquid becomes depleted in oxygen and forms a dense liquid layer above the inner core? Or, it is also possible that crystallization is too slow and does not provide power large enough to drive liquid core convection. The authors can discuss the cooling rate of the Martian liquid core quantitatively. I believe such modeling is not difficult.

Specific comments:

Line 265~:

When one talks about crystallization at the top and “iron snow”, a dense Fe-rich phase descends and then melts away. It makes compositional stratification in a liquid core (depletion in light element in a deeper part). The solid inner core appears only after light element concentration in a liquid becomes low enough for solid Fe to crystallize in-situ at the centre. The process is more complicated than written here.

Line 275~ in the main text and Line 724~ in Methods:

They discuss the velocity of liquid stoichiometric Fe₃S (I believe it is not a typo), but liquid Fe₃S does not crystallize solid Fe at 35 GPa (Stewart et al., 2007). Moreover, why does a core liquid have a stoichiometric composition?

Line 290:

FeO is not an alloy (metal) but a compound.

Line 293:

“leaving only trace amounts of O and C in the liquid outer core”

Why? More explanations are necessary.

Line 716:

“Sulfur (S) is considered to be the primary light element in the Martian core, with a content ranging from 10.6 wt.% to 16 wt.%”

Yoshizaki and McDonough (2020 GCA) proposed 6.6 wt% S.

Line 758:

“VP of Fe₃C and FeH_x can be derived following the Birch’s law”

First of all, why not argue the velocity of FeO here? It is critical. Second, this equation is not for FeH_x but for stoichiometric FeH. Third, the Birch’s relation is likely to be temperature dependent (see Sakamaki et al., 2016 Sci.Adv.), and thus the authors should, at least discuss the effect of temperature.

Extended Data Fig. 8:

The figure shows a difference by 0.01 wt% H makes a large difference in the density jump across the Martian ICB. But it is most likely wrong. In addition, the main text always considers the 7% density jump across the ICB, but why 8–12% here?

Extended Data Fig. 9:

“The content is indicated as mol percent”

It should be a fraction.

(Remarks on code availability)

Version 2:

Reviewer comments:

Referee #1

(Remarks to the Author)

First, I want to thank the authors for all their hard work on responding to my many comments. I feel that their revisions and responses have made the article of the highest quality possible given the challenges in the InSight data. The supplement is incredibly thorough, and addresses many key questions to my satisfaction. I'm content with the paper as is (one minor cosmetic comment below) and don't have any additional questions/suggestions for the authors at this time.

I am happy to be identified as a reviewer.

Cheers,

Nick Schmerr

One minor comment: Figure 4: The colors used here don't match the implied mineralogy; for the Earth the upper mantle is blue, the transition zone aquamarine, and the lower mantle light blue, while for Mars the mantle is light blue (implying similarity to Earth's lower mantle, although Mars doesn't have a lower mantle). Shouldn't the implied mineralogy match (i.e., the martian mantle should be nearly wholly blue, with some aquamarine (post spinel) at the bottom? Or perhaps use a different shade of blue entirely given the differences in composition between the Earth and Mars?

(Remarks on code availability)

Referee #3

(Remarks to the Author)

This is a re-review of the paper, and the overall summary of the key points of the paper remain as before: i.e. that the authors

perform vespagram analysis of InSight mars quakes between 27 and 40 degrees distance and identify PKKP significantly earlier than predicted by existing liquid core models as well as a PKiKP phase indicative of a solid inner core at a radius of ~600 km. They argue that such a core is consistent with solidification of an O-rich inner core, although this conclusion (rather than a more C-rich core) relies on determination of a density contrast from an amplitude ratio observed on only one event, which I would argue is not a particularly strong constraint. I feel that the authors have addressed most of my major concerns from my previous review, and I would encourage this to be published after minor revisions.

I would say my most significant remaining concern is about the observation of the two possible PKKP arrivals interpreted as possibly arising from reflections off a Molten Silicate Layer (MSL) in addition to a reflection from the CMB. There are indeed 2 possible arrivals in the vespagrams, and it is interesting that they seem to correspond to the separation that would be expected for a 150 km layer. However, in each of the individual events analyzed for PKKP in the Supplementary Material B, only one PKKP arrival is identified, although it is not clear if the time window shown in those analyses actually would show the earlier potential PKKP arrival. The synthetics do show the amplitude of the reflection from the top of the MSL is stronger than the true CMB for those models, so it is perhaps not surprising that it may be harder to detect in individual events, but seeing both arrivals clearly in one or more events would help give confidence in seeing both. If that is not, however, possible, it might be good to call that out in the main text, suggesting that such an MSL is possibly consistent with the data, but not necessarily required (which I think is a reasonable interpretation).

More minor typographical and presentational comments:

Page 6, line 112: "models" should be plural here, not "model"

Fig.2: Many labels in this figure are quite difficult to read, particularly the white font on the blue vespagram background. The font should likely either be enlarged or have a box behind it. Also, the ranges of the y axis on the VRM-HRM plot are a little strange. Is there a significance to the choice for the lower limit on the axis? Showing this down to 0 (or lower) may make it easier to tell how much above the noise level of this particular metric the identified signals are. If you can define a significance threshold based on the statistics of the variation of that metric, showing that as a dotted line would be preferable to simply cutting the plot off at that level. And if that level is not defined by the data, cutting it off there feels very arbitrary. I think this is a real observation, but the limited range seems strange to me.

Page 8, line 173: Once again, as stated in my last review, not inverting for mantle structure does not "minimize possible issues with mantle heterogeneity", but instead simply neglects it.. I agree that this is the right decision for this study, but the text should reflect this. I would instead say that you look at the sensitivity to mantle structure by using a range of possible mantle models and achieve similar results, suggesting that not simultaneously inverting for mantle structure does not have a large impact on your results.

Page 12, line 248: I understand that this is the best constraint on density jump that you can achieve, but it is a very weak constraint. Individual reflections off an interface that may have topography can vary significantly due to focusing and defocusing. If you have lots of observations, that variation may cancel out and give you confidence in using amplitude ratios to solve for the reflection coefficient and therefore density jump, but I would argue using the ratio from only one seismogram could have errors of factors of 2, 3, or more, meaning that just getting an error bar on the density jump by only considering the possible error in determining the amplitude in the presence of noise, but still assuming a layer cake model for the reflection coefficient is vastly understating the uncertainty. You do already make reference to this in the text, but I would call it out again when discussing the possible core composition models. It is fine to prefer the O-enriched model to match this density contrast, but I think you need to clearly call out that the density jump could be very different, and so it is difficult to eliminate the C-rich model. While simply using that density jump would also likely lead to a model with too high a mass and too low moment of inertia if everything else is kept constant, it would likely be quite possible to solve for a model consistent with mass and moment of inertia if you inverted for whole planet structure. I do too think this is required for this study, but it should be acknowledged that the density constraint is not very strong.

This review is by Mark Panning.

(Remarks on code availability)

I did look at the code link, and I am happy to see that the authors have added in the vespagram codes that were missing from the last submission, but I did not attempt to review the code in detail.

Referee #4

(Remarks to the Author)

Re-review of Bi et al.

I am happy to see that the authors now argue for the FeO inner core, consistent with their observations. On the other hand, their discussion in pages 14-16 on core cooling and convection and Martian dynamo is still very poor and provides the least new insights (the same comment as I made in the previous round of review). I recommend the authors to limit their discussion only to the FeO inner core, which is directly related to their seismological observations reported in this paper. Further discussion written in pages 14-16 should be fully removed. More specific comments are found below.

Major issues:

1) Page 13

The authors argue the inner core constituent (Fe, Fe₃C, or FeO) based only on the density and velocity "jumps". However,

the inner core VP profile is given Fig. 3a, and they should directly compare the observed VP and those of Fe, Fe₃C, or FeO at corresponding high pressure and high temperature. It seems that the VP of FeO matches the observations.

2) Page 14

The authors mention “the presence of a solid Martian IC would imply efficient core cooling if the core was very hot in the past”, but the presence of the inner core does not necessarily mean efficient core cooling (temperature could have been low from the beginning). The following discussion in Page 14 is not new/important and should be fully removed.

3) Page 15–16

Their discussion on Martian core convection and dynamo in the last 2 pages is not well written and not meaningful. They should remove the last three pages (pages 14–16) from the paper. Indeed, their finding of the FeO inner core is good enough for this paper.

As I requested in the previous review, they can briefly discuss why the FeO inner core crystallization does not drive liquid core convection. This is a very important question directly relevant to the FeO inner core, but the authors did not really respond to my request. The compositional buoyancy of liquid derived by crystallizing FeO at the inner core can be approximated by a comparison between the Martian outer core liquid density and the density of liquid FeO. The latter is obtained by its liquid equation of state reported by Morard et al. (2022 JGR) (see their Supporting Information S2).

Minor comments:

4) Line 266–268

Please revise the text into “the crystallization of an Fe₃C-dominated solid IC requires > 4 wt.% C in the liquid core, resulting in an Fe-S-O-C outer core”.

5) Line 268–269 & 271–272

“such a composition (12-16 wt.% S, 0-6 wt.% O, and 4.0-4.7 wt.% C)”

This composition suddenly appears, which confuses readers including myself. I realized that this specific range of composition is from Fig. 7a in Yokoo & Hirose (ref. 21) by considering the volume of the inner core observed in this study. This should be explicitly mentioned.

“12-16 wt.% S, 6.7-9.0 wt.% O, and ≤ 3.8 wt.% C”

Same comment as above.

6) Line 278–279

“an O-enriched core can crystallize even at elevated core temperatures exceeding 2200 K at the ICB, potentially supporting the existence of a MSL above the core”

This statement is correct but is hard to understand. Please add more explanations. It is possible that the Martian core temperature is higher than the estimate by Khan et al. (2022), which was employed by Yokoo & Hirose (2024) who proposed these C-rich and O-rich possible Martian outer core compositions. If this is the case, a liquid core containing >9.0 wt% O can crystallize FeO when >2200 K at the centre (see Fig. S11 in Yokoo & Hirose, 2024). Such high core temperatures might support the existence of a MSL above the core.

7) Line 308–309

“Additionally, even if there is some ongoing compositional convection, the presence of stable thermal stratification in the outermost parts of the core may inhibit dynamo action.”

This is not true. Such thermal stratification due to high thermal conductivity has been proposed to explain the observed 300-km thick low-velocity layer atop the present-day outer core of the Earth. It does not inhibit a dynamo action.

(Remarks on code availability)

Dear Editor John VanDecar and Reviewers,

We thank you and the reviewers for thorough and insightful comments. We believe the reviewer's impression is due to insufficient clarity in our original manuscript, particularly regarding on the robustness of the phase identification in the vespagram analysis and the uncertainty analysis. We appreciate the valuable suggestions on technical details, including the importance of amplitude and polarity information. These constructive comments have enhanced our understanding of the data and strengthened the robustness of the identified inner-core related phases. Below, we briefly outline the major changes we have made.

1. We have significantly revised the "Methods" section by adding subsections "Travel Time Picks and Measurement Uncertainties" and "Amplitude and Polarity of PKKP and PKiKP" to clarify how we determine uncertainty, relative amplitude, and polarity between PKKP and PKiKP during the array analysis.
2. We adopted a more quantitatively way to enhance core phase identification and uncertainty estimation. Using a bootstrap resampling method, we calculate the occurrence percentage with high energy for each grid in the vespagram. By fitting the distribution, we derived mean values and associated uncertainties for the targeted phases, improving the robustness of our phase identification, as illustrated in the new Figure 2.
3. We included discussions on the choice of different stacking methods for both waveform and envelope data. Thanks for these important comments. We find that by deconvolving PKiKP waveform from PKKP, we achieved a stable vespagram and successfully measure the relative amplitude and polarity between PKKP and PKiKP.
4. We recalculated vespagram over a much larger time and slowness window, and all vespagrams are now presented with a more appropriate color scale.
5. As noted by reviewers, a robust vespagram analysis is the central to this manuscript. Therefore, in the revised manuscript, we focused on vespagram analysis and did inversions using travel times picked from the vespagram. Most results and discussions related to the individual picks have been moved to the supplementary materials.
6. Following reviewers' comments, we have shifted our focus to the identification of the Martian inner core. Instead of discussing detailed models of Martian core evolution, we now emphasize possible mineral explanations for the inner core velocity and density, as well as implications for Martian absence of a current dynamo.

The quotes from the reviewers are followed by our responses in *blue italics*.

Referee #1:

Review of "Seismic Detection of a 600-km Solid Inner Core on Mars"

This paper gives seismic evidence provided by the InSight mission's seismic experiment SEIS for the presence of a solid inner core in Mars. The authors used a source array analysis of 23 marsquakes to search for core interacting phases and found candidate arrivals that match predictions for PKKP and another arrival that the authors interpret as PKiKP, a reflection from

a postulated inner core. The authors then inverted the arrival times of these phases to determine that the PKiKP phase indicates an inner core of ~610 km, with a P-wave contrast of ~30% into the inner core and used waveform modeling to infer a density contrast of ~10%. The authors then interpret their findings using mineral physics to infer that the inner core must contain light elements and be relatively low temperature; and state that their findings have implication for the generation of dynamo in the martian past and understanding of the evolution of terrestrial planets in general.

The detection of an inner core inside Mars is an important and somewhat puzzling finding; the martian core has generally been considered too small and enriched in light elements to support formation of an inner core, and thus a robust detection of such a structure would be an important contribution to understanding the evolution of smaller bodies in the Solar System. That said, Nature seems like an excellent fit for presenting the authors' work and getting the conversation started in the literature about this topic from an observational standpoint. A similar finding for the Moon by Weber et al., 2011 in Science, required a core that is much higher in sulfur (or other light elements) content than previously thought, and it seems that a similar implication would exist for Mars. The presence of a lunar inner core is still controversial. Presumably such inner core formation for Mars would be recent, so it is perplexing that the authors chose to discuss implications for the past dynamo, but it seems like that the compositions required to get to an inner core may have a more important role in that. However, all of this is predicated on the robustness of the seismology observations (the key component of this paper), which is essential for determining the presence, size, and properties of the inner core (if it exists). The InSight data are challenging in a variety of ways (a vast understatement perhaps), but at the same time are powerful constraints on Mars; my familiarity is primarily in the realm of seismology, so I will focus my comments and criticisms there.

I want to say from the forefront, that the seismological analysis here is thorough and carefully follows many of the established approaches used by the InSight team itself. I must lament that most of that care is buried in the supplement to the paper, but that is the challenge of shorter papers. The authors have done a detailed job of using multiple working lines of evidence for their phase detection analysis (vespagrams created with source arrays, filter banks to identify phases on single seismograms), which closely mirror some of the same approaches used by the InSight team itself to establish the phases detected by the mission (like ScS, and SKS, mantle triplications and other seismic phases) that are now primary constraints on the deep martian interior. Martian seismology is notably tough, so congratulations to the authors for doing such a thorough and careful job! They present several lines of evidence for the presence of an inner core: 1) An early detection of PKKP (on vespagrams and single seismograms with polarization filtering), which interestingly enough seems to also have evidence for the somewhat controversial basal magma layer postcursor and is early enough (~100 seconds) that it makes sense for a high velocity thing (like an inner core) to be in the way. 2) Detection of a PKiKP phase both in vespagrams and in single seismograms at a depth that would put the inner core around ~600 km in radius, 3) the identification of other core interacting phases and several mantle phases to illustrate the robustness of their approach. I've outlined a few things I'd like to see to better shore up those detections, but to my eye their detections seems nearly

as robust as any of the other InSight data papers published in the literature about body waves detected in Mars and should certainly be entered into the scientific record (and hopefully follow on studies can refute or verify). My main seismology concern is that the answers do not seem to fully incorporate some of the uncertainties on the measured arrival times and amplitudes, and that the vespagram evidence presented in the main paper and supplements sometimes seems to have discrepancies that need some explanation. In terms of the seismological modeling and inversion, these approaches to my eye are robust and use standard seismological methodologies (Bayesian approaches, waveform matching, forward wave propagation modeling); but again, I'd like to see a more complete picture of the uncertainties since those flow readily into the geodynamical and mineral physical models that follow. I will let colleagues more fluent in the mineral physics and nature of cores comment on the implied compositional models, but there are a few things that seem rather odd, like an oxygen enriched inner core, which even to my novice eye is seemingly not compatible with how elements partition into solidifying Fe that would form the inner core. Also, the authors do not address the thermal implications of an inner core (lower core temperature) and how that matches with the thermal implications of a basal magma layer (temperatures high enough to sustain a silicate melt layer). Regardless of the interpretation for the mechanism generating an inner core, it seems that the seismological implications for the martian core and its evolution will be a big contributor to those future discussions.

Thank you for your thoughtful and encouraging comments. We greatly appreciate your recognition of the potential contribution of our paper. We wholeheartedly agree with you and Review#2 regarding the challenges of handling InSight seismic data and the limitations posed by a single seismometer. We have found that working with even the most common mantle seismic phase can be quite complex, and we are aware the need for caution when interpreting any single observation on any phases. We appreciate your emphasis on the multiple observations supporting our conclusion on the existence of the Martian inner core (IC). The detection of PKiKP, the early arrived PKKP, the P'P'_df sampling the IC, and the new PKIIKP phase you suggested all provide complementary evidence that reinforces our findings.

We apologize for not presenting the vespagram analysis and related uncertainty assessments in a clearer manner in the original manuscript. In response to your and Reviewer #2's suggestions, we have revised the "Methods" sections and added new subsections on "Travel time picks and measurement uncertainties" and "Amplitude and polarity of PKKP and PKiKP". In addition, we have included a more extensive discussion on the distribution of the light elements in the core and the potential thermal implications of an inner core.

Detailed Comments

Title: "Core on Mars" – shouldn't it be core "in" Mars? I'd recommend that change for clarity.

We have changed the title as "Seismic Detection of a 600-km Solid Inner Core in Mars".

Line 31: The Earth's inner core is postulated to be relatively young (a few billion years), and may not have a huge role in the history of the magnetic field; thus this opening statement is a bit puzzling. Are the authors suggesting the inner core has been around since the Noachian? If so, what is the evidence for that? For Mars, it appears that the dynamo died very early in the evolution of the planet, so it's not entirely clear how this statement holds in terms of shaping

the magnetic evolution of the planet. Perhaps there are clues in the presence of an inner core on the composition and temperature of the early martian core, but I'd suspect that any presence of an inner core would have little effect.

Sorry for the confusion here. Here, we want to state that the formation and growth of an inner core could potentially have important effect on the operation style of the dynamo by introducing compositionally driven convection in the outer core [Breuer et al., 2015; Stevenson et al., 1983]. You are correct that, as shown in Hemingway and Driscoll's [2021] calculations, the presence of an inner core does not necessarily guarantee dynamo activity in most cases. Thus, we have revised this sentence to "The existence of an inner core (IC) within a planet holds significant importance in planet evolution, as its growth profoundly impacts the planet's thermal state and dynamo processes." at Lines 30-32.

Line 35: There is seismological evidence for a lunar inner core (Weber et al., 2011) from the Apollo seismic experiments, which consisted of 4 seismic stations (over Insight's singular station). How is that detection any less decisive than the identification made in this paper? The manuscript actually says this exact thing later on Line 148. Or are you distinguishing terrestrial "planets" from "moons" – if so, I'd argue that's not a very meaningful distinction.

Thank you for highlighting this. We have revised this sentence to "While Earth has a well-established IC, decisive identification of an IC has not been made for other planetary bodies, aside from the Moon." at Lines 34-36.

Line 39: "absence of chemical convection associated with inner core growth" – I'd think that the terrestrial dynamo existed well before the presence of an inner core on Earth (the oldest rocks tell us these things), so this statement seems strange. Dynamos need a conductive fluid, planetary rotation, and a source of convective motions; on Earth this could be a combination of things, but I'd suspect that the nature of heat exchange over the CMB plays a bigger role than the inner core, and would be a feature present for nearly all of Earth (and martian) history.

We agree that the dynamo is not exclusively driven by chemical convection, although it may play an important role in the late stages of the core evolution. To avoid confusion, we have deleted this sentence and rewritten the sentences at Lines 37-41.

Line 40: "snowing down" this is a pretty speculative mechanism, and I don't know if we can sort this out on the Earth where geophysical data is abundant, much less so Mars, where we have a very limited dataset from seismology and much less so from other data types. The real finding here is that Mars could have an inner core, do we have to have a model that explains the observation for it to be an observation?

We have removed this sentence concerning different types of crystallization.

Line 57: "potentials" should be potential.

Have corrected.

Line 63, Fig. 1: I'm not sure the elevation data here are particularly helpful, the color palette makes it hard to see the bounce points and some of the lines are over the labels for events. A

few geographic labels are really all that is needed to establish the locations of seismic information, so I'd recommend simplifying panel c by removing the topography.

Thank you for this suggestion. We have updated the topography map with a grayscale color and adjusted the colors for other labels accordingly.

Line 98 (and Line 103): "Striking" and "closely aligning" and "strong evidence" are all qualitative and imprecise descriptions for the observations you show here. The observation of P'P'n in Fig. 2b falls outside the prediction window, and PKKP is over 100 seconds from the prediction (more on that later). It should not come as any surprise that the "reference" models would be imprecise where the martian core is concerned (we have limited observations), but is that error commensurate with the uncertainties? To address this, I would recommend that you quantify your results, the arrivals shown in Fig. 2a-b come between the predictions, for example, in 2a (P'P'r_ab) falls at a slowness of 9.1 s/deg (+/- some error), and arrival time of 1830 s (+/- some error); this falls within the +/- 10 seconds from the prediction, and just below the predicted slowness of -9.5 to -10.5 s/deg (and predictions here are qualitative too, given the uncertainties in the deep martian models).

Thank you for this suggestion. We have removed the qualitative and imprecise descriptions here and throughout the manuscript. We have now provided a detailed explanation of how we obtain the uncertainties for both travel time and slowness in the "Methods" section, presenting them in a more quantitative way. In Lines 109-112, we now state "Examples of P'P'r_ab and P'P'n are shown in Supplementary A Fig. S16, with travel time of 1835 ± 4 s and 2008 ± 3 s, slowness of -12.1 ± 0.7 s $^\circ$ and -5.0 ± 0.6 s $^\circ$, relative to P, respectively. These arrivals fall within the predicted travel times and are slightly below the predicted slowness of -10.8 to -9 s $^\circ$ and -4.0 to -3.7 s $^\circ$ for both phases."

Line 139: Why not show where this prediction would fall for the two models proposed on Fig. 2d? I think that would help to strengthen this point.

We have reorganized Fig. 2 by including the predicted vespagrams for the model with an IC (new Fig. 2c and Fig. 2f for PKKP and PKiKP, respectively), as suggested. However, since the predicted energy for PKKP_{CMB} is lower than that of PKKP_{MSL}, making it difficult to see the PKKP_{CMB} nicely while using the same colorbar for all plots, we opted to only plot the predicted vespagram for the model without the molten silicate layer. We have marked the predicted travel time and slowness for both PKKP_{MSL} and PKKP_{CMB} in Fig. 2c.

Line 141, Figure 2: I agree that the early arrivals of PKKP and its postcursor are commensurate with a faster object in between and that this is a good line of evidence for a solid inner core (or at least a faster lower core). But how much faster? You do not quantify this (yet). Furthermore, I am a bit puzzled by the difference between the results in Fig. 2d and those in Fig. S13; the vespagrams are not identical, and the earlier arrival near 1290s seems to be lower amplitude in Fig. S13 than in Fig. 2d. What is the source of this discrepancy (see note about vespagrams below)?

We have quantified the velocity change here and modified this sentence in Lines 128-132 as: "the significantly early arrival of PKKP indicates a much faster core toward the center, such as a velocity gradient of about ~ 0.0025 s $^{-1}$ steeper (0.25 km/s faster over 100 km) in the central

880 km of the core compared to the shallow core (Supplementary Information A Section 3.2.2.1.3)".

The Fig. 2d and Fig. S13 in the original manuscript are slightly different. Fig. 2d shows the mean vespagram of PKKP calculated from 200 times bootstrap resampling tests. In contrast, Fig. S13 is generated from a straight vespagram analysis using all 23 events. While we expect them to be similar, they are not identical. Furthermore, in Fig. 2d, the vespagram amplitudes are normalized to the maximum value. We apologize for the unfortunate decision to use different color scales, which made comparison difficult, as noted by Review#2. In the revised manuscript, we have reorganized supplementary materials and use the same color scale for all vespagram. We believe these changes have significantly improved the clarity of both the Methods and results sections.

Line 155 (and Line 208): If PKKP is indeed sampling the inner core and getting sped up, technically, wouldn't this phase then be called PKIKKIKP? Wouldn't you technically predict a PKIIKP as well (a reflection from the other side of the core), on top of P'P'r_df? Did you search for this phase after PKiKP? It seems like that phase should exist if PKKP does (and P'P'r_df), unless the inner core is strongly attenuating for P waves (in this case you wouldn't predict P'P'r_df to show up!), or the geometry is not particularly good for producing the reflection (e.g., it would be at a different distance). But it seems like it should be there if these other phases are showing up as well. Another line of evidence that could further support (or refute) your interpretation?

Yes, a PKKP transiting the inner core should be named as PKIKKIKP. Here, for consistency with a model without IC, we simply adopt the name as PKKP. We have included this clarification in the caption for Fig. 1, at Lines 70-71.

Thank you for this wonderful suggestion on PKIIKP. We examine the signals after the PKiKP and indeed observe arrival consistent with the expected slowness and travel time of PKIIKP (new Extended Data Fig. 6g). Although PKIIKP is tough to detect due to its amplitude being only half that of PKKP and possible interference of the strong PcSScP, we are quite excited about this new evidence, which further supporting the existence of the IC. This new observation also validates the effectiveness of our vespagram analysis. We have incorporated this finding into the section of "Further support for the inner core model" at Lines 248-251.

Line 166: Why do you report a different error (± 90 km) here than in the abstract (± 50 km)? For this number, I am a bit skeptical of even ± 90 given the uncertainty in the martian reference models (e.g., the structure of the crust and upper mantle, plus the uncertainty in core properties and lowermost martian mantle) that you can get anything this precise. I agree that the paths through the mantle are similar for PKKP and PKiKP, but the CMB on Mars seems like it is relatively complicated (i.e., can really slow down a wave at something like the basal magma layer). Regardless if you support that model or not, it seems reasonable that CMB structure would be highly variable and could map onto the PKKP and PKiKP differential times (much as it does on Earth). What happens in the inversion if you increase the uncertainties of the 1-D reference models?

Line 175; Fig 3. Why only use SKS_GD? And what happens when you remove the 3 events at greater distance?

Sorry for the confusing here. The ± 90 km noted in Line 166 of the original manuscript is from an inversion of the entire core velocity based on the SKS_GD mantle model. In the Abstract, we present a value that assumes fixed velocities for both the mantle and outer core as the SKS_GD model, produces a reduced uncertainty of ± 50 km, as described in Lines 195-200 in the original manuscript.

Our explanation of the inversion processes may not have been clear. To clarify, we have explored different mantle models, including the mean, lower, upper bound V_P models from the AK_subset models, as well as the two inverted V_P models incorporating SKS travel times (SKS_GD, SKS_GP), and the two models with the basal magma layer (MSL_IPGP, MSL_ETH) (new Extended Data Table 2), which cover a wide range of mantle model spaces.

We have revised this section of the manuscript to avoid confusion. In Fig. 3a-c, we use the inversion results from the SKS_GD model as a representative example. Results for other mantle models and different inversion strategies are now included in the new Extended Data Table 2, along with corresponding plots in new Fig. 3d and Extended Data Fig. 5. As shown in Fig. 3d, the values of R_{IC} are quite consistent across the different mantle models we tested. In the revised manuscript, we have averaged the results from different mantle models and use it in the Abstract.

We totally agree a heterogenous mantle, particularly the lowermost mantle, could significantly affect the differential travel time between PKKP and PKiKP ($\delta T_{PKKP-PKiKP}$). We tested several 1D models, including structures resembling ultralow velocity zone at the CMB in the original mantle models or perturbations in the bottom 500 km of the mantle. As shown in the new Supplementary A Fig. S29 and Section 4.1, $\delta T_{PKKP-PKiKP}$ for these new models varies little. However, as you commented, 3D mantle structures are likely to exist and would indeed affect the travel times of PKKP and PKiKP differently. Here, we also consider models in which we perturb the outer core velocity by $-10 \sim 10\%$, which can cause $\delta T_{PKKP-PKiKP}$ varying up to 60 s. Alternatively, a 5% change in the outer core velocity could cause a variation of R_{IC} by as much as 100 km. It is important to note that such changes would also significantly affect the travel times of $P'P'$. Thus, an inversion incorporating all four phases could provide more robust constraints on the outer core velocity.

To address these uncertainties in our mantle and outer core models, we might choose to include them in the uncertainties of the travel time measurements. Nevertheless, in the current manuscript, we chose to focus on presenting evidence of the existence of IC and adopt a simplified inversion approach. We have included a discussion at Lines 198-200 as “However, we recognize that three-dimensional mantle structures or uncertainties in the outer core velocity could increase the uncertainty in the estimated IC size (Supplementary A Fig. S29).” and Lines 679-682 as “Since PKiKP and PKKP travel with nearly identical paths in the mantle (Fig. 1b), their differential travel time are primarily affected by the core structure, with only minimal impact from the 1D mantle structure variations (see Supplementary Information A Section 4.1 for more details)” in the “Methods”.

Line 192: I'd report your error bars on these values, which seems to be ± 100 km for the R_{oc} , and ± 0.5 km/s for the $V_{p_{oc}}$?

We have added the uncertainties in the revised manuscript.

Line 199: This seems like it's on the high end of predicted solid core velocities from Antonangeli et al., 2015 Figure 2 showing the predicted P-wave velocity for a solid martian core? <https://www.pnas.org/doi/full/10.1073/pnas.1417490112>. A light element enriched solid seems unusual, based on my limited knowledge of Fe-partitioning. Why would there be any oxygen at all? Should that all stay in the fluid?

Under Martian core conditions, Fe is expected to be in the fcc structure. Here we used the experimental results of Kantor et al. [2007] for the velocity and density of fcc Fe which are in agreement with reported results in Antonangeli et al. [2015]. It should be noted that the inclusion of light elements increases the V_P and decreases the density of Fe. This explains why V_P in our modeling is greater than the reported value in Antonangeli et al. [2015] for the pure fcc-Fe.

The formation of Martian inner core is quite complex and strongly dependent on the pressure, temperature and composition. When Fe coexists with Fe alloys, a complex eutectic system is formed. Below, we use the FeO-Fe system as an example. In Oka et al. [2019] with 4.1 wt.% O in the whole system, only ~ 0.4 wt.% O remains in the liquid Fe at 29 GPa and 2260 K when FeO starts to solidify. Seagle et al. [2008] also showed that the eutectic line in the Fe-FeO system at 50 GPa is 2500 K. Below the eutectic temperature line, the system is entirely solid which is composed by Fe and FeO. When the temperature exceeds the eutectic line but is below ~ 2650 K, oxygen will partition between the solid and liquid phases depending on its content. Combining with the phase diagrams of the Fe-O system in literature works, we infer that under the pressure conditions corresponding to the Martian core, the eutectic line of the Fe-O system is ~ 2300 K, and the system would be completely molten at ~ 2450 K [Boehler, 1992].

During the evolution of the Martian core, the initial high temperatures likely caused all the Fe-alloys to be in a molten state. Based on the experimental results of Oka et al. [2019], when the core temperature gradually decreased, FeO would precipitate from the molten state to form the solid inner, leaving only minor amount of O in the residue liquid phase. As reported by Oka et al. [2019], the residue O in the liquid phase would constitute only about one-tenth of the total.

[REDACTION]

[REDACTION]

Left: Phase diagram of Fe-O system from Seagle et al. [2008] under 50 GPa

Right: Melting temperature of Fe-FeO system up to 100 GPa

Line 221, Extended Data Fig. 5: Examining the fits to the data in this figure suggests that dp 5% is the better fit for both the bandpass and the polarization filter of S0235b over the author's preferred 10%; how was the 15+/-5% arrived at? It seems you could define a waveform misfit and quantify the density misfit at the ICB. This number has big implications for the following discussion, so I'd like to know more about how this range was arrived at.

Thanks for this comment. In the original manuscript, we did not formalize the uncertainty analysis on the density jump at the ICB ($\delta\rho_{ICB}$). We simply determined the value by comparing the amplitude ratio between PKiKP and PKKP between observation and synthetics. As shown in the original Extended Data Fig. 5a for the bandpass filtered waveform data, a 15+/-5% provide a reasonable fit of the amplitude. It is worth to mention that we do not use the polarization data here, because the amplification factors for PKKP and PKiKP are different in the polarization filter.

In the revised manuscript, we formalized the estimation of the $\delta\rho_{ICB}$ as shown in the new Extended Data Fig. 7c. In the bandpass filtered data, the amplitudes of PKKP and PKiKP of event S0235b are determined in the target windows and their amplitude uncertainties are measured as the maximum amplitude in the noise window, defined as 5-15 s before the main phases. This allows us to calculate the amplitude ratio between PKKP and PKiKP along with its uncertainty, which shows to determine a $\delta\rho_{ICB}$ of $7 \pm 5\%$ (new Extended Data Fig. 7c). In the "Methods" section, we have included a subsection of "Amplitude and polarity of PKKP and PKiKP" for details. Subsequently, we also update the mineral physics interpretation in the new Extended Data Fig. 9.

Line 256: "Fe-Ni alloy with 4-9 wt% O, 0-4 wt% C, and 0-2 wt% H in the IC" – How do you get O (oxygen) in the Fe-Ni alloy? Isn't oxygen incompatible with the solid alloy (iron) and would preferentially stay in the melt? It isn't clear to me how this modeling was done or how the light element partitioning into the inner core would work.

Please see our reply to the question above for the O content in the Martian core.

Similar to O, the phase diagram of the Fe-C system exhibits a strongly dependence on pressure, temperature, and composition (please see the phase diagram of Fe-C below from Fei et al. [2014]. Based on Fei et al. [2014], the eutectic temperature of Fe-C system is 1600 K at 10 GPa and 1800 K at 20 GPa. We thus estimated that the eutectic temperature of the Fe-C system under Martian core pressure is ~2100 K. To form a solid inner core with the presence of S above 10 wt.% [Stewart et al., 2007], the temperature of the Martian inner core should be less than ~2000 K. Then under these conditions, the Fe-C system would not melt, and most carbon should be crystallized to the inner core. The precipitation process of carbon should be similar to that of O, with at most one-tenth of the carbon remaining in the outer core. The Fe-H system has a relatively low melting point, but detailed studies on how the eutectic temperature of the Fe-H system varies with pressure and hydrogen content are still lacking. Based on previous observations, we assume that $[H/Fe]_{\text{melt}}/[H/Fe]_{\text{solid}}=1.2$ [Okuchi, 1997].

In our modeling, we have considered the density and velocity jumps across the ICB assuming two formation mechanisms of the martial inner core [Stewart et al., 2007]. When the S content in the Martian core is above 14 wt.%, Fe_3S will be present in both liquid outer and solid inner core. In this case, the density and V_P jumps across the ICB are ~15% and ~13%, respectively. No matter how we change the light elements in the Martian core, we cannot satisfy the observed seismic density and V_P jumps. In the second scenario, Martian core contains less amount of S, ~10.6 wt.%. Most of S will leave on the liquid outer core. Assuming no other light elements in the Martian core but only S, the density and V_P jumps across the ICB are 14% and 5%, respectively. Adding light elements to the inner core can increase the magnitude of V_P jump across the ICB but decrease the density jump. When the H content in the Martian core is greater than 0.03 wt.% and the C content is 0.28 wt.%, the density and V_P change from the liquid Fe_3S outer core to a solid inner core will be 10.6% and 28.3%, respectively, consistent with our seismic observations (new Extended Data Fig. 8). With a slightly higher H content of 0.07 wt.%, we only need 0.04-0.06 wt.% C to produce a density jump of 8% and V_P jump of 25% across the Martian ICB. A higher S content of 12 wt.% in the Martian core requires a slightly greater H and C concentration in the inner core to satisfy the observed V_P and density jump across the ICB.

We also tried to take O into consideration. As showed in new Extended Data Fig. 9, incorporation of 4-9% O, 0-4% C, and 0-2% H can achieve density and velocity profiles which perfectly match our observed V_P and density of the IC (high density model of seismology observation). We agree with the reviewer that temperature is also important for the crystallization of inner core. Our seismic observation has provided strong evidence for the existence of the solid inner core. To solidify the inner core with ~10.6 wt.% S, a temperature of ~2000 K is required. This temperature is consistent with previous studies [Bertka and Fei, 1997; Hauck II and Phillips, 2002; Stewart et al., 2007; Williams and Nimmo, 2004] and approaches the lower limit of the estimated temperature of the Martian core in previous studies [Irving et al., 2023; Khan et al., 2023; Samuel et al., 2023; Stähler et al., 2021].

Based on a recent work on a more complicated but completed Fe-S-O-C [Yokoo and Hirose, 2024], we have further revised this section as in Lines 294-304. Please see the response for the next comment.

Phase diagram of Fe-C system from Fei et al.[2014] under 5, 10 and 20 GPa

Lines 262-283: What about the dynamical role of a postulated basal magma layer? Wouldn't this impede efficient heat loss across the martian CMB that would drive inner core crystallization? Interestingly enough, the Moon also is thought to have such a layer, and the core also appears to be layered? Is there a similar process at work there? And how do you rectify the thermal state of the core, where the BML would suggest a hotter core, and an inner core would suggest a colder core? Aren't these two hypotheses in conflict with each other?

We acknowledge this apparent contradiction between the low core temperature required for a solid inner core and the high temperature needed to maintain a silicate melt layer, which has also puzzled us. Interestingly, a recent study by Yokoo and Hirose [2024] on Fe-S-C-O alloys under Martian core conditions suggests that a more complex light element distribution, rather than a pure Fe-S system, can reconcile the existence of a solid inner core at high temperatures. This new finding aligns remarkably well with our results, as it satisfies both the velocity and density constraints for the inner core and supporting the feasibility of a basal magma layer. Based on this new study, we have expanded our discussion on the inner core composition and basal magma layer as in Lines 294-304.

The comment on the similarity between the Moon and Mars is very intriguing. However, given the large differences in pressure and temperature and the corresponding phase diagrams between the two, we feel that a direct comparison is difficult. Nonetheless, this is an excellent idea that warrants further exploration in a future study.

Methods:

Vespagram results: I am familiar with this method and have a few questions for the authors.

1) For a detection to be considered robust, how large of a time window and slowness window was searched? The plots in the main text (Fig. 2) and in the supplement (multiple figures of vespagrams) are very narrow slowness ranged (-6 to -8 s/deg for example). A robust detection would be a slowness that is well resolved and isn't surrounded by other arrivals at lower (or higher slownesses). Just showing the window for the predicted phase isn't enough, does the

vespagram capture other body wave phases at the predicted times? It's easy to get some "noise" in a window that looks like "signal" you are searching if you don't evaluate it against the other arriving phases (for example, I'd expect P to have a nicely constrained beam as well as other phases, but the window is so tight there is no way to evaluate the quality of the vespagram outside the window shown, nor is the P result shown). Your Supplement showing the analyses of PPPP and PPPPP is a good start (I appreciate the care given to show some other phases in the time window!), but I'd like to see at least one global (large range of time and slowness) vespagram showing the same for early arriving phases as well (P, PP, S, and surface waves where appropriate).

In the vespagram process, we examined a large window. However, in the original manuscript, we naively thought it would be more informative to show results in a smaller window. In the revised Fig. 2 and other vespagrams, we have now plotted them in a much larger window. For instance, the window for PKiKP in Fig. 2d-f has slowness ranging from -10 to 0 s° , and time span also expands to $550 \sim 700$ s, which allows us to better capture the characteristics of the target phases.

In the new Supplementary A Fig. S9, we also provide examples for other global phases, such as S (Fig. S9d) and ScS (on T-component, Fig. S9e). It is important to note that all phases are align on P, so the P phase has strong energy in the vespagram but shows zero slowness. However, we are not able to distinguish the PP or PPP energy, suggesting that their energy may not be consistent across different events, possibly due to complicated shallow structures.

When aligning on P, the vespagram for ScS show a little bit scattered energy around the predicted slowness and travel time. In contrast, when the waveforms are aligned on the S arrivals, the vespagram for ScS displays a much stronger arrival being consistent with the prediction in both slowness and travel time (Fig. S9f), which further support the effectiveness of our vespagram analysis.

2) Several of the vespagrams show multiple arrivals (e.g., PKKP in Fig. 2d) but power is normalized in the window to 1. In the supplement this is not done; and the other arrivals are similar in power to the one picked (e.g., P'P'r_ab in FS10 has multiple arrivals, and PKiKP in FS29a has two big arrivals). What determined your choice/selection of arrivals? Peak energy alone? Peak energy plus slowness? Either phase could be correct (or noise). The relative uncertainty in the following inversions should consider the uncertainties in these picks (and grow as a result), the bootstrap helps but doesn't necessarily avoid noise (you could be stacking noise, which isn't as easily suppressed if you use envelopes!). There also seems to be differences between what is presented in Fig. 2 and the supplemental vespagrams. What is the source of the amplitude discrepancy?

Thank you for this critical comment. In the original manuscript, we identified phases based on bootstrapping analysis by examining the peak energy around the target slowness in the mean vespagrams, which should also show small uncertainties in the corresponding 95% confidence level plot. For the uncertainty analysis, as shown in the old Fig. S23, we selected the energy peak in each vespagram during the bootstrapping tests. By repeating this process, we created a distribution of peaks and calculated the mean and standard deviation for travel time and slowness, respectively. However, as you and Review#2 commented, identifying

phases among those energy clouds can be challenging, particularly considering the noise effects, and our phase identification can be objective. In the old supplementary materials, we attempted to demonstrate that the picked phases are indeed a robust arrival, so we performed a lot of tests, such as focusing on events with small distance or events with small distance uncertainties, and dropping some particular events, which produced many vespagrams different from those in the old Fig. 2. Unfortunately, we did not present these tests clearly, and they may have been distracting.

In the revised manuscript, we have improved our approach in identifying phases and estimating the associated uncertainties, by following several recent studies on applying uncertainty estimation in array analysis with earthquake data [Ritsema et al., 2020; Yuan et al., 2021]. In each vespagram from the resampling tests, we now assign a value of 1 to grids with energy exceeding 85% of the peak, while others receive a value of 0. We then sum the grids valued at 1 and calculate their percentage occurrence across all tests. As shown in new Fig. 2c and Fig. 2e, this approach produces a much improved and more focused image for phase identification. We then focus on the window with the highest percentage occurrence and identify it as the candidate phase. Finally, we fit the distribution separately along the travel time and slowness axes with Gaussian functions (new Fig. 2e) to derive the uncertainties, represented by the 1σ values. We believe this approach offers a more robust phase identification and may also reduce the impact of noise. We have revised the “Methods” section and reorganized supplementary figures for improved clarity and presentation.

It is important to note that, despite our new approach, we still need to make a decision on which energy peak should be picked. For example, as shown in new Fig. 2e, there is another energy peak at a slowness of $-4s^{\circ}$ that arrivals at similar time as the identified PKiKP, but it has a lower percentage occurrence. While we cannot definitively determine the source of this seismic phase based on existing models, its similar slowness to ScS (new Supplementary A Fig. S22) suggests a potential relationship to ScS, such as ScS plus a segment of top reflection at a mantle interface to match the travel time. Although we cannot completely rule out other possibilities, such as noise or anomalous arrivals, our picked PKiKP shows the strongest energy in the vespagram, with the highest percentage occurrence in the bootstrapping test. We have included such a discussion as in the caption of Fig. 2d (Lines 155-157).

3) Azimuth resolution and the corresponding slowness resolution. In the Supplemental tests, the vespagrams degrade fairly rapidly with decreasing event count (thank you for showing this analysis), which is not unexpected; but given the low number of events in the stack for the vespa analysis, your slowness resolution will be primarily determined by the small number of events at different azimuths and distances (e.g., with a peak at 30-31 deg), and singular events at higher distances. I am less concerned about azimuth, but it seems like the exclusion of a singular distance can change the result quite a bit; the bootstrap handles this somewhat, but how do you equalize the events (the FPGA might help, but then again if a weaker event is flooded by noise, the event might not show a lower amplitude phase), given that the source magnitudes are not well constrained and attenuation of Mars is also poorly known?

This is a very good and important point. To address this comment, we performed an additional bootstrapping test (new Type II bootstrap resampling test in the “Methods”) by

randomly selecting half of the events from the distance range of 29–32°, along with events at other distances, to balance the number of events at different distances and generate the vespagram. This allows us to examine whether noise dominated records at this distance range could significantly affect the results. As shown in the new Supplementary A Fig. S15 for P'P'r_ab and Fig. S23 for PKiKP, the mean vespagram and the corresponding width of a 95% confidence interval confirm the clear presence of these core phases. Furthermore, as shown in Fig. S10, it is true that, as you noted, removing the 5 events with the farthest distance leads to some smearing of the energies for P'P'r_ab and P'P'n, causing a reduction in slowness resolution. However, the overall patterns of the vespagrams remains consistent with those obtained using all events, which suggests that the current approach produces stable results, at least for the 23 LF events included in the analysis.

Line 442, Methods: I was completely surprised (and a bit appalled) that the authors did not cite the Kim et al. 2021 InSight “Pitfalls” paper here. Their analysis is so thorough, why wasn't this paper referenced? The authors even go so far as to label some of the problems in the data as pitfalls(!), so it seems like there is some awareness of the findings in that essential paper for the martian seismologist. I would highly recommend the authors read this paper if they haven't, but I will qualify that recommendation in that their processing steps seem to have captured many of the essential problems that have been readily documented in the InSight data. Reference follows:

Doyeon Kim, Paul Davis, Ved Lekić, Ross Maguire, Nicolas Compaire, Martin Schimmel, Eleonore Stutzmann, Jessica C. E. Irving, Philippe Lognonné, John - Robert Scholz, John Clinton, Géraldine Zenhäusern, Nikolaj Dahmen, Sizhuang Deng, Alan Levander, Mark P. Panning, Raphaël F. Garcia, Domenico Giardini, Ken Hurst, Brigitte Knapmeyer - Endrun, Francis Nimmo, W. Tom Pike, Laurent Pou, Nicholas Schmerr, Simon C. Stähler, Benoit Tauzin, Rudolf Widmer - Schnidrig, William B. Banerdt; Potential Pitfalls in the Analysis and Structural Interpretation of Seismic Data from the Mars *InSight* Mission. *Bulletin of the Seismological Society of America* 2021;; 111 (6): 2982–3002.

Thank you for reminding this reference. We were aware this reference but inadvertently removed it from our reference list, due to the limitation on the number of references, and overlooked its inclusion in the supplementary materials during the editing processes. We have now properly cited this paper in the revised manuscript. Furthermore, considering the possible influence from glitches, we performed additional vespagram analysis on both the raw data and data only contains identified glitch signals (new Supplementary A Fig. S22). The results further support that the energy of the core phases identified on the vespagram is indeed a true feature.

Line 624: “it is impossible to have a solid inner core if the S content in the Martian core is greater than 12 wt.%" I'm not entirely sure if this is a constraint, higher sulfur contents for the Moon were proposed in Antonangeli et al., 2015; and the same models could extend to Mars. Why is it impossible? More recent literature doesn't seem to agree. It also seems like there is a strong dependence on the assumption of temperature as well (and the cited paper on Mars was published 2007, which is well before we had any seismology on the martian core). The presence

of a basal magma layer seems like it should put some constraint on the CMB, are there a reasonable set of temperature models then for the core that also satisfy the MOI constraints?

While Antonangeli et al. [2015] proposed higher S content for the Moon, directly applying that model to the Martian condition can be difficult due to the differing phase diagram of the Fe-FeS system at different pressure. At the Martian core condition, the eutectic composition at 35 GPa is 16 wt.% S. As shown in Stewart et al. [2007], with 12 wt.% S, the liquidus temperature would be ~ 1500 K, which is lower than the estimated Martian core temperature and also has difficulty to explain the presence of a basal magma layer. In contrast, the lunar core, with a eutectic composition of 37 wt.% S (21 wt.% S) at 5 GPa, is under a lower temperature. Therefore, having a solid inner core with S content in the Martian core greater than 12 wt.% is difficult, given the temperature and pressure conditions. However, as you noted, recent Insight results suggest otherwise and support a light element rich core. A recent publication by Yokoo and Hirose [2024] offers a solution by considering a more complex but completed Fe-S-O-C system, which can have a higher liquidus temperature than the Martian geotherm. This system allows for the crystallization of FeO and Fe₃C in the presence of small amounts of O or C in liquids, as favored by recent seismic studies. Moreover, we find that an inner core with FeO and Fe₃C can well explain our observed velocity and density (new Extended Data Fig. 9), and the elevated liquidus temperature makes the existence of the basal magma layer feasible alongside the inner core. We have rewritten this sentence as “Given that this temperature is much higher than the liquidus temperature for S content greater than 12 wt.%, the existence of a solid inner core under these conditions is unlikely.” at Lines 719-721. We have also expanded our discussion on the possible composition of the inner core and the implications of a basal magma layer based on the findings of Yokoo and Hirose [2024] at Lines 294-304.

Phase diagram of Fe-S system from (a) Stewart et al. [2007] for Mars and (b) Antonangeli et al. [2015] for Moon.

Referee #2:

This well-written and interesting paper examines seismic data from Mars, proposing to detect two signals influenced by a solid inner core. One of these signals is identified as a reflection from an inner core (PKiKP), the other as an early arrival of a phase which traverses the inner core twice (PKIKKIKP, described in the paper as PKKP). Inversions are performed to obtain best matching seismic velocity profiles, utilizing the travel times and the proposed interpretation of these signals. The results are consistent with an inner core of radius approximately 610 km, and a compressional wave velocity jump of 30% at the inner core boundary. As a result, the authors propose an inner core rich in light elements generated by a snowfall model. While I enjoyed reading the paper, and am very supportive of the interdisciplinary aspect, I feel that the overarching conclusions related to Martian inner core growth are a little too speculative. The most fundamental issue is that the seismic observations central to the premise of the paper do not appear robust in the way they are currently presented.

My primary concern is that the main observations (vespagrams in Figure 2) are not convincing in the form they are shown, and this is the underpinning result on which the rest of the paper's conclusions and inferences are formed. Envelopes are used due to difficulties in combining events with varying magnitudes, but the P amplitude could be used as a first approximation. Even the use of normalized amplitude retaining polarity would be an improvement, as the envelopes remove the polarity information which is required to identify reflections. Phase weighted stacking could be used in place of fourth root stacking. The vespagrams have unsuitable colour scales/palettes meaning they are difficult to evaluate as the smaller amplitude signals are masked. Colour scales should be sequential and even so as to image all signals, and these other signals should be identified as far as possible. Larger time/slowness windows should be used to place the signals in the wider context of others. This will provide a much more convincing and robust set of observations, and potentially more phases to include in the inversions.

Thank you for your insightful and constructive comments. As highlighted by you and Review#1, the robustness of the phase identification is the key here. We acknowledge that our original manuscript lacked organization in presenting materials. We agree that relying solely on using envelope in the vespagram analysis, especially with normalization, would lose important information on the relative amplitude and polarity. Our choice for the envelope over waveform is mainly driven its ability to produce more stable vespagram results. In the revised Supplementary A Fig. S8, we provide a synthetic vespagram example for the direct P, assuming events with different mechanism, depth, and source duration. While this example is somewhat extreme, it suggests that stacking envelope provides more stable result. Directly stacking the complicated phase arrivals from different events does not enhance the energy, even with phase weighted stacking.

Motivated by your and Review#1's insights on amplitude and polarity, we have tested vespagram analysis on both envelope and waveforms, after normalizing the whole traces on

the P or PKiKP amplitude. We are excited to report that it does work for measuring the amplitude ratio between PKKP and PKiKP, particularly when normalizing on the PKiKP (new Supplementary A Fig. S25). For the polarity analysis, we adjusted the traces to ensure consistent polarity among all P or PKiKP waveforms, and examine the polarities of other phases in the vespagram. In addition, we also tried deconvolving the PKiKP waveform from the PKKP waveform to remove possible source complexity and propagation effects at shallow depths. Remarkably, this deconvolution works very well and provides robust amplitude ratio between PKKP and PKiKP as well as their reversed polarities (new Extended Data Fig. 3), which agrees with the prediction. We are grateful for your and Review#1's comments, which has not only significantly improved the robustness of our results but also enhanced our understand of the marsquake data. We have revised the "Methods" section and supplementary materials to incorporate these new tests and adopted a more suitable color scale for the vespagram for better illustration.

Specific comments on Figure 2 and related observations:

- Amplitudes are presented as normalised envelopes meaning there is no information on polarity. This is especially important for identifying reflected signals.
- Amplitudes are normalised by panel and so there is no information on relative amplitude between panels.

Following your intriguing comments, we have performed vespagram analysis by normalizing on the P or PKiKP amplitudes. The amplitude ratio between PKKP and PKiKP is ~ 0.5 ($A_{PKKP}/A_{PKiKP} \sim 0.5$), as shown in new Supplementary A Fig. S25.

For polarity analysis, we first adjust all traces to ensure all P waveforms having positive polarity. However, this approach did not yield focused energy for PKiKP or PKKP (new Supplementary A Fig. S26), considering the effects of varying source mechanisms. Despite the difficulty of determining the polarity of the PKiKP, we also attempted to set the polarity of all targeted PKiKP to positive and analyze the vespagram for PKKP, but the results were not satisfactory.

Alternatively, we deconvolved PKiKP waveforms from the PKKP, and the resulting vespagrams show more focused energy (new Supplementary A Fig. S28). Importantly, this deconvolution allows us to simultaneously estimate the relative polarity and amplitude between PKKP and PKiKP. As shown in new Extended Data Fig. 3, PKKP and PKiKP exhibit reversed polarity, with an amplitude ratio of A_{PKKP}/A_{PKiKP} of ~ 0.5 . This amplitude ratio is also consistent with measurement from the single event S0235b (new Extended Data Fig. 7). However, we must admit that extracting amplitude and polarity information for these core phases are extremely challenging.

We have added a subsection of "Amplitude and polarity of PKKP and PKiKP" in the "Methods" section, along with the related Figures as well as discussion in the main text at Lines 255-261.

- There are lots of arrivals that are not identified, particularly obvious in panel d, but the choice of colour scale means that they cannot be discerned easily.
- A diverging colour scale is not suitable for the measurements here; a sequential scale should

be used. Otherwise, the small amplitude signals are mostly masked.

- The colour scales are uneven, so that the mid-point is not at the middle of the scale, which effectively masks smaller amplitude signals. The mid-point also varies between panels, and particularly noticeable for panel c, so they cannot be compared.

Thanks for these important comments. In the original manuscript, it was not our intention to mask signals with moderate amplitude; however, we mistakenly used a diverging color scale. In the revised manuscript, we have selected a sequential color scale and applied it to all vespagrams in both the main figures and supplementary materials.

- The time/slowness windows are too small, panels a-c in particular. Larger windows would be more informative.

This is similar to Review#1's comment. In the revised Fig. 2 and other vespagrams, we have now expanded them in a much larger window. For instance, the window for PKiKP in Fig. 2e has slowness ranging from -10 to 0 s°, and time span also expands to 550 ~ 700 s, which allows us to better capture the characteristics of the target phases.

- Panel a appears to have very poor slowness resolution (streaks) – the phase appears to continue down beyond -11.0 s/deg. Similar streaks also apparently in panel c but masked by colour scale.

Review#1 raised similar concerns. These streaks in the vespagram are true features that affect the phase identification. In the revised manuscript, we have improved the way for identifying phases with bootstrapping resampling test. In each vespagram from the resampling tests, we now assign a value of 1 to grids with energy exceeding 85% of the peak, while others receive a value of 0. We then sum the grids valued at 1 and calculate their percentage occurrence across all tests. As shown in new Fig. 2c and Fig. 2e, this approach produces a much clear and more focused image for phase identification. By focusing on the window with the highest percentage occurrence, we can more confidently identify the phases. Furthermore, by fitting the distribution, we provide a robust uncertainty analysis. We have modified the "Methods" section accordingly.

- Synthetic vespagrams presented alongside the observations would further help strengthen the figure, for example a comparison of a solid inner core vs a high velocity fluid inner.

Thanks for this great suggestion. We have added synthetic vespagram for the model with a solid inner core for PKKP and PKiKP in the new Fig. 2c and Fig. 2f, respectively. However, since there is no PKiKP for a high velocity fluid inner core, we did not include that synthetic vespagram in the main figure. Instead, we include the synthetic PKKP vespagram in new Supplementary A Fig. S13 for a model including both high velocity fluid inner core and molten silicate layer at the CMB.

- Vespagrams in supplement should be plotted using the same colour scale.

We have replotted all vespagrams with the same colour scale.

- Some of the supplementary vespagrams are not convincing, e.g. Fig. S21a is very streaky.

The original Fig. S21 presents mean vespagrams for two types of bootstrap resampling tests. In particular, in Fig. S21b, before slant stacking, we randomly shifted each trace within the range of -10 and 10 s, producing a streaky image that lose resolution on the slowness. However, we would argue that, even with this amount of time shift, the arrivals for the target phases are still quite strong. In the revised manuscript, we have reorganized the figures to improve the clarity.

Minor general comments:

- 49-51 Normal mode data and the observation of PKJKP provide the conclusive observational evidence that the Earth's inner core is solid.

Thank you for this important comment. We have rewritten this sentence as “The discovery of the IC-transiting seismic phase (PKIKP), which passes the core shadow caused by the liquid outer core (OC), has been critical in defining the Earth's IC structure, while normal modes and PKJKP observations further confirm the presence of a solid IC.” at Lines 50-53.

- x is not defined on Figure 2.

We have modified Fig. 2 and defined all symbols.

- I really appreciate the inclusion of the code, and in a format straightforward to run online, but it is not commented at all.

Referee #2 (Remarks on code availability):

I am not familiar enough with Python to provide a meaningful review for the code. It seems to run fine, but it is barely commented, and some of the comments are not in English.

Thank you for this important comment. We have included more comments to the code, and all comments are in English now.

References

- Antonangeli, D., G. Morard, N. C. Schmerr, T. Komabayashi, M. Krisch, G. Fiquet, and Y. Fei (2015), Toward a mineral physics reference model for the Moon's core, *Proceedings of the National Academy of Sciences*, *112*(13), 3916-3919, <https://doi.org:10.1073/pnas.1417490112>.
- Bertka, C. M., and Y. Fei (1997), Mineralogy of the Martian interior up to core-mantle boundary pressures, *Journal of Geophysical Research: Solid Earth*, *102*(B3), 5251-5264, <https://doi.org:10.1029/96JB03270>.
- Boehler, R. (1992), Melting of the FeFeO and the FeFeS systems at high pressure: Constraints on core temperatures, *Earth and Planetary Science Letters*, *111*(2), 217-227, [https://doi.org:10.1016/0012-821X\(92\)90180-4](https://doi.org:10.1016/0012-821X(92)90180-4).
- Breuer, D., T. Rueckriemen, and T. Spohn (2015), Iron snow, crystal floats, and inner-core growth: modes of core solidification and implications for dynamos in terrestrial planets and moons, *Progress in Earth and Planetary Science*, *2*(1), 39, <https://doi.org:10.1186/s40645-015-0069-y>.
- Fei, Y., and E. Brosh (2014), Experimental study and thermodynamic calculations of phase relations in the Fe–C system at high pressure, *Earth and Planetary Science Letters*, *408*, 155-162, <https://doi.org:10.1016/j.epsl.2014.09.044>.
- Hauck II, S. A., and R. J. Phillips (2002), Thermal and crustal evolution of Mars, *Journal of Geophysical Research: Planets*, *107*(E7), 6-1-6-19, <https://doi.org:10.1029/2001JE001801>.
- Hemingway, D. J., and P. E. Driscoll (2021), History and Future of the Martian Dynamo and Implications of a Hypothetical Solid Inner Core, *Journal of Geophysical Research: Planets*, *126*(4), e2020JE006663, <https://doi.org:10.1029/2020JE006663>.
- Irving, J. C. E., et al. (2023), First observations of core-transiting seismic phases on Mars, *Proceedings of the National Academy of Sciences*, *120*(18), e2217090120, <https://doi.org:10.1073/pnas.2217090120>.
- Kantor, A. P., I. Y. Kantor, A. V. Kurnosov, A. Y. Kuznetsov, N. A. Dubrovinskaia, M. Krisch, A. A. Bossak, V. P. Dmitriev, V. S. Urusov, and L. S. Dubrovinsky (2007), Sound wave velocities of fcc Fe–Ni alloy at high pressure and temperature by mean of inelastic X-ray scattering, *Physics of the Earth and Planetary Interiors*, *164*(1), 83-89, <https://doi.org:10.1016/j.pepi.2007.06.006>.
- Khan, A., D. Huang, C. Durán, P. A. Sossi, D. Giardini, and M. Murakami (2023), Evidence for a liquid silicate layer atop the Martian core, *Nature*, *622*(7984), 718-723, <https://doi.org:10.1038/s41586-023-06586-4>.
- Oka, K., K. Hirose, S. Tagawa, Y. Kidokoro, Y. Nakajima, Y. Kuwayama, G. Morard, N.

Coudurier, and G. Fiquet (2019), Melting in the Fe-FeO system to 204 GPa: Implications for oxygen in Earth's core, *American Mineralogist*, 104(11), 1603-1607, <https://doi.org:10.2138/am-2019-7081>.

Okuchi, T. (1997), Hydrogen Partitioning into Molten Iron at High Pressure: Implications for Earth's Core, *Science*, 278(5344), 1781-1784, <https://doi.org:10.1126/science.278.5344.1781>.

Ritsema, J., S. Kaneshima, and S. M. Haugland (2020), The dimensions of scatterers in the lower mantle using USArray recordings of S-wave to P-wave conversions, *Physics of the Earth and Planetary Interiors*, 306, 106541.

Samuel, H., et al. (2023), Geophysical evidence for an enriched molten silicate layer above Mars's core, *Nature*, 622(7984), 712-717, <https://doi.org/10.1038/s41586-023-0693>.

Seagle, C. T., D. L. Heinz, A. J. Campbell, V. B. Prakapenka, and S. T. Wanless (2008), Melting and thermal expansion in the Fe-FeO system at high pressure, *Earth and Planetary Science Letters*, 265(3), 655-665, <https://doi.org:10.1016/j.epsl.2007.11.004>.

Stähler, S. C., et al. (2021), Seismic detection of the martian core, *Science*, 373(6553), 443-448, <https://doi.org:10.1126/science.abi7730>.

Stevenson, D. J., T. Spohn, and G. Schubert (1983), Magnetism and thermal evolution of the terrestrial planets, *Icarus*, 54(3), 466-489, [https://doi.org:10.1016/0019-1035\(83\)90241-5](https://doi.org:10.1016/0019-1035(83)90241-5).

Stewart, A. J., M. W. Schmidt, W. van Westrenen, and C. Liebske (2007), Mars: A New Core-Crystallization Regime, *Science*, 316(5829), 1323-1325, <https://doi.org:10.1126/science.1140549>.

Williams, J.-P., and F. Nimmo (2004), Thermal evolution of the Martian core: Implications for an early dynamo, *Geology*, 32(2), 97-100, <https://doi.org:10.1130/g19975.1>.

Yokoo, S., and K. Hirose (2024), Melting experiments on Fe-S-O-C alloys at Martian core conditions: Possible structures in the O- and C-bearing core of Mars, *Geochimica et Cosmochimica Acta*, 378, 234-244, <https://doi.org:10.1016/j.gca.2024.06.027>.

Yuan, Y., D. Sun, W. Leng, and Z. Wu (2021), Southeastward dipping mid-mantle heterogeneities beneath the sea of Okhotsk, *Earth and Planetary Science Letters*, 573, 117151, <https://doi.org:10.1016/j.epsl.2021.117151>.

Dear Editor John VanDecar and Reviewers,

We greatly appreciate insightful and constructive comments from you and the three reviewers. We have made further significant changes, in particular, we further enhanced the robustness of the phase picks in the vespagram and significantly revised the discussion section. Motivated by Review#3's comments, we refined our vespagram analysis by amplifying the vertical or horizontal rectilinear motion and examined the effect of threshold choices on affecting the bootstrap occurrence percentage, which improve both clarity and reliability of our observations. Following Review#4's comments, we now offer a clearer interpretation of the Martian core composition and discuss scenarios in which inner core crystallization may not lead to a dynamo. While these revisions have strengthened both our results and interpretations, as all Reviewers emphasized, the primary goal of this manuscript remains the seismic detection of Martian inner core, with discussion provided to place our findings in the context of possible composition, dynamo and crystallization scenarios. Given the many uncertainties, we have kept our interpretations open and non-conclusive.

The quotes from the reviewers are followed by our responses in *blue italics*.

Referee #1:

As this is a re-review, I will focus on the changes made by the authors from the original manuscript. I want to thank them for so carefully addressing my numerous comments and suggestions (and those of the other reviewer). To me, this is a really interesting paper and I think that it will generate discussion for years to come!

First the vespagram analysis is now incredibly thorough, and the inclusion of synthetic tests demonstrates that the data have the ability to resolve the slowness of arrivals when examined in this approach. The addition of the synthetics was a really useful extra test, and I appreciate the significant extra work that was likely required to include it. By incorporating a bootstrap approach, the authors have also further demonstrated the robustness of the detections and quantified if an arrival is resolved or not. Personally, I like this approach, it gives a quantitative way to evaluate which arrivals are robust, and is a bog standard approach in seismology for detecting small amplitude seismic phases. The FDPA and filter bank detections of the waveform phases really knock home that the authors are finding energy at the right polarization, component of motion, and are self-consistent across events at greater distance with the source array analysis of the vespagrams. I feel like the seismic detections are pretty rock solid at this point (well as solid as single station approaches get), and feel like the authors have done above and beyond the work to show that there is energy where they indicate in the data. It's a great example of how a source array approach can improve the science return from a single station. Bravo.

The inversion of the data observations are also now much clearer, and the results are better articulated, and the statistics better explained with uncertainties tied to confidence intervals (95%) and standard statistical tools (bootstrap, rms, etc) are being applied. The original manuscript was a bit vague on the actual uncertainties, and I think that the authors have much better clarified that density and velocity range (and depth range) uncertainty in their detections. These values (R_ICB of 613 +/-67 km, dVP_ICB of 32 +/- 8%, $drho_ICB$ of 7 +/- 5%) are the primary conclusions of the paper, and I think that the authors have done a great job using statistical tools and Bayesian inversion to put constraints on the data observations.

I'm not a mineral physicist, but I think that the hypothesis of a light-element enriched inner core is a bit more thoroughly explored in this version and a bit better justified than in the prior version. I cannot comment on its validity, but it explored some of the new findings in mineral physics about the more complex C-H-O-S systems at relevant martian pressure and temperatures that I think was missing from the original manuscript. I think that this part of the paper is far more interpretative and less well constrained, but in my book that is okay, as I think that the seismology result is provocative by itself and might require some reconsideration about what happens inside planetary cores by the modelers and experimentalists. A subject matter expert might have more detailed comments for them on it.

We sincerely appreciate your generous remarks regarding the contribution of our paper. We are also grateful to you and the other reviewers for your insightful comments, which have substantially enhanced the robustness of our seismological observations. Many of the approaches and suggestions you provided have not only strengthened our current analysis but also offer valuable directions for future exploration of Martian seismic data. Thank you once again for your thoughtful and encouraging comments.

Line by Line Comments/Corrections:

Line 47: I think there is a typo here, S is repeated.

Have corrected.

Line 111: Predicted travel times from what model?

Line 112: Is the predicted slowness discrepancy from the uncertainties in the model (i.e., the slowness and travel time rely on know the velocity structure of the core/inner core which are unknown?)

Thank you for pointing this out, and we apologize for the lack of clarity at Line 111. The predicted travel times and slownesses are based on currently available seismic core models for Mars, including those proposed by Stähler et al. (2021), Irving et al. (2023), Khan et al. (2023), and Samuel et al. (2023). We have now clarified this information in the revised manuscript as new Lines 111-112, "derived from the available seismic core model^{7,20,33,34}".

You are right that the discrepancies in predicted slowness does come uncertainties in the available velocity models. As shown in Fig. S10 of Supplementary Information A, the predicted slowness values vary across these models. It is interesting to note that, compared to the P'P'n, the P'P'r_ab samples deeper part of the core and its uncertainties in slowness are much larger. This suggests that existing models differ most significantly in the deep part of the core, as shown in the Fig. S1. We have added such a description in the Caption of Fig. S10. "Notably, compared to the P'P'n, the P'P'r_ab samples deeper part of the core and its uncertainties in slowness are much larger. This indicates that existing models differ most significantly in the deep core (Fig. S1)."

Line 130: this line is confusing, why not just say 0.25 km/s spread over 100 km?

Sorry for the confusion here. We have revised this description in the revised manuscript.

Figure 2 (part g) the time scale for this panel is different from the panels above. Please plot on the same scale.

Thank you for this suggestion. We have modified the Fig. 2g to use the same time scale as the panels above.

Line 177: What would the prediction suggest here? You have synthetics you can draw from to quantify?

We apologize for the lack of clarity. Here, we refer to synthetic seismograms shown in Extended Data Fig. 3e-f, where the amplitude of PKKP is approximately half that of PKiKP, with $A_{PKKP}/A_{PKiKP} \approx 0.5$. To better illustrate this, we have added a dashed line at an amplitude of 0.5 and related text in Extended Data Fig. 3f in the revised manuscript.

Figure 3: Part b seems to be missing the colored text. What is the orange part? Part d has a typo, “tabel” is misspelled. To be honest, I don’t think you need part d at all, the information is in the table and text. Part c has a strange title that is cut off. It would be helpful to see the spread in the velocity fit at the best fitting IC radius (this is written about in the text).

Thank you for your detailed comments. The “colored text” in Part b represents the mean value and 85% confidence interval for the distributions shown by the orange histogram. In the revised manuscript, we have changed the orange color to red for better clarity.

In the figure caption, we also clarify that the red and gray histogram represent the marginal distributions of the inverted inner core radius (R_{IC}) and the priori distribution of the R_{IC} , respectively.

Regarding part d, we have corrected the typo (“tabel”), and following your suggestion, moved this panel to Extended Data Fig. 5, as its content is already represented in the main text and table. Additionally, we have removed the truncated title in part c.

We are not entirely certain we have fully understood the last comment, but we believe you are referring to Part c, specifically asking for a figure showing the velocity models with the best-fitting IC radius for each case. In fact, in Extended Data Fig. 5i, we have plotted the velocity profiles with the best fit for each case (1-8) as listed in Extended Data Table. 2. As shown in the Figure, although the initial mantle models may differ, the best inverted core models are quite similar. We hope this is clearer.

To further address this comment, we have also added a new Fig. S37k (Fig. R1) to show the marginal distributions of inverted core velocities sampled within the 25% confidence interval (interpreted as “best”, $R_{IC}=613 \pm 18$ km) of the inferred IC radius in Fig. 3b. As shown in Fig. R1, their distributions are not significantly different from those in Fig. S33, further indicating the core velocities are less resolved.

Fig. R1 (new Fig. S37k). Marginal distributions of inverted core velocities sampled within the 25% confidence interval of the inferred inner core (IC) radius ($R_{IC}=613 \pm 18$ km), using the M_{vesp} method based on SKS_GD model (case 1). (a) P velocity at the CMB (V_{P_CMB}), (b) P velocity of the outer-core side at the ICB ($V_{P_OC_ICB}$), and (c) P velocity jump at the ICB (δV_{P_ICB}). Mean values and 1σ are indicated with colored text at the top.

Line 227: Report the value from Irving et al, to illustrate the overlap in values

Thank you for this suggestion. We have modified this in the revised manuscript as: “the mean value of our inverted R_{OC} of 1799 ± 66 km (Supplementary A Fig. S33), is similar to 1,780-1,810 km in Irving et al. (2023).”

Line 267: maybe replace “form” with produce. Also, droplets sound like a liquid. Maybe crystals or metal particulates?

Thank you for this suggestion. Following Reviewer4’s comments, we have rewritten this section in the revised manuscript.

Line 267: The colder areotherm in direct conflict with a basal magma layer model (which requires a reasonably high temperature in the core, which would impede the snow model). You need a sentence here suggesting that the snow model can also explain the BML if the light element model allows for a hotter core. This was written about in the response to reviewers, but I did not see it discussed in the main manuscript.

Line 279: This is a pretty abrupt transition that there must be light elements in an inner core for this to work. Are there any other models aside from bottom up or top down crystallization? Like in an inner mush layer? What about anisotropy in the core itself (we do have some of that in the Earth’s inner core), could that throw up a relatively high contrast (you’re only sampling the core from one azimuth). How about energy loss, either from scattering or attenuation of PKiKP at the ICB that would make it weaker than PKKP that could be throwing off the reflection coefficient (and thereby underestimating the density jump)? What if the boundary isn’t sharp, and more of a gradient? Would that affect the estimate?

Thank you for these important comments. Since we have rewritten the entire section, we respond to both comments together here.

It is indeed true that a relatively hot core is necessary to support the existence of the BML. As suggested by Review#4, we have explored in greater detail the effects of different light elements on the seismic velocities and density of Martian inner and outer core in the revised manuscript. We demonstrate that an IC with FeO and Fe₃C can not only explain the observed velocity and density, and the elevated liquidus temperature under these conditions makes the existence of the BML compatible with the IC. While a FeO-rich IC favors a bottom-up crystallization, we do not directly address specific models of core solidification. Instead, we remain open to different possibilities, as resolving this question requires more detailed studies in the future.

We fully agree that many uncertainties can influence the amplitude of PKiKP and PKKP, thereby affecting our determination on the elastic properties contrasts across the ICB. Following your comments, we have added a sentence at the new Lines 250-252 to acknowledge this important comment as: “Furthermore, the potential presence of IC anisotropy and scatterers, similar to those observed in Earth^{9,39,40}, could also affect the accuracy of determining $\delta\rho_{ICB}$.”.

Line 621: exercise “caution”

Have corrected.

Line 1096: Martian is misspelled

Have corrected.

Referee #3:

This study is a very careful analysis of long period events observed on Mars by the InSight mission looking for phases associated with a potential inner core. The authors perform extremely careful data analysis, including all recommended pre-processing of a challenging dataset learned and published through the experience of the InSight team, and including a wide variety of techniques to pull out challenging phases to detect. The key observations are vespagram analysis of two phases identified as PKKP and PKiKP. The PKKP is significantly

earlier than predicted by a fluid core model consistent with previously published PKP observations, while the presence of a PKiKP phase is only possible with the existence of a solid inner core. The vespagram analysis is based on a source array approach for the significant number of observed marsquakes in the 27–40 degree distance range, which is a similar technique to that used to identify ScS, which was a key constraint on the size of the martian core. Beyond the vespagrams, the authors also identify these phases and other core phases in individual events using a range of techniques also previously employed on InSight data for identifying body wave phases. The authors then also look for other lines of evidence to support the interpretation of these phases as inner core sensitive by looking at polarization and amplitude ratio, which while difficult to perform, also seem to be consistent with the phase identification. In response to reviewers, they've expanded the presented vespagrams to cover a broader region of time/slowness space, making the picks of these phases more convincing. If true, the presence of an inner core on Mars could have strong implications on the thermal and chemical state of the core, which is a critical element of understanding the planetary evolution of Mars. While this data is challenging, and the observation still has some potential weaknesses I will discuss later, this is about as robust evidence as possible with the amount of data we have for Mars, and appears to be strongly suggestive of the existence of an inner core on Mars.

We appreciate your recognition of the multiple observations supporting our conclusion on the existence of the Martian IC, and we thank you for your encouraging words regarding our efforts to robustly detect weak core phases in the InSight data. We are also grateful for your thoughtful and important comments on the vespagram analysis, which inspired us to develop an exciting advanced approach, as discussed below, to improve the clarity and reliability of our observations.

The biggest concerns I have with the data analysis in the current manuscript relate to a) the existence of possible interfering energy at the arrival time of the PKiKP at greater slowness interpreted as a possible ScS-related phase, and the relatively low "occurrence percentage" derived from the bootstrap analysis of the PKKP phase.

First, for the energy interpreted as ScS-related energy, I suggest possibly trying to better analyze the polarization of the data. All vespagrams are "polarization-filtered" in this study, but I'll admit that I did not have the time to go through the code to see exactly how this was implemented. However, based on the description, this polarization filtering was done to emphasize linearly polarized signals. While both PKiKP and ScS would be linearly polarized (absent the effect of any shear wave splitting on ScS), the relative proportion of vertical and horizontal rectilinear motion should be very different between the two. Would it be possible to take advantage of this to more conclusively identify the origin of the energy at the two different slownesses in figure 2d, and therefore increase the confidence of the interpretation of the lower slowness arrival as PKiKP?

Regarding the first concern about potential interfering energy near the PKiKP arrival (at 604 s and -4 s $^\circ$) in Fig. 2d-e, we would first like to apologize for the confusion here. Our intention was not to identify this as an ScS phase directly, but rather to suggest that it may be related to an ScS phase plus a top-side S-to-P reflection at a mantle interface, given the similar slowness.

We found your suggestion to emphasize the vertical and horizontal rectilinear motion among different phases in the analysis particularly insightful. Inspired by this, we further process the data by multiplying the VRM-HRM (or HRM-VRM) ratio to the polarization-filtered waveforms and re-performed the vespagram analysis. As shown in Fig. R2 (same as new Fig. S24), this processing significantly enhances both PKiKP on the Z-component (Fig. R2f) and ScS on the T-component (Fig. R2g), compared to the original vespagram analysis (Fig. R2a-c).

Fig. R2 (new Fig. S24). Comparison of vespagram analyses with and without further modification with the difference between vertical rectilinear motion (VRM) and horizontal rectilinear motion (HRM). (a)–(c) Vespagram of PKiKP and ScS on the T, R, and Z components, respectively, without further polarization filtering. (d)–(f) Vespagram on the T, R, Z components aligned on the P arrival after multiplying the VRM–HRM factor. (g)–(i) Same as (d)–(f), but aligned on the S arrival and multiplied by the HRM–VRM factor. In panels (d)–(f), amplitudes are normalized to PKiKP, with adjusted color scales to accommodate low amplitudes on the T and R components. In panels (g)–(i), amplitudes are normalized to ScS.

Although the energy at (604 s, -4 s $^\circ$) is somewhat reduced on the Z-component using this new approach (Fig. R2f), it remains a relatively strong arrival, suggesting an incident P wave at the station. Furthermore, we generated synthetics for a model incorporating a low-velocity zone (LVZ) with $\delta V_p = -10\%$ and $\delta V_s = -15\%$ at a depth of 150 km. While this model produces a weak ScSS150P (ScS plus a segment of S-to-P reflection at the top of the LVZ), the amplitude is still not sufficient to match observations (new Fig. S25a). Interestingly, such a LVZ model does generate a stronger S-to-S reflection (ScSS150S), which may correspond to the slightly delayed energies at -4 s $^\circ$ on the horizontal components (Fig. R2d-e). Moreover, we recently work examining the late arrivals in the P-RFs and do find some signals that may be associated with a LVZ at a depth of ~ 100 km beneath the InSight station. However, we prefer not to overstate this interpretation. Rather, we suggest a localized, strong LVZ presents a plausible explanation for producing additional signals near PKiKP.

Furthermore, uncertainties in marsquake locations may also contribute to this signal. Synthetic tests by introducing random shifts within ± 5 s or ± 10 s to individual trace prior stacking produce scattered energies with consistent arrival times but varying slownesses (new Fig. S25b). This suggests that some of the observed signals near 600 s may originate from marsquake location uncertainties. Nonetheless, PKiKP remains a prominent arrival across all vespagrams. However, given the limited data, we are unable to conclusively determine the origin of the signal at 604 s and -4 s $^\circ$ at the current stage.

We have introduced the new approach in the Methods at Lines 527-532 and have included new Figs. S24-S25 and related discussion at Supplementary Information A Section 3.3. To avoid the confusion, we also changed the caption for Fig. 2 at Lines 149-150 as: “The energy at (604 s, -4 s $^\circ$) has a similar slowness to that of ScS (see Supplementary A Figs. S24-S25 and Supplementary Information A Section 3.3 for possible origins of this arrival).”

Second, for the bootstrap "occurrence percentage" plots, the PKKP arrival seems to peak at 25% of the bootstrap samples, which does not sound like a very high level of confidence that the signal is required by the data. I suspect the low value, though, is somewhat driven by the definition of occurrence percentage used, which only includes energy above 85% of the peak within the time-slowness space explored. Since this space includes at least 2 peaks (potentially

interpreted in the paper as reflections from the CMB and the top of a Mantle Silicate Layer), it seems likely that the relative amplitude of those peaks could vary greatly between bootstrap resamples, meaning sometimes one or the other could be more emphasized by this plotting approach. This gives me a little pause on the use of this approach, as it is obviously highly dependent on the space explored and the choice of threshold, leading me to question the use of it to quantitatively define the picks and uncertainties, although it does appear to be a decent tool to understand which features in the vespagram are most robust.

Regarding the second concern about the relatively low occurrence percentage (~25%) for the PKKP phase in the bootstrap analysis, we agree that the definition of “occurrence percentage” is sensitive to both the chosen slowness-time window and the energy threshold. As shown in Fig. R3 (same as new Fig. S16), we first examine the energy distributions within signal and selected noise windows, derived from the type I bootstrap resampling test (Fig. R3a). Then, we evaluate three candidate thresholds based on the maximum energy: 50% (below the noise level), 70% (just above the noise level), and 85% (approximately the first quartile of the signal distribution) (Fig. R3). As shown, with a threshold of 70%, the occurrence percentage of PKKP already reaches 70%. However, while lower thresholds increase detection frequency, they also result in scattered energy distributions (Fig. R3c). Nevertheless, the PKKP arrival remains the strongest arrival with the highest occurrence percentage, regardless of the threshold chosen.

Similar to PKiKP, we also applied the VRM-HRM scaling to the PKKP data and re-perform the vespagram analysis. However, this approach did not significantly enhance the signal, possibly due to the relative low amplitude of PKKP compared to PKiKP. Additionally, as a late arrival, PKKP may be contaminated by other phases as a late arrival, which complicates its robust identification. Nevertheless, despite these complications, we consistently observe energy focused around ~1,340 s.

Fig. R3 (new Fig. S16). Threshold sensitivity tests. (a) Average vespagram from 200 iterations of slant stacking by randomly resampling two-thirds of all 23 events (type I bootstrap resampling test). Red and blue rectangles mark the signal and selected noise window, respectively. (b) Energy distributions within the signal (red) and noise (blue) windows in (a) across all 200 stacks. Violin plots show kernel density estimates, with white dots indicating medians and gray rectangles spanning the 1st to 3rd quartiles. Black dashed lines mark candidate thresholds at 50%, 70%, and 85% of the maximum energy. (c) Occurrence percentages of the PKKP phase for each threshold (top to bottom: 50%, 70%, 85%).

Here's some more specific comments on the manuscript:

Line 47: "S, O, and H in addition to S" should presumably be "C, O, and H in addition to S"

Have corrected.

Line 94: Defining slowness as the reciprocal of ray parameter seems wrong. Most textbooks I know of define the ray parameter in units of slowness, so in horizontally layered media, the ray parameter is the horizontal slowness. Perhaps this is a convention that is not universal, though, as both $\sin \theta/v$ or $v/\sin \theta$ would remain constant in classic applications of Snell's Law.

Thanks for the clarification. You are correct that the ray parameter refers to the horizontal slowness and is only valid for a horizontally layered media. For a spherical model, the ray parameter should be $r \times \frac{\sin \theta}{v}$. To avoid confusion, we have removed "(reciprocal of the ray parameter)" in the revised manuscript.

Line 122: This is the first of a couple places where the authors make a point about the early arrival being too early for a pure liquid core. However, as the authors do correctly point out in other parts of the manuscript, the velocity and velocity gradient of the liquid core is only very weakly constrained at this time, and the early arrival of PKKP could be explained by a higher velocity gradient with depth which given the combined uncertainty of composition, temperature, and behavior of various chemical systems at martian temperature and pressure regimes is difficult to exclude a priori.

Thank you for highlighting the large uncertainties when we deal with the Martian deep structure. In fact, when we first see the PKKP data, our first intention is to explain it with a pure liquid core with a higher velocity gradient. As you pointed out, it will very difficult to rule out this possibility purely from the PKKP observations. However, later on, we confirm the presence of the PKiKP and other IC-related phases, which give a more robust indication of an IC. Thus, we modified the last sentence of this paragraph at Lines 132-135 as "However, having a much steeper velocity gradient toward the center may be difficult for a pure liquid core. Alternatively, an IC with a higher velocity, which is sampled by the PKKP, provides a more feasible explanation, further supported by the identification of other IC-related phases."

Line 181: The choice to not invert for mantle structure certainly does not "avoid possible issues with mantle heterogeneity and different measurement errors". It is a reasonable choice to make for this study, which is focused on core phases, and certainly the available data would not be enough to independently resolve mantle heterogeneity that may complicate measurements, but not attempting to use all data and invert for a whole Mars model, doesn't mean you're not sensitive to it. You're just trying make a reasonable simplifying assumption, while exploring some possible impact of this decision by using a range of possible models for the structure you don't invert for. I want to emphasize that I think this is the right and reasonable decision for this study, but I took a little issue with the statement that not inverting for mantle structure and using all available picks "avoided" the problem.

Thank you for this important comment. As you noted, our choice to fix the mantle structure (rather than jointly invert for it) is a pragmatic simplification, given the study's focus on core phases and the limited data available to resolve mantle heterogeneity. Moreover, the inverted IC radius is largely independent of our choice of mantle models, some of which differ significantly, further supporting the effectiveness of this simplified approach. However, we agree that a future inversion of the entire Martian interior structure, combining all available travel time picks, will be possible and important. We have revised the manuscript to clarify that this strategy aims to "minimize" rather than "avoid" the impact of mantle heterogeneity (Line 173).

In the discussion section, the authors emphasize recent studies of Fe-S-O-C systems and suggest

cores with significant amounts of C and/or O may be consistent with the presence of an inner core without requiring the core to be perhaps unreasonably low temperature. This was work I was not aware of, and the authors appear to make a convincing case that this is a possible model, which could even be reconciled with the existence of a molten silicate layer above the CMB, if that does exist as some studies have suggested. This is not my field of expertise, so I cannot assess these models thoroughly, but it does seem to set up reasonable spaces to explore for future mineral physics studies and geodynamic simulations.

Overall, while I do still think the measurements are challenging, this may be as robust an observation as possible without future seismic data, and definitely serves as a reasonable interpretation of the data that should drive future work and serve as motivation for possible future missions if possible.

I do not need to be anonymous, and this review is from Mark Panning.

Thank you once again for your thoughtful comments.

Referee #3 (Remarks on code availability):

I appreciate the presence of the code, but I unfortunately did not have the time to thoroughly review it. The included code appeared relatively readable, but seemed to only demonstrate the MCMC inversion. Unless I missed it, the code for the vespagram calculation and presentation was not included, and this seems more central to the paper than the velocity inversions. The key significance of the paper is the observation of inner core phases, and the available code I saw does not reproduce that analysis, only the model inversion.

Thank you for pointing this out. We have now included the code for the vespagram calculation and visualization, using Fig. 2f as an example. We agree that this part of the analysis is crucial to the result, and we appreciate your comment. The code is now fully documented to ensure reproducibility and clarity.

Referee #4:

Review of Bi et al.

This manuscript reports the seismic detection of the Martian inner core and argues possible core composition, in particular of the solid inner core. The presence of the Mars' inner core by itself has far-reaching implications, and I will be supportive of the publication of this paper as long as their seismological analyses are robust. Such robustness must be critically assessed by other referees, considering the fact that the presence of the Martian inner core has never been reported in a series of earlier seismological studies based on identical dataset. Since I am not a seismologist, my comments focus on the discussion part relevant to Mars' core composition and convection.

Thank you for your encouragement and critical comments. We have largely rewritten the discussion section based on your comments, and we believe it is now clearer and easier to understand.

First of all, the authors' arguments on the possible core composition are hard to follow unless readers are really familiar with the phase diagram of Fe alloyed with possible light elements. They discuss chemical composition (practically liquid composition), then crystallization, and finally consistency with seismological observations. Alternatively, they may first discuss which phase (Fe, FeO, Fe₃C, or FeH) matches the inner core observations.

They argued a mixture of FeO, Fe₃C, and stoichiometric FeH in the current ms, but indeed mixture (or layering) is not really likely. Yokoo et al. (2024) demonstrated that >4wt% C is necessary for Fe₃C crystallization to occur. It may not be feasible because of the known simultaneous solubility limit of S and C in liquid Fe. In addition, the crystallization of FeH is also unlikely because its melting temperature (in other words, crystallization temperature) is low below 40 GPa where the melting temperature of stoichiometric FeH is not a temperature maximum in the Fe-H liquidus phase diagram yet (Tagawa et al., 2022). These suggest the Martian inner core, if it really exists, is most likely to be single-phase FeO (not a mixture or layered). Considering that the inner core is composed only of FeO, they may argue the possible range of the liquid outer core composition, whose liquidus phase is FeO and density and velocity are consistent with observations. It is much simpler and readable.

According to your comments, we have made corresponding revisions to the manuscript. The specific modifications are as follows:

- 1. Based on previous studies, hydrogen (H) drastically lowers the melting temperature of Fe-H to below 1800 K under Martian core pressures (25-40 GPa) (e.g., Tagawa et al., 2022; Piet et al., 2021). Consequently, a solid inner core would be physically unsustainable under such conditions, which directly contradicts the seismologically inferred existence of a solid inner core. While we cannot entirely rule out trace amounts of H in the Martian core, our prior calculations demonstrate that such minimal H content would negligibly affect the core's density and seismic velocity, which are critical to reconciling with seismological observations. Consequently, in the revised manuscript, we have excluded hydrogen as a plausible component in our model of Martian core.*
- 2. Sulfur (S) is the most dominant light element in the Martian core, with a concentration of 6.6-16 wt.% (Stewart et al., 2007; Yoshizaki et al., 2020). Given the potential coexistence of multiple light elements in the Martian core, we have restructured the discussion into two steps in the revised manuscript, as suggested by you.*

First model: We considered that S and C are dominant light elements in the Martian core. When the C content exceeds 4.0 wt.%, O will reside in the outer core with S (Fei et al., 2014; Yokoo et al., 2024). With considering the partitioning of C between the outer and inner core together with a 12-16 wt.% S and 0-6 wt.% O, the density and seismic velocity jumps at the inner core boundary for a Fe-S-C-O composition would be 27-37% and ~22%, respectively, which fail to match our seismological constraints.

In the second model, we considered that O instead of C is abundant in the Martian core. According to the phase diagrams and calculations by Yokoo et al. (2024), we adopted a core composition of 12-16 wt.% S, 6.9-9.0 wt.% O, and 3.8-0 wt.% C. With this configuration, the predicted density and seismic velocity jumps at the ICB are 3-9% and 24-30% (Fig. R4), closely aligning with seismological observations. The revised figure is shown below.

Fig. R4 (Extended Data Fig. 8). Influence of light elements on the density (ρ) and velocity (V_P) jumps across the inner and outer core boundary assuming an O-enriched Martian core. The amount of O, C, and S and their partitioning between outer and inner core are from Yokoo et al. (2024). (a) Amount of O, C, and S in the Martian core used in our modeling; (b) Influence of O content on the density and velocity jumps across the inner and outer core boundary.

The parameters used in our calculations are as follows. In the model, densities and seismic velocities of solid FeO and Fe₃C were from Fischer et al. (2011) and Takahashi et al. (2019) (Table R1):

Table R1

	K_{T0} (GPa)	K_{T0}'	V_0 (Å ³)	θ_0 (K)	γ_0	q
Fe ₃ C	311(17)	3.4(1)	148.8(10)	314(fixed)	1.06(42)	1.92(173)
FeO	149(1)	3.60(4)	20.36(fixed)	417(fixed)	1.41(5)	0.5(fixed)

$$V_{\text{Fe}_3\text{C-solid}} = 1.09 \times \rho_{\text{Fe}_3\text{C}} - 1.79;$$

$$V_{\text{FeO-solid}} = [1.55 + 4.3 \times 10^{-5} \times (T - 300)] \times \rho_{\text{FeO}} - 2.03 - 5.6 \times 10^{-4} \times (T - 300);$$

In the liquid outer core, the effect of O, C, and S on the density and seismic velocity was calculated from a theoretical study (Huang et al., 2023).

$$\rho_{\text{Fe-S-O-C-liquid}} = 8.63 - 2.71 \times \text{Cont.}_C - 4.36 \times \text{Cont.}_O - 5.24 \times \text{Cont.}_S$$

$$V_{\text{Fe-S-O-C-liquid}} = 5.76 + 2.88 \times \text{Cont.}_C - 0.002 \times \text{Cont.}_O - 1.24 \times \text{Cont.}_S$$

Second, the last paragraph of the main text on the Martian dynamo includes almost nothing really meaningful. It is very important to discuss why the Martian core undergoes inner core crystallization but it does not drive liquid core convection. Does the authors show that upon

FeO crystallization, a residual liquid becomes depleted in oxygen and forms a dense liquid layer above the inner core? Or, it is also possible that crystallization is too slow and does not provide power large enough to drive liquid core convection. The authors can discuss the cooling rate of the Martian liquid core quantitatively. I believe such modeling is not difficult.

We acknowledge that the manuscript was not sufficiently clear about the relationship between core crystallization and dynamo generation. In the revised manuscript, we have largely re-written the last few paragraphs in such a way that will hopefully be clearer for readers. In particular, we have clarified that multiple conditions need to be met in order to drive a dynamo and that, even with ongoing crystallization, a dynamo is not guaranteed. For example, as you point out, inner core crystallization may be too slow and/or the residual liquid forming at the inner core boundary may not be buoyant—we now mention both of these possibilities explicitly in the revised manuscript. Regarding quantification of the core cooling rates, we want to avoid being too specific since a full thermal evolution model is complicated and depends on a number of uncertain parameters and initial conditions. The primary objective of this work is to confirm the existence of a Martian inner core through seismological observations. By combining mineralogical experiments with theoretical calculations, we demonstrate that the currently understood composition of Martian core materials can satisfy the observed density and velocity jumps between the inner and outer cores. While the specific mechanism of inner core formation is interesting, and important for understanding the Martian dynamo and the evolution of the planet's magnetic field, a detailed treatment of this topic is beyond the scope of the current manuscript. Nevertheless, we do discuss all of the above for context and argue that our results are consistent with the range of possibilities previously considered for core crystallization and dynamo evolution.

Specific comments:

Line 265~:

When one talks about crystallization at the top and “iron snow”, a dense Fe-rich phase descends and then melts away. It makes compositional stratification in a liquid core (depletion in light element in a deeper part). The solid inner core appears only after light element concentration in a liquid becomes low enough for solid Fe to crystallize in-situ at the centre. The process is more complicated than written here.

In the revised manuscript, we have considered the influence of O and C on the formation of solid inner core. Under these conditions, FeO or Fe₃C would precipitate as the primary phases (Yokoo et al., 2024). Please see our reply to the previous comment. We have largely rewritten this section based on your comments.

Line 275~ in the main text and Line 724~ in Methods:

They discuss the velocity of liquid stoichiometric Fe₃S (I believe it is not a typo), but liquid Fe₃S does not crystallize solid Fe at 35 GPa (Stewart et al., 2007). Moreover, why does a core liquid have a stoichiometric composition?

In the revised manuscript, we discuss the formation of the Martian core in the Fe-S-C-O system, as well as the density and velocity discontinuities between the inner and outer core. The new model shows better agreement with recent experimental and theoretical results. For the revised discussion on the composition of the Martian core, please refer to Lines 256-282.

Line 290:

FeO is not an alloy (metal) but a compound.

Have corrected.

Line 293:

“leaving only trace amounts of O and C in the liquid outer core”
Why? More explanations are necessary.

We have revised this section. In the current model, the partitioning of O and C between the inner and outer core is considered in both of our models (Yokoo et al., 2024).

Line 716:

“Sulfur (S) is considered to be the primary light element in the Martian core, with a content ranging from 10.6 wt.% to 16 wt.%”

Yoshizaki and McDonough (2020 GCA) proposed 6.6 wt% S.

Thanks. This whole section has been rewritten and moved to Supplementary A Section 7.

Line 758:

“VP of Fe₃C and FeH_x can be derived following the Birch’s law”

First of all, why not argue the velocity of FeO here? It is critical. Second, this equation is not for FeH_x but for stoichiometric FeH. Third, the Birch’s relation is likely to be temperature dependent (see Sakamaki et al., 2016 Sci.Adv.), and thus the authors should, at least discuss the effect of temperature.

In the revised manuscript, we considered only S, C, and O in the Martian core, as hydrogen significantly lowers the melting point and thus affects the formation of a solid inner core. In our previous response, we have elaborated on the effects of C and O on the density and seismic velocities of the inner and outer core. We kindly refer you to our earlier reply for details.

Extended Data Fig. 8:

The figure shows a difference by 0.01 wt% H makes a large difference in the density jump across the Martian ICB. But it is most likely wrong.

In the revised manuscript, we no longer consider the effect of hydrogen. The related discussions and figures have been revised or removed accordingly.

In addition, the main text always considers the 7% density jump across the ICB, but why 8–12% here?

According to our current model, the density jump is 3-9%, which matches the seismological observation of ~7%. We have made the corresponding revisions.

Extended Data Fig. 9:

“The content is indicated as mol percent”

It should be a fraction.

We have deleted this figure in the revised manuscript.

References:

Fei, Yingwei, and Eli Brosh. "Experimental Study and Thermodynamic Calculations of Phase Relations in the Fe–C System at High Pressure." Earth and Planetary Science Letters 408 (2014/12/15/ 2014): 155-62.

Fischer, Rebecca A., Andrew J. Campbell, Oliver T. Lord, Gregory A. Shofner, Przemyslaw Dera, and Vitali B. Prakapenka. "Phase Transition and Metallization of Feo at High Pressures and Temperatures." Geophysical Research Letters 38, no. 24 (2011).

Huang, Quancheng, Nicholas C Schmerr, Scott D King, Doyeon Kim, Attilio Rivoldini, Ana-Catalina Plesa, Henri Samuel, et al. "Seismic Detection of a Deep Mantle Discontinuity within Mars by Insight." Proceedings of the National Academy of Sciences 119, no. 42 (2022): e2204474119.

Irving, Jessica C. E., Vedran Lekić, Cecilia Durán, Mélanie Drilleau, Doyeon Kim, Attilio Rivoldini,

- Amir Khan, et al. "First Observations of Core-Transiting Seismic Phases on Mars." [In en]. *Proceedings of the National Academy of Sciences* 120, no. 18 (2023/05/02/ 2023): e2217090120.
- Khan, A., D. Huang, C. Durán, P. A. Sossi, D. Giardini, and M. Murakami. "Evidence for a Liquid Silicate Layer atop the Martian Core." [In en]. *Nature* 622, no. 7984 (2023/10/26/ 2023): 718-23.
- Mittelholz, A., C. L. Johnson, J. M. Feinberg, B. Langlais, and R. J. Phillips. "Timing of the Martian Dynamo: New Constraints for a Core Field 4.5 and 3.7 Ga Ago." *Science Advances* 6, no. 18 (2020): eaba0513.
- Piet, H., K. Leinenweber, E. Greenberg, V. B. Prakapenka, and S.-H. Shim. "Effects of Hydrogen on the Phase Relations in Fe-FeS at Pressures of Mars-Sized Bodies." *Journal of Geophysical Research: Planets* 126, no. 11 (2021): e2021JE006942.
- Samuel, Henri, Mélanie Drilleau, Attilio Rivoldini, Zongbo Xu, Quancheng Huang, Raphaël F. Garcia, Vedran Lekić, et al. "Geophysical Evidence for an Enriched Molten Silicate Layer above Mars's Core." [In en]. *Nature* 622, no. 7984 (2023/10/26/ 2023): 712-17.
- Stähler, Simon C., Amir Khan, W. Bruce Banerdt, Philippe Lognonné, Domenico Giardini, Savas Ceylan, Mélanie Drilleau, et al. "Seismic Detection of the Martian Core." [In en]. *Science* 373, no. 6553 (2021/07/23/ 2021): 443-48.
- Stevenson, David J. "Mars' Core and Magnetism." [In en]. *Nature* 412, no. 6843 (2001/07// 2001): 214-19.
- Stewart, Andrew J., Max W. Schmidt, Wim van Westrenen, and Christian Liebske. "Mars: A New Core-Crystallization Regime." *Science* 316, no. 5829 (2007/06// 2007): 1323-25.
- Tagawa, Shoh, George Helffrich, Kei Hirose, and Yasuo Ohishi. "High-Pressure Melting Curve of FeH: Implications for Eutectic Melting between Fe and Non-Magnetic FeH." *Journal of Geophysical Research: Solid Earth* 127, no. 6 (2022): e2022JB024365.
- Takahashi, S., E. Ohtani, T. Sakamaki, S. Kamada, H. Fukui, S. Tsutsui, H. Uchiyama, et al. "Sound Velocity of Fe₃C at High Pressure and High Temperature Determined by Inelastic X-Ray Scattering." [In English]. *Comptes Rendus Geoscience* 351, no. 2-3 (Feb-Mar 2019): 190-96.
- Yokoo, Shunpei, and Kei Hirose. "Melting Experiments on Fe-S-O-C Alloys at Martian Core Conditions: Possible Structures in the O- and C-Bearing Core of Mars." *Geochimica et Cosmochimica Acta* 378 (2024/08/01/ 2024): 234-44.
- Yoshizaki, Takashi, and William F. McDonough. "The Composition of Mars." *Geochimica et Cosmochimica Acta* 273 (2020/03/15/ 2020): 137-62.

Dear Reviewers,

We greatly appreciate the comments from you. We have removed most of the discussion on core cooling and dynamo processes from the main text and have also improved the overall clarity and presentation, following your suggestions.

The quotes from the reviewers are followed by our responses in *blue italics*.

Referee #1:

First, I want to thank the authors for all their hard work on responding to my many comments. I feel that their revisions and responses have made the article of the highest quality possible given the challenges in the InSight data. The supplement is incredibly thorough, and addresses many key questions to my satisfaction. I'm content with the paper as is (one minor cosmetic comment below) and don't have any additional questions/suggestions for the authors at this time.

I am happy to be identified as a reviewer.

Cheers,
Nick Schmerr

Thank you again for all the great suggestions and comments.

One minor comment: Figure 4: The colors used here don't match the implied mineralogy; for the Earth the upper mantle is blue, the transition zone aquamarine, and the lower mantle light blue, while for Mars the mantle is light blue (implying similarity to Earth's lower mantle, although Mars doesn't have a lower mantle). Shouldn't the implied mineralogy match (i.e., the martian mantle should be nearly wholly blue, with some aquamarine (post spinel) at the bottom? Or perhaps use a different shade of blue entirely given the differences in composition between the Earth and Mars?

Thank you for this great suggestion. Using the same color for the Martian mantle and Earth's lower mantle could lead to confusion. Since our focus is on the core, we think using a single color for the Martian mantle might be more appropriate. Following your suggestion, we have used a different blue color scheme for the Martian mantle in the revised Fig. 4 to clearly distinguish it from Earth's mantle.

Referee #3:

This is a re-review of the paper, and the overall summary of the key points of the paper remain as before: i.e. that the authors perform vespagram analysis of InSight marsquakes between 27 and 40

degrees distance and identify PKKP significantly earlier than predicted by existing liquid core models as well as a PKiKP phase indicative of a solid inner core at a radius of ~600 km. They argue that such a core is consistent with solidification of an O-rich inner core, although this conclusion (rather than a more C-rich core) relies on determination of a density contrast from an amplitude ratio observed on only one event, which I would argue is not a particularly strong constraint. I feel that the authors have addressed most of my major concerns from my previous review, and I would encourage this to be published after minor revisions.

Thank you for your thoughtful and important comments, which have helped clarify key scientific aspects and significantly improve the presentation.

I would say my most significant remaining concern is about the observation of the two possible PKKP arrivals interpreted as possibly arising from reflections off a Molten Silicate Layer (MSL) in addition to a reflection from the CMB. There are indeed 2 possible arrivals in the vespagrams, and it is interesting that they seem to correspond to the separation that would be expected for a 150 km layer. However, in each of the individual events analyzed for PKKP in the Supplementary Material B, only one PKKP arrival is identified, although it is not clear if the time window shown in those analyses actually would show the earlier potential PKKP arrival. The synthetics do show the amplitude of the reflection from the top of the MSL is stronger than the true CMB for those models, so it is perhaps not surprising that it may be harder to detect in individual events, but seeing both arrivals clearly in one or more events would help give confidence in seeing both. If that is not, however, possible, it might be good to call that out in the main text, suggesting that such an MSL is possibly consistent with the data, but not necessarily required (which I think is a reasonable interpretation).

We sincerely appreciate your thoughtful comments on the evidence for a potential Molten Silicate Layer (MSL). We fully agree that identifying two PKKP phases in individual events would provide stronger support for the presence of an MSL. However, as you pointed out, the small amplitude of PKKP make such detection challenging. As shown in the Fig. R1 (new Supplementary A Fig. S11), in fact, six events exhibit two arrivals within the time windows corresponding to the two slownesses in Fig. 2b. In addition, their high VRM-HRM values further support their potential identification. These observations appear to consistent with the presence of an MSL. However, as you noted, reliable identification of these phases in individual record remains difficult, so we have tried to be cautious in our interpretation.

We have rewritten the sentences in Lines 107-110 as “Thus, two arrivals at $1,290 \pm 3$ s and $1,341 \pm 5$ s in Fig. 2b may correspond to $PKKP_{CMB}$ and $PKKP_{MSL}$, respectively. While similar two arrivals are also observed for several individual events (Supplementary A Fig. S11), their relatively low amplitudes make the MSL interpretation less definitive.”

Fig. R1. Identification of candidate two PKKP arrivals in individual events. (a) Vespagram analysis for PKKP using all 23 events, showing two coherent energies at $\sim 1,290$ s ("Signal 1"; red cross) and $\sim 1,340$ s ("Signal 2"; red plus). (b) Polarization analysis of six events confirm the identification of the two signals in (a). In each panel: (top) bandpass- (grey line), (middle) polarization-filtered (blue line) waveforms with envelopes, and (bottom) vertical-horizontal summed FDPA intensity (VRM-HRM). Grey and blue dashed lines indicate the predicted travel times for the two PKKP phases at their respective slownesses from (a), with ± 5 s uncertainties denoted by shaded regions. Horizontal dash-dotted line marks the mean value of VRM-HRM in the 100 s window preceding PKKP ("Signal 2").

More minor typographical and presentational comments:
 Page 6, line 112: "models" should be plural here, not "model"

Have corrected.

Fig.2: Many labels in this figure are quite difficult to read, particularly the white font on the blue vespagram background. The font should likely either be enlarged or have a box behind it. Also, the ranges of the y axis on the VRM-HRM plot are a little strange. Is there a significance to the choice for the lower limit on the axis? Showing this down to 0 (or lower) may make it easier to tell how much above the noise level of this particular metric the identified signals are. If you can define a significance threshold based on the statistics of the variation of that metric, showing that as a dotted line would be preferable to simply cutting the plot off at that level. And if that level is not defined by the data, cutting it off there feels very arbitrary. I think this is a real observation, but the limited

range seems strange to me.

Thanks for these suggestions. We have made changes in Fig.2 following your comments.

(1) The white fonts in Fig. 2 have been enlarged for better readability.

(2) We have replotted the VRM-HRM results in the revised Fig. 2g-h. For each phase, we calculated the mean and standard deviation of the VRM-HRM values within a 100 s window preceding the target arrivals (1250–1350 s and 500–600 s for PKKP and PKiKP, respectively). The dash lines in the figures indicate the mean value plus one standard deviation. As shown in the Fig. 2g-h, both PKKP and PKiKP exhibit statistically significant amplitudes, with PKiKP appearing more prominent due to its larger amplitude, as further discussed in later sections.

Page 8, line 173: Once again, as stated in my last review, not inverting for mantle structure does not "minimize possible issues with mantle heterogeneity", but instead simply neglects it. I agree that this is the right decision for this study, but the text should reflect this. I would instead say that you look at the sensitivity to mantle structure by using a range of possible mantle models and achieve similar results, suggesting that not simultaneously inverting for mantle structure does not have a large impact on your results.

We apologize for not organizing the text clearly in the previous manuscript. We have revised the text in Lines 133-138 as: "Here, rather than performing an entire Martian velocity structure inversion by combining our new measurements with those from previous studies, we focus on inverting the P-wave velocity of the core (Methods). To assess sensitivity to mantle structure, we test a range of possible mantle models and obtain consistent results (Supplementary Information A Section 4), suggesting that not jointly inverting for mantle structure does not have a large impact on the final core model."

Page 12, line 248: I understand that this is the best constraint on density jump that you can achieve, but it is a very weak constraint. Individual reflections off an interface that may have topography can vary significantly due to focusing and defocusing. If you have lots of observations, that variation may cancel out and give you confidence in using amplitude ratios to solve for the reflection coefficient and therefore density jump, but I would argue using the ratio from only one seismogram could have errors of factors of 2, 3, or more, meaning that just getting an error bar on the density jump by only considering the possible error in determining the amplitude in the presence of noise, but still assuming a layer cake model for the reflection coefficient is vastly understating the uncertainty. You do already make reference to this in the text, but I would call it out again when discussing the possible core composition models. It is fine to prefer the O-enriched model to match this density contrast, but I think you need to clearly call out that the density jump could be very different, and so it is difficult to eliminate the C-rich model. While simply using that density jump would also likely lead to a model with too high a mass and too low moment of inertia if everything else is kept constant, it would likely be quite possible to solve for a model consistent with mass and moment of inertia if you inverted for whole planet structure. I do too think this is required for this study, but it should be acknowledged that the density constraint is not very strong.

Thank you for this important comment. While it is critical to have both δV_P and $\delta \rho$ to constrain

the composition of the Martian inner core, we totally agree that the estimation of the density jump is difficult and has large uncertainties due to the limited observations. In fact, on top of the single event observation, we do have another observation from the vespagram analysis (Extended Data Fig. 3), which show a similar ratio between PKKP and PKiKP as that for the individual event. Furthermore, as shown in the Supplementary A Fig. S48, to simultaneously fit the observed density and moment of inertia, a density jump less than ~10 % at the ICB is preferred. Although the vespagram analysis should also suffer those issues you mentioned and the calculation for moment of inertia also has large uncertainties, they tend to agree a small density jump is preferred.

In Lines 209-215, we have explained why a carbon-enriched inner core is not favored. Crystallizing a solid inner core enriched in carbon would require at least 4 wt.% C, which would result in density and velocity jumps across the ICB of approximately 20-27% and ~22% (Supplementary A Fig. S49), respectively. These values, particularly the density jump, are significantly larger than what we observe.

Although the $\delta\rho$ we determined carries certain uncertainties, which may introduce some errors in constraining the specific types and abundances of light elements in the Martian core, current high pressure-temperature experimental results, together with our $\delta\rho$ constraints, suggest that an O-enriched solid Martian inner core is more consistent. That said, we hope that future and improved observational data will provide better constraints on the composition of the Martian inner core.

To more clearly emphasize the most likely large uncertainties associated with determining the density jump, we have added a sentence at the end of the core composition section at Lines 229-231: "In addition, large uncertainties in constraining the velocity and especially the density jump across the ICB further complicate precise estimates of core composition."

This review is by Mark Panning.

Referee #3 (Remarks on code availability):

I did look at the code link, and I am happy to see that the authors have added in the vespagram codes that were missing from the last submission, but I did not attempt to review the code in detail.

Referee #4:

Re-review of Bi et al.

I am happy to see that the authors now argue for the FeO inner core, consistent with their observations. On the other hand, their discussion in pages 14-16 on core cooling and convection and Martian dynamo is still very poor and provides the least new insights (the same comment as I made in the previous round of review). I recommend the authors to limit their discussion only to the FeO inner core, which is directly related to their seismological observations reported in this paper. Further discussion written in pages 14-16 should be fully removed. More specific comments are found below.

Thank you for these important comments. We agree with you and the Editor that discussions

on core cooling and Martian dynamo are not the central to the main findings of this paper, which should focus on the seismological observations and their potential mineral physical interpretations. Accordingly, we have moved most of the discussion on the core dynamics to the Supplementary Information A Section 7.2, where it may still be of interest to a broader audience.

Major issues:

1) Page 13

The authors argue the inner core constituent (Fe, Fe₃C, or FeO) based only on the density and velocity “jumps”. However, the inner core VP profile is given Fig. 3a, and they should directly compare the observed VP and those of Fe, Fe₃C, or FeO at corresponding high pressure and high temperature. It seems that the VP of FeO matches the observations.

Thanks for your constructive suggestions. We have added the calculated velocity and density of solid Fe, Fe₃C, FeO in Fig. R2 (new Extended Data Fig. 8) for comparison. Compared to Fe₃C, the density and velocity of FeO are in better agreement with our observation (Fig. R2b).

We also revised the manuscript in Lines 220-221 accordingly: “Meanwhile, the absolute velocity of an FeO-rich solid IC is also consistent with our observed seismic velocity.”

Fig. R2. Influence of light elements on the density and velocity jumps across the ICB assuming an O-enriched Martian core. (a) From top to bottom: Amount of O, C, and S in the Martian core used in our modeling; Influence of O content on the density and velocity jumps across the inner and outer core boundary. The amount of O, C, and S and their partitioning between outer and inner core are from Yokoo et al., (2024). (b) Calculated density and velocity of Fe-light element alloys in the Martian inner core. Grey lines: seismic observations in this study (Case 8); green: Fe; blue: Fe₃C; red: FeO.

2) Page 14

The authors mention “the presence of a solid Martian IC would imply efficient core cooling if the core was very hot in the past”, but the presence of the inner core does not necessarily mean efficient core cooling (temperature could have been low from the beginning). The following discussion in Page 14 is not new/important and should be fully removed.

Thanks for your constructive suggestions. We have retained two sentences on the discussion of dynamics for Martian inner core in Lines 232-247, and the rest has been moved to the Supplementary Information A Section 7.2.

3) Page 15–16

Their discussion on Martian core convection and dynamo in the last 2 pages is not well written and not meaningful. They should remove the last three pages (pages 14–16) from the paper. Indeed, their finding of the FeO inner core is good enough for this paper.

As I requested in the previous review, they can briefly discuss why the FeO inner core crystallization does not drive liquid core convection. This is a very important question directly relevant to the FeO inner core, but the authors did not really respond to my request. The compositional buoyancy of liquid derived by crystallizing FeO at the inner core can be approximated by a comparison between the Martian outer core liquid density and the density of liquid FeO. The latter is obtained by its liquid equation of state reported by Morard et al. (2022 JGR) (see their Supporting Information S2).

Thanks for your suggestion. Most of the discussion on the Martian core dynamics has now been moved to the Supplementary Information A Section 7.2.

We agree that understanding the compositional buoyancy associated with FeO crystallizing is essential for assessing its role in driving liquid core convection. As in Morard et al. (2022), the density of the liquid FeO at 39 GPa and 4000 K is 5.94 g/cm³, lower than the outer core density at the ICB inferred from the SKS_GD model (6.3–6.75 g/cm³, Irving et al., 2023) and MSL_ETH model (6.77–7.11 g/cm³, Khan et al., 2023). If these estimates are correct, the residual fluid after FeO crystallization would be denser than the rest of outer core, implying it would not drive convection. However, given the substantial uncertainties in determining density of the Martian outer core from both seismological and mineral physics constraints, a comprehensive assessment of this effect remains challenging. Thus, we have chosen not to overinterpret this aspect in the current manuscript.

Nevertheless, I have included such a discussion in the Supplementary Information A Section 7.2 as: “In the bottom-up (Earth-like) IC growth regime, core crystallization could drive a dynamo if the IC grows rapidly and if the light element partitioning between the solid and liquid phases results in a residual fluid that is buoyant with respect to the rest of the OC [74,75]. For example, Morard et al. (2022) [79] report a liquid FeO density of 5.94 g/cm³ at 39 GPa and 4000 K, lower than OC density at the ICB inferred from the SKS_GD (6.3– 6.75 g/cm³) [2] and MSL_ETH (6.77– 7.11 g/cm³) [3] models. In such a case, the residual fluid after FeO crystallization would be denser than the surrounding core and unlikely to drive convection. However, large uncertainties in Martian core density estimates preclude a definitive assessment.”

Minor comments:

4) Line 266–268

Please revise the text into “the crystallization of an Fe₃C-dominated solid IC requires > 4 wt.% C

in the liquid core, resulting in an Fe-S-O-C outer core”.

We have rewritten this sentence in Lines 209-211: “Previous experimental results suggest that the crystallization of an Fe₃C-dominated solid IC requires > 4 wt.% C^{45,46}, resulting in an Fe-S-O-C outer core”

5) Line 268–269 & 271–272

“such a composition (12-16 wt.% S, 0-6 wt.% O, and 4.0-4.7 wt.% C)”

This composition suddenly appears, which confuses readers including myself. I realized that this specific range of composition is from Fig. 7a in Yokoo & Hirose (ref. 21) by considering the volume of the inner core observed in this study. This should be explicitly mentioned.

“12-16 wt.% S, 6.7-9.0 wt.% O, and \leq 3.8 wt.% C”

Same comment as above.

Thanks for your constructive suggestions. We have revised the manuscript in Lines 211-217: “Under such conditions, the Martian core is estimated to contain approximately 12-16 wt.% S, less than 6 wt.% O, and 4.0–4.7 wt.% C²¹. However, such a composition would, upon crystallization, produce density and velocity jumps across the ICB of 20-27% and ~22%, respectively (Supplementary A Fig. S49), which are inconsistent with our observations. In contrast, models with 12-16 wt.% S, 6.7-9.0 wt.% O, and \leq 3.8 wt.% C²¹ favor the crystallization of an FeO-rich solid IC under Martian core pressure-temperature conditions.”

6) Line 278–279

“an O-enriched core can crystallize even at elevated core temperatures exceeding 2200 K at the ICB, potentially supporting the existence of a MSL above the core”

This statement is correct but is hard to understand. Please add more explanations. It is possible that the Martian core temperature is higher than the estimate by Khan et al. (2022), which was employed by Yokoo & Hirose (2024) who proposed these C-rich and O-rich possible Martian outer core compositions. If this is the case, a liquid core containing >9.0 wt% O can crystallize FeO when >2200 K at the centre (see Fig. S11 in Yokoo & Hirose, 2024). Such high core temperatures might support the existence of a MSL above the core.

Thank you for this suggestion. We have revised the manuscript in Lines 223-227: “While Martian core temperature estimates remain uncertain, the liquidus of the Fe-O-C-S system suggests that even if the core temperature is ~10% higher than current estimates, FeO can still crystallize to form a solid IC²¹. Such high core temperature might also support the existence of a MSL at the CMB^{33,34}.”

7) Line 308–309

“Additionally, even if there is some ongoing compositional convection, the presence of stable thermal stratification in the outermost parts of the core may inhibit dynamo action.”

This is not true. Such thermal stratification due to high thermal conductivity has been proposed to explain the observed 300-km thick low-velocity layer atop the present-day outer core of the Earth.

It does not inhibit a dynamo action.

This is a good point. We have deleted this description in the current Supplementary Materials.

References:

1. G. Morard, D. Antonangeli, J. Bouchet, A. Rivoldini, S. Boccato, F. Miozzi, E. Boulard, H. Bureau, M. Mezouar, C. Prescher, S. Chariton, and E. Greenberg. *Structural and Electronic Transitions in Liquid FeO Under High Pressure. Journal of Geophysical Research: Solid Earth*, 127(11), e2022JB025117 (2022), <https://doi.org/https://doi.org/10.1029/2022JB025117>